



# CMEMS-LSCE: A global 0.25-degree, monthly reconstruction of the surface ocean carbonate system

Thi-Tuyet-Trang Chau[1], Marion Gehlen[1], Nicolas Metzl[2], and Frédéric Chevallier[1]

[1]Laboratoire des Sciences du Climat et de l'Environnement, LSCE/IPSL, CEA-CNRS-UVSQ, Université Paris-Saclay, F-91191 Gif-sur-Yvette, France

[2]Laboratoire LOCEAN (IPSL), Sorbonne Université, CNRS–IRD–MNHN, Paris, F-75005, France

**Correspondence:** Thi-Tuyet-Trang CHAU (trang.chau@lsce.ipsl.fr, thi.tuyet.trang.chau@gmail.com)

**Abstract.** Observation-based data reconstructions of global surface ocean carbonate system variables play an essential role in monitoring the recent status of ocean carbon uptake and ocean acidification as well as their impacts on marine organisms and ecosystems. So far ongoing efforts are directed towards exploring new approaches to describe the complete marine carbonate system and to better recover its fine-scale features. In this respect, our research activities within the Copernicus Marine Environment Monitoring Service (CMEMS) aim at developing a sustainable production chain of observation-derived global ocean carbonate system datasets at high space-time resolution. As the start of the long-term objective, this study introduces a new global $0.25°$ monthly reconstruction, namely CMEMS-LSCE, for the period 1985-2021. The CMEMS-LSCE reconstruction derives datasets of six carbonate system variables including surface ocean partial pressure of $CO_2$ ($pCO_2$), total alkalinity ($A_T$), total dissolved inorganic carbon (DIC), surface ocean $pH$, and saturation states with respect to aragonite ($\Omega_{ar}$) and calcite ($\Omega_{ca}$). Reconstructing $pCO_2$ relies on an ensemble of neural network models mapping gridded observation-based data provided by the Surface Ocean $CO_2$ ATlas (SOCAT). Surface ocean $A_T$ is estimated with a multiple linear regression approach, and the remaining carbonate variables are resolved by $CO_2$ system speciation given the reconstructed $pCO_2$ and $A_T$. $1\sigma$-uncertainty associated with these estimates is also provided. Here, $\sigma$ stands for either ensemble standard deviation of $pCO_2$ estimates or total uncertainty for each of the five other variables propagated through the processing chain with input data uncertainty. We demonstrate that the $0.25°$-resolution $pCO_2$ product outperforms a coarser spatial resolution ($1°$) thanks to a higher data coverage nearshore and a better description of horizontal and temporal variations in $pCO_2$ across diverse ocean basins, particularly in the coastal-open-ocean continuum. Product qualification with observation-based data confirms reliable reconstructions with root-of-mean–square–deviation from observations less than $8\%$, $4\%$, and $1\%$ relative to the global mean of $pCO_2$, $A_T$ (DIC), and $pH$. The global average $1\sigma$-uncertainty is below $5\%$ and $8\%$ for $pCO_2$ and $\Omega_{ar}$ ($\Omega_{ca}$), $2\%$ for $A_T$ and DIC, and $0.4\%$ for $pH$ relative to their global mean values. Both model-observation misfit and model uncertainty indicate that coastal data reproduction still needs further improvement, wherein high temporal and horizontal gradients of carbonate variables and representative uncertainty from data sampling would be taken into account in priority. This study also presents a potential use case of the CMEMS-LSCE carbonate data product in tracking the recent state of ocean acidification.



## 1 Introduction

Between 1750 and 2019, the ocean took up an estimated $25\%$ (or $170\pm20$ PgC) of total cumulated anthropogenic $CO_2$ ($685\pm75$ PgC) emitted to the atmosphere (IPCC AR6 - the Sixth Assessment Report of the United Nations Intergovernmental Panel on Climate Change, Canadell et al., 2021). While the uptake of anthropogenic $CO_2$ mitigates global warming it also profoundly modifies seawater chemistry in a suite of well-understood reactions (Orr et al., 2005) leading to an increase in hydrogen ion concentration ($[H^+]$), as well as a decrease in carbonate ion concentration ($[CO_3^{2-}]$) and in the saturation state of seawater ($\Omega$) with respect to calcium carbonate minerals ($CaCO_3$). The increase in hydrogen ion concentration ($[H^+]$) is commonly reported as a decrease in $p$H ($p$H = - $\log[H^+]$) and referred to as ocean acidification.

Changes in carbonate chemistry impact calcifying plankton and benthos as a direct result of decreasing seawater saturation state with respect to $CaCO_3$ (Fabry et al., 2008; Thomsen et al., 2015). Ocean acidification also modifies the production of marine trace gases exchanged at the air-sea interface (Hopkins et al., 2020), the availability of nutrients fueling primary production (Doney et al., 2009), as well as the speciation of pollutants (Millero et al., 2009; Hoffmann et al., 2012). These chemical changes interact with warming and ocean deoxygenation to drive major changes in marine ecosystems (Doney et al., 2020) and to alter global biogeochemical cycles with the potential for feeding back on radiative forcing (Gehlen et al., 2011; Hopkins et al., 2020). The likelihood for major disruptive impacts of ocean acidification on marine ecosystems, if future $CO_2$ emissions were to go unabated, is reflected by the Sustainable Development Goal 14.3 (SDG 14.3) - "Reduce Ocean Acidification: minimize and address impacts of ocean acidification" (https://www.globalgoals.org/14-life-below-water, last access: 20/03/2023). Albeit not specifically mentioned, moving towards SDG 14.3 implies the understanding of historical and contemporary carbonate chemistry, its mean state, trends and variability.

Earth system models have been widely used to track changes in ocean $p$H over the historical period and to project its future evolution under different $CO_2$ emission pathways (Bopp et al., 2013; Gattuso et al., 2015; Kwiatkowski et al., 2020; Cooley et al., 2022; Jiang et al., 2023). The present-day global surface ocean $p$H is roughly $0.1$ $p$H units less than at the beginning of the industrial era (Gattuso et al., 2015; Jiang et al., 2019) corresponding to an increase in hydrogen ion concentration of $26\%$ (Doney, 2010). By the end of the $21^{st}$ century, the $p$H is projected to decrease by $0.16\pm0.002$ $p$H units in response to the IPCC AR6 low emission scenario (SSP1-2.6), respectively by $0.44\pm0.005$ $p$H units in response to the IPCC AR6 high emission pathway (SSP5-8.5) relative to 1870–1899 (Kwiatkowski et al., 2020). Understanding impacts on marine biota requires to move towards finer spatial and temporal scales than resolved by the current generation of Earth system models (Torres et al., 2021), as well as to expand the analysis from $p$H to other carbonate system variables such as the saturation state with respect to calcium carbonate minerals and the buffer capacity. The development and implementation of environmental management strategies equally rely on understanding and attributing the variability of the carbonate system from diurnal to decadal time scales to underlying physical-chemical-biological processes.

In situ time series have played an important role in monitoring ocean acidification over the last decades (Bates et al., 2014; Lauvset et al., 2015; Sutton et al., 2019; Pérez et al., 2021; Leseurre et al., 2022; Skjelvan et al., 2022). At these sites, seawater $p$H has been either directly measured or calculated from measurements of other carbonate system variables. These variables





include surface ocean partial pressure of $CO_2$ ($pCO_2$), total alkalinity ($A_T$), and dissolved inorganic carbon (DIC). While changes in time series of carbonate system variables well reflect impacts of enhanced anthropogenic $CO_2$ uptake on ocean chemistry at a local scale (Steinberg et al., 2001; González-Dávila and Santana-Casiano, 2009; Dore et al., 2009; Bates et al., 2014; Pérez et al., 2021), the reliable upscaling to large ocean regions or entire basins requires a significant extension of the existing observing network (Lauvset et al., 2015; Bakker et al., 2016; Sutton et al., 2019; Lauvset et al., 2022).

Time series data are completed by bottle data from international cruises. These data are synthesized by the Global Ocean Data Analysis Project v2.2022 (GLODAPv2.2022) and include about $1.4$ million measurements of surface-to-interior ocean $p$H, $A_T$, DIC, and other parameters (Lauvset et al., 2022, https://www.glodap.info/, last access: 30/9/2022). Likewise, underway measurements of near-surface $CO_2$ fugacity, i.e., $pCO_2$ corrected for non-ideal gas behavior, are compiled in the Surface Ocean $CO_2$ Atlas (SOCAT) since its first release in 2011 (Pfeil et al., 2013). That latest version SOCATv2022 yields approximately $33.7$ million high-quality controlled data (Bakker et al., 2022, http://www.socat.info/, last access: 17/6/2022). Despite millions of observations available, data coverage is still modest, e.g., $CO_2$ fugacity samples over the global ocean cover less than $2\%$ of its surface for each month in the last three decades (Bakker et al., 2016; Hauck et al., 2020). Mapping methods have become an essential tool in ocean carbon cycle research allowing to extrapolate these sparse measurements into space-time varying fields of carbonate system variables (e.g., Rödenbeck et al., 2015) and used for global carbon budget estimates (Friedlingstein et al., 2022).

Recent years have seen the rapid development of machine learning approaches to map global surface ocean $pCO_2$ (see Rödenbeck et al., 2013; Landschützer et al., 2016; Denvil-Sommer et al., 2019; Gregor et al., 2019; Chau et al., 2022b, for instance). Thanks to these efforts, the carbon cycle community can now draw on an ensemble of reconstructions for the observation-based assessment of the ocean carbon sink (Friedlingstein et al., 2022). However, only a few global observation-based reconstructions are available for $p$H, $A_T$, DIC, and $\Omega$ with respect to calcite and aragonite (see Gregor and Gruber, 2021, for a review). The reconstruction of global distributions of these variables is hampered by an insufficient amount of direct measurements (Bakker et al., 2016; Lauvset et al., 2022). Alternatively, the complete carbonate system can be obtained by speciation given the information of any couple of $pCO_2$, $p$H, $A_T$ or DIC together with chemical (e.g., phosphate, silicate, nitrate) and physical variables (e.g., temperature, salinity), as well as corresponding dissociation constants (Park, 1969; Lewis and Wallace, 1998; Dickson et al., 2007).

Regardless of the developments in different observation-based estimation methods, Takahashi et al. (2014), Iida et al. (2021), and Gregor and Gruber (2021) propose global climatologies or monthly varying fields of all variables of the carbonate system, i.e., $pCO_2$, $p$H, $A_T$, DIC, and $\Omega$. These data products have a spatial resolution of $1°$ ($\sim 100\text{km} \times 100\text{km}$) or even coarser. Nevertheless, the variations of carbonate system variables over the coastal regions where their instantaneous gradients are driven by smaller-scale features like ocean upwelling, wind turbulence, eddies, water runoff, and sharp biological productivity (Jones et al., 2012; Bakker et al., 2016; Laruelle et al., 2017) are poorly described at such spatial resolutions. Here we improve on existing studies by providing a global $0.25°$, monthly observation-based surface ocean carbonate system product consisting of datasets of six core variables of the marine carbonate system (see Table 1 and Appendix A for definitions) and their associated $1\sigma$-uncertainty. This high-resolution data product covers the years from 1985 to 2021. Laboratoire des Sciences du Climat et de



l'Environnement (LSCE) is in charge of the product within the European Copernicus Marine Environment Monitoring Service (CMEMS). Our product is referred to as CMEMS-LSCE hereafter.

The reconstruction of surface ocean carbonate system variables starts with the reconstruction of surface ocean $p\mathrm{CO}_2$ and $\mathrm{A_T}$ in each regular grid of $1\mathrm{month} \times 0.25° \times 0.25°$. Next, variables $p\mathrm{H}$, DIC, and $\Omega$ are derived by speciation. Advantages of the combination of $p\mathrm{CO}_2$ and $\mathrm{A_T}$ over others for the speciation of the carbonate system are: (1) $p\mathrm{CO}_2$ is the most extensively measured parameter, (2) $\mathrm{A_T}$ can be accurately predicted from salinity, temperature, and nutrient concentrations, and (3) the combination of these two prior variables results in the slightest uncertainty of $p\mathrm{H}$ estimates (Zeebe and Wolf-Gladrow, 2001; Lauvset and Gruber, 2014; Takahashi et al., 2014; Orr et al., 2018). The three main successive modules used in the CMEMS-LSCE production chain are summarized as follows.

**Table 1.** CMEMS-LSCE carbonate system variables.

| Standard names | Notations | Units |
|---|---|---|
| 1. Partial pressure of $\mathrm{CO}_2$ in surface seawater | $p\mathrm{CO}_2$ | µatm |
| 2. Total alkalinity in surface seawater | $\mathrm{A_T}$ | $\mathrm{µmol\,kg^{-1}}$ |
| 3. Surface ocean dissolved inorganic carbon | DIC | $\mathrm{µmol\,kg^{-1}}$ |
| 4. Surface seawater $p\mathrm{H}$ reported on total scale | $p\mathrm{H}$ | - |
| 5. Saturation state for surface seawater with respect to aragonite | $\Omega_{ar}$ | - |
| 6. Saturation state for surface seawater with respect to calcite | $\Omega_{ca}$ | - |

    i) *Reconstruction of $p\mathrm{CO}_2$* (Sect. 3.1): a modified version of the CMEMS-LSCE-FFNN approach (Chau et al., 2022b) is applied to map gridded datasets of SOCATv2022 $\mathrm{CO}_2$ fugacity and predictors in order to reconstruct $p\mathrm{CO}_2$ at a spatial resolution of $0.25°$ for every month in the period 1985-2021 (444 months in total). The CMEMS-LSCE-FFNN works

on an ensemble of 100 feed-forward neural networks (FFNNs). By design, 100-member ensemble model outputs allow to yield the best model estimate (i.e., ensemble mean) and model uncertainty (i.e., ensemble standard deviation) for surface ocean $p\mathrm{CO}_2$ in each grid cell and each month. The primary modification of this study's version and the original CMEMS-LSCE-FFNN (Chau et al., 2022b) is an increase of 16-fold in model spatial resolution.

    ii) *Reconstruction of $\mathrm{A_T}$* (Sect. 3.2): locally interpolated alkalinity regression (LIAR; Carter et al., 2016, 2018) is chosen

to estimate total alkalinity on regular grids of $1\mathrm{month} \times 0.25° \times 0.25°$ over the global surface ocean for the years 1985-2021. LIAR works with multiple linear regression models, each representing a combination of predictor variables. The best linear model, which has the lowest prediction error among the others, is retained for the final estimation of



$A_T$. Various reconstruction methods for $A_T$ exist (see Carter et al., 2016; Broullón et al., 2019; Gregor and Gruber, 2021, for a review), but we choose LIAR due to its global applicability, simplicity in setting, and accuracy compared
to other published approaches (Carter et al., 2018; Gregor and Gruber, 2021). Importantly, LIAR allows determining reconstruction uncertainty propagated from multiple sources of input uncertainties at desired model resolutions.

iii) *Reconstruction of pH, DIC, and saturation states with respect to aragonite* $(\Omega_{ar})$ *and calcite* $(\Omega_{ca})$ (Sect. 3.3): CO2SYS (Lewis and Wallace, 1998; Van Heuven et al., 2011) is a standard software used for the speciation of carbonate parameters in the marine $CO_2$ system (see Olsen et al., 2016; Bresnahan et al., 2021; Gregor and Gruber, 2021; Woosley, 2021, for
a few). The CO2SYS speciation is built on a set of equilibrium equations (Dickson et al., 2007; Dickson, 2010). Given the reconstructed $pCO_2$ and $A_T$, non-$CO_2$ acid-base constituents, physical variables, and equilibrium constants, this method allows solving $pH$, DIC, $\Omega_{ar}$, and $\Omega_{ca}$ at the same input resolutions. A complementary of the CO2SYS software developed by Orr et al. (2018) is used to quantify the uncertainty associated with these carbonate system variables. All the input data uncertainties are propagated through the CO2SYS processing chain.

The global monthly, $0.25°$-resolution datasets of $pCO_2$, $A_T$, $pH$, DIC, $\Omega_{ar}$, and $\Omega_{ca}$ are intensively evaluated against different observation-based products independent from our model fitting at a global scale to in situ locations (Table 3). In Section 4, multiple metrics are proposed for product analyzes and assessments. Results are presented in section 5 with emphasis on the evaluation of the best reconstruction and associated model uncertainty for each variable (Sect. 5). This section also highlights the advantages obtained with an increase in spatial resolution and presents an application of the CMEMS-LSCE product in
tracking ocean acidification over the last three decades. Section 6 summarizes key results, discusses the potential for future model upgrades, and introduces possible product use cases.

## 2    Data used and reprocessing

### 2.1    Input data products for surface ocean carbonate system reconstructions

Many observation-based products are used as predictors of our target carbonate system variables (Table 2). Global ocean maps
of sea surface temperature (SST), salinity (SSS), height (SSH), chlorophyll-*a* (Chl-*a*) come from the Copernicus Marine Environment Monitoring Service (CMEMS: Good et al., 2020; Nardelli et al., 2016; Droghei et al., 2018; Maritorena et al., 2010). Mixed layer depth (MLD) fields belong to Estimating the Circulation and Climate of the Ocean project Phase II (ECCO2, Menemenlis et al., 2008). $CO_2$ mole fractions ($xCO_2$) are derived from the $CO_2$ atmospheric inversion of the Copernicus Atmosphere Monitoring Service (CAMS, Chevallier et al., 2005, 2010; Chevallier, 2013). Surface ocean concentrations of nitrate
($NO_3$), silicate ($SiO_2$), and phosphate ($PO_4$) are extracted from the World Ocean Atlas 2018 (WOA18, Garcia et al., 2019). The climatological $pCO_2$ ($pCO_2^{clim}$) product is provided by Lamont Doherty Earth Observatory (LDEO, Takahashi et al., 2009). Details of these products including resource access, data coverage, and resolutions are presented in Table 2.

With the exception of $xCO_2$, nutrient concentrations, and $pCO_2^{clim}$, these input data products have original resolutions equivalent to or even finer than a spatial resolution of $0.25°$ and a temporal resolution of monthly. When mismatches in data



**Table 2.** Input data used in the reconstructions of CMEMS-LSCE carbonate system variables over the global ocean in 1985-2021.

| Variables | Notations | Units | Products | Resolutions | References |
|---|---|---|---|---|---|
| 1. $CO_2$ fugacity | $fCO_2$ | µatm | Surface Ocean $CO_2$ Atlas version 2022 (SOCATv2022, 1985-2021) | monthly, $1°$ (open ocean) and $0.25°$ (coastal ocean) | Bakker et al. (2022) |
| 2. Sea surface temperature | SST | °C | CMEMS SST_GLO_SST_L4_REP_OBSERVATIONS_010_011 and SST_GLO_SST_L4_NRT_OBSERVATIONS_010_001 (1985-2021) | daily, $0.05°$ | Good et al. (2020) |
| 3. Sea surface salinity | SSS | PSU | CMEMS MULTIOBS_GLO_PHY_S_SURFACE_MYNRT_015_013 (1993-2021) | monthly, $0.25°$ | Nardelli et al. (2016); Droghei et al. (2018) |
| 4. Sea surface height | SSH | m | CMEMS SEALEVEL_GLO_PHY_L4_MY_008_047 and SEALEVEL_GLO_PHY_L4_NRT_OBSERVATIONS_008_046 (1993-2021) | daily, $0.25°$ | CLS-TOULOUSE |
| 5. Mixed layer depth | MLD | m | Estimating the Circulation and Climate of the Ocean project Phase II (ECCO2, 1992-2021) | daily, $0.25°$ | Menemenlis et al. (2008) |
| 6. Chlorophyll-$a$ | CHL-$a$ | $mg\,m^{-3}$ | CMEMS OCEANCOLOUR_GLO_CHL_L4_REP_OBSERVATIONS_009_082 and OCEANCOLOUR_GLO_CHL_L4_NRT_OBSERVATIONS_009_033 (1998-2021) | daily, $0.25°$ | GLOCOLOUR, Maritorena et al. (2010) |
| 7. $CO_2$ mole fraction | $xCO_2$ | ppm | $CO_2$ atmospheric inversion from the Copernicus Atmosphere Monitoring Service (CAMS, 1985-2021) | 3-hourly, $1.9° \times 3.75°$ | Chevallier et al. (2005, 2010); Chevallier (2013) |
| 8. $pCO_2$ climatology | $pCO_2^{clim}$ | µatm | Lamont Doherty Earth Observatory (LDEO, climatology) | monthly, $4° \times 5°$ | Takahashi et al. (2009) |
| 9. Nitrate 10. Silicate 11. Phosphate | $NO_3$ $SiO_2$ $PO_4$ | $µmol\,kg^{-1}$ | World Ocean Atlas 2018 (WOA18, climatologies) | monthly, $1°$ | Garcia et al. (2019) |

\* Data products 1-8 are used in the $pCO_2$ reconstruction. Products 2-3 and 9-11 are used to compute $A_T$, DIC, $pH$, $\Omega_{ar}$, and $\Omega_{ca}$.

\*\* Last access was on 15/4/2022 for all input databases except for SOCATv2022 data (17/6/2022) and WOA18 data (30/7/2022).

resolutions appear, input data products are interpolated to fit the pre-defined model resolutions. The datasets of SST and $xCO_2$ - the two key variables driving global $pCO_2$ changes (Bates et al., 2014; Gruber et al., 2019; Landschützer et al., 2019; Chau et al., 2022b; Friedlingstein et al., 2022) - cover the full learning period and the whole globe as expected. The other predictor data are not available before the 1990s, when new types of satellite measurements started, and one of them (i.e., Chl-$a$) does not cover the high latitudes of the winter hemisphere. We therefore gap-fill the time series in an ad hoc manner, as in previous

studies (Landschützer et al., 2016; Gregor et al., 2019; Chau et al., 2022b). Monthly climatologies of SSS, Chl-$a$, and MLD computed on the available data are used for each missing year. Likewise, climatologies plus linear trends of SSH following global warming effects serve for the pre-1993 period. Missing Chl-$a$ data in the high latitudes of the winter hemisphere are replaced by the minimum concentration of Chl-$a$ over the available data for the same grid cell ($\sim$0.01 $mg\,m^{-3}$). WOA18 nutrients and LDEO $pCO_2^{clim}$ are already climatolgies per se and we apply them for all the analysis years 1985-2021.

$CO_2$ fugacity from Surface Ocean $CO_2$ ATlas version 2022 (SOCATv2022, Bakker et al., 2022) is used as the target data in our monthly $pCO_2$ reconstructions. The SOCAT project collects and qualifies underway observations via international vessels, moorings, or autonomous platforms. It grids the observations at spatial resolutions of $1°$ or $0.25°$ resulting in the two





major SOCAT gridded data products. The temporal resolution of these two products is monthly. While the $1°$-data product (SOCATv2022r100) covers the global ocean, the $0.25°$ covers solely the coastal regions. The SOCAT coastal areas is within
$400$ km from the shoreline (Sabine et al., 2013; Bakker et al., 2016); see Fig. A1a for an illustration. To merge the two resolutions, we first duplicate the $1°$-open-ocean SOCATv2022 data ($\sim 2 \times 10^5$ data points) over its sixteen $0.25°$ sub-cells. This $0.25°$-open-ocean data are then combined with the $0.25°$-coastal-ocean SOCATv2022 data ($\sim 4 \times 10^5$ data points) to generate a global monthly $0.25°$ ocean data product fed to our reconstruction model of $pCO_2$ (Sect. 3.1). The merged SOCATv2022 product at monthly, $0.25°$ resolutions is referred to as SOCATv2022r025 hereafter. The assumption of open-ocean data homogeneity of $pCO_2$ within $1°$-grid boxes ($\sim 100$ km $\times 100$ km) does not degrade the reconstruction skill over the global open ocean (see Sect. 5 for results) where $pCO_2$ observations are spatially auto-correlated within a global median distance of $400 \pm 250$ km (Jones et al., 2012). The data distribution of SOCATv2022 $CO_2$ fugacity before and after combining is shown in Fig. A2 and Table 4.

## 2.2 Product qualification and comparison

The monthly, $0.25°$-resolution reconstructions of carbonate system variables are qualified with gridded observation-based datasets and in-situ time series which are not used in our model fitting (Table 3).

- The SOCAT data in each reconstruction month are excluded from the model fitting, which avoids overfitting and ensures fairness in the model evaluation (Chau et al., 2022b). The global monthly CMEMS-LSCE-FFNN $pCO_2$ fields at a spatial resolution of $0.25°$ can therefore be evaluated against the $pCO_2$ data converted from SOCATv2022 $CO_2$ fugacity
(Eq. A2) at the same resolution. Doing this, CMEMS-LSCE-FFNN $pCO_2$ is assessed with more than $32 \times 10^5$ open-ocean data and $4 \times 10^5$ coastal-ocean data (Table 4). The SOCATv2022 measurements have low random uncertainty (2-5 $\mu$atm) but the spatio-temporal sampling bias from the month and grid centers is significant (Bakker et al., 2016). The $0.25°$-data reconstruction is also compared to its previous version with a spatial resolution of $1°$ (Chau et al., 2022a, b).

- The monthly, $0.25°$ reconstructions of $A_T$, DIC, and $pH$ are qualified based on Global Ocean Data Analysis Project bottle
data version 2.2022 (GLODAPv2.2022, Lauvset et al., 2022). GLODAP provides non-gridded datasets of ocean carbon variables which have been compiled and bias-corrected from water samples taken at various depths. The measurement uncertainty is 4 $\mu$mol kg$^{-1}$ in $A_T$ and DIC and between $0.01 - 0.02$ in $pH$. Only direct measurements at depths shallower than 10 m and with a flag of 2 (best quality control) are selected for this evaluation. Measurements in each box of $1\text{month} \times 10\text{m} \times 0.25° \times 0.25°$ are averaged to obtain representative data of surface $A_T$, DIC, and $pH$ at $0.25°$-grid
cells for months in the period 1985-2021. This results in roughly $16 \times 10^3$ data points for $A_T$ and DIC over the global ocean (Table 5). Only half of that amount stems from direct $pH$ measurements. Another half, referred to as indirect measurements (i.e., $pH$ calculated with $A_T$ and DIC), is excluded from this data evaluation. Over $30\%$ of these data are distributed along the coasts. The number of the GLODAPv2.2022 gridded data (Table 5) is much less than the SOCATv2022 gridded data (Table 4).



- In situ time series of direct measurements of carbonate system variables ($p$CO$_2$, A$_T$, DIC, and $p$H) are used to qualify our product at local scale (Table 3). Sutton et al. (2019) present data over multiple sites equipped with autonomous moorings measuring surface ocean $p$CO$_2$ and $p$H from the open ocean to the continental shelves since 2004. These time series were used to qualify the CMEMS-LSCE-FFNN reconstruction in Chau et al. (2022b). This study only revisits eight coastal sites (Table A2) where both $p$CO$_2$ and $p$H have been measured and the 1°-reconstruction poorly constrains most of these measurements (see later in Sect. 3.1). The eight stations are located along the US coast, the Gulf of Mexico, and in a Caribbean coral reef (Fig. A1b and Table A2). Measurement uncertainty is up to 2 µatm reported for $p$CO$_2$ and 0.02 for $p$H. For A$_T$ and DIC, we consider four time series: (1) Bermuda Atlantic Time Series (BATS, Michaels and Knap, 1996; Steinberg et al., 2001), (2) Atmospheric Flux Dynamics Time Series in the Mediterranean (DYFAMED, Coppola et al., 2021), (3) European Station for Time-Series in the Ocean Canary islands (ESTOC, González-Dávila and Santana-Casiano, 2009), and (4) Hawaii Ocean Time-series (HOT, Dore et al., 2009). The first three stations are in the North Atlantic while the latter is located in the North Pacific (Fig. A1b). These long-term time series provide insights into changes in the surface ocean carbonate system over the recent decades (Bates et al., 2014; Coppola et al., 2020; Gregor and Gruber, 2021; Pérez et al., 2021). The HOT and ESTOC stations provide surface ocean observations of A$_T$ and DIC, and BATS and DYFAMED data are extracted at seawater depth shallower than 10 m. A monthly average is applied for all the mentioned time series in order to be compatible with output from the CMEMS-LSCE chain of models.

## 3 Reconstruction methods

### 3.1 Ensemble $p$CO$_2$ mapping feed-forward neural networks

The CMEMS-LSCE-FFNN (Chau et al., 2022b) is based on an ensemble of 100 feed-forward neural network (FFNN) models mapping SOCAT CO$_2$ fugacity ($f$CO$_2$) and predictor variables (Eq. 1).

$$f\mathrm{CO}_2 = \mathrm{FFNN}\,(\mathrm{SST}, \mathrm{SSS}, \mathrm{SSH}, \mathrm{Chl}-a, \mathrm{MLD}, x\mathrm{CO}_2, f\mathrm{CO}_2^{clim}, \text{latitude}, \text{longitude}) \tag{1}$$

The predictors of $f$CO$_2$ include sea surface temperature (SST), salinity (SSS), surface height (SSH), chlorophyll-$a$ (Chl-$a$), mixed layer depth (MLD), CO$_2$ mole fraction ($x$CO$_2$), $f$CO$_2$ climatologies ($f$CO$_2^{clim}$), and the geographical coordinates (latitude and longitude). The datasets of SOCAT $f$CO$_2$ and predictors are first reprocessed to match model fitting requirements (Sect. 2.1). After excluding the data in the reconstruction month, the data within the 3-month window are separated into FFNN training and validation subsets with a ratio of $2:1$. The excluded SOCATv2022 datasets are used in model evaluation. The CMEMS-LSCE-FFNN approach was originally developed for $p$CO$_2$ reconstructions at monthly, 1° resolutions where $p$CO$_2$ is converted from $f$CO$_2$ following the formulation (A2) by Körtzinger (1999). The model best estimate and its uncertainty are defined as the ensemble mean ($\mu$) and ensemble spread ($\sigma$) of 100 model outputs of $p$CO$_2$.

This study slightly modifies the CMEMS-LSCE-FFNN ensemble approach by Chau et al. (2022b) to achieve $p$CO$_2$ reconstructions at monthly, 0.25° resolutions. Some of the input datasets presented here (Table 2) are different from those presented



**Table 3.** Data sources used in product evaluation and comparison.

| | Product | Data type | Evaluation variables | Reference |
|---|---|---|---|---|
| Global ocean | 1. Surface Ocean $CO_2$ Atlas version 2022 (SOCATv2022, 1985-2021), last access: 17/6/2022 | observation-based gridding, resolution: $1°$ (global ocean) and $0.25°$ (coastal ocean), monthly | $pCO_2$ | Bakker et al. (2022) |
| | 2. CMEMS global ocean surface carbon product (MULTI-OBS_GLO_BIO_CARBON_SURFACE_REP_015_008, 1985-2021), last access: 05/12/2022 | SOCAT-based reconstruction, resolution: $1°$, monthly | $pCO_2$ | Chau et al. (2022a, b) |
| | 3. Global Ocean Data Analysis Project bottle data version 2.2022 (GLODAPv2.2022, 1985-2021), last access: 30/9/2022 | observation | $A_T$, DIC, $pH$ | Lauvset et al. (2022) |
| Time series stations | 4. Autonomous time series from surface buoys since 2004 (see details in Table A2), last access: 15/10/2022 | observation | $pCO_2$ | Sutton et al. (2019) |
| | 5. Bermuda Atlantic Time Series (BATS, $31.7°$N-$64.2°$W, 1988-2021), last access: 30/10/2022 | observation | $A_T$, DIC | Michaels and Knap (1996); Steinberg et al. (2001) |
| | 6. Atmospheric Flux Dynamics Time Series in the Mediterranean (DY-FAMED, $43.5°$N-$7.9°$E, 1998-2017), last access: 23/03/2023 | observation | $A_T$, DIC | Coppola et al. (2020, 2021) |
| | 7. European Station for Time-Series in the Ocean Canary islands (ES-TOC, $29.2°$N-$15.5°$W, 1995-2009), last access: 30/10/2022 | observation | $A_T$, DIC | González-Dávila and Santana-Casiano (2009) |
| | 8. Hawaii Ocean Time-series (HOT, $22.5°$N-$158.1°$W, 1988-2020), last access: 30/10/2022 | observation | $A_T$, DIC | Dore et al. (2009) |

in Chau et al. (2022b) (Table S1). The up-to-date input datasets have higher resolutions and a better coverage over the coastal ocean as well as in the high latitudes. Furthermore, the new CMEMS data resources offer space-time varying uncertainty fields which are important in quantifying carbonate system variable uncertainties.

For comparable evaluations in this study, we execute 100-member ensembles of FFNN models at spatial resolutions of both $1°$ (FFNNr100) and $0.25°$ (FFNNr025) using the same lot of input data resources (Table 2). Remind that the training data of $f CO_2$ is extracted from the SOCATv2022r100 product for FFNNr100 while it comes from the SOCATv2022r025 product (i.e., the merged product of the $1°$-open-ocean dataset and the $0.25°$-coastal-ocean dataset) for FFNNr025. All input datasets are reprocessed with respect to each model resolution (Sect. 2.1). Sect. 3.1 compares these two CMEMS-LSCE-FFNN versions and highlights the skill of the finer resolution data product.

## 3.2    Locally interpolated alkalinity regression

Locally interpolated alkalinity regression (LIAR; Carter et al., 2016, 2018) is an ensemble-based regression method developed for the global reconstruction of total alkalinity ($A_T$). Regression coefficients were learned on GLODAPv2 data (Olsen et al., 2016) binned within regular windows of $5° \times 5°$. For prediction, the LIAR software interpolates between the regression



coefficients to arbitrary resolutions specified by the users. This study employs eight LIAR models (Carter et al., 2018, Table 2) for calculating $A_T$ at monthly, $0.25°$ resolutions. Each model represents a combination of predictor variables (see the full presentation in Eq. 2),

$$A_T = \text{LIAR}\,(SSS, SST, NO_3, SiO_2). \tag{2}$$

The eight regression models include salinity (SSS) - the predominant predictor of $A_T$ - and some combinations of temperature (SST), nitrate ($NO_3$), and silicate ($SiO_2$). The model which has the smallest propagation uncertainty is chosen to provide the best estimate of $A_T$.

Global monthly total alkalinity and $1\sigma$-uncertainty are estimated with given input data from the monthly CMEMS SSS and SST fields and from the WOA18 datasets of nutrient concentrations (Table 2). Uncertainty of the $A_T$ field is estimated systematically through input uncertainty propagation along the processing chain (Carter et al., 2018). Here we define the input uncertainty of predictors in terms of standard deviations ($1\sigma$). Input uncertainty fields associated to the monthly CMEMS SSS and SST are products' analysis errors (see e.g., Fig. A8) while uncertainties of the WOA18 $NO_3$ and $SiO_2$ climatologies are set to $15\%$ of data values per cell. The $15\%$ quantity refers to the median percentage of standard analysis errors against climatological means of nutrient concentrations (see product standard errors in Table 6, Garcia et al., 2019). The WOA18 standard analysis errors are defined as misfits between their interpolated data and GLODAPv2 bottle data (Olsen et al., 2016). Spatial distribution of the error percentage of the WOA18 nutrient concentrations at the ocean surface is illustrated in Fig. A7.

## 3.3 Carbonate system speciation

The CO2SYS speciation software was first developed by Lewis and Wallace (1998) to determine carbonate system parameters in the marine $CO_2$ system based on a set of equilibrium equations (Dickson et al., 2007). Here we use the speciation program written by Van Heuven et al. (2011) and its extension with uncertainty propagation proposed by Orr et al. (2018). To obtain a complete description of the ocean carbonate system, the CO2SYS initialization requires the following input conditions:

i) values of any couple of the parameters $pCO_2$, $A_T$, DIC, and $pH$,

ii) temperature and pressure,

iii) total concentrations of all the non-$CO_2$ acid-base systems,

iv) equilibrium constants used to describe seawater acid-base chemistry.

The (iii)-condition involves total concentrations of both conservative and non-conservative constituents in the non-$CO_2$ acid-base systems. The amount of conservative constituents such as borate, fluoride, and sulfate in surface seawater is estimated with salinity. The total concentration of non-conservative constituents (nutrients) is computed approximately with silicate ($SiO_2$), and phosphate ($PO_4$). Further information of the carbonate system speciation can be found in Dickson et al. (2007) and Dickson (2010).



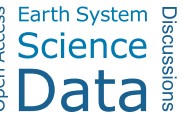

With the reconstructions of $pCO_2$ and $A_T$ (Sects 3.1 and 3.2), the CO2SYS speciation software is used to derive $pH$, DIC,
$\Omega_{ar}$, and $\Omega_{ca}$, and determine their uncertainty over the ocean surface at a resolution of $0.25°$. Equation 3 expresses all input-output variables of CO2SYS for this study. Note that the estimates for other carbonate system variables such as hydrogen ion $(H^+)$ concentration and Revelle Factor (RF) - a measure of the carbonate buffer capacity- are also available (Figs. A4 and A6) but beyond the scope of our data evaluation.

$$pH,\ \text{DIC},\ \Omega_{ar},\ \Omega_{ca} = \text{CO2SYS}\ (pCO_2, A_T, \text{SST}, \text{SSS}, P, SiO_2, PO_4, \text{constants}) \tag{3}$$

The FFNN best estimate (ensemble mean) of $pCO_2$ reconstructions (Sect. 3.1) and the LIAR outputs of $A_T$ (Sect. 3.2) are used as the prior inputs of the CO2SYS at each grid cell for every month in the period 1985-2021. We take the same data products of SST, SSS, and nutrient concentrations as for the previous reconstructions (Table 2). Pressure (P) is assumed to be $0\ \text{dbar}$ at the ocean surface. For equilibrium constants, we choose the best empirical values recommended by Dickson et al. (2007) and Dickson (2010). These settings include (1) the dissociation constants $K_1$ and $K_2$ from Lueker et al. (2000) and $K_{\text{HSO}_4}$ from
Dickson (1990) in combination with the total boron-ratio-salinity formulation by Uppstrom (1974).

The uncertainty of the CO2SYS variables is estimated by error propagation (Orr et al., 2018). Inputs for the CO2SYS error propagation include the reconstruction uncertainty of $pCO_2$ (FFNN ensemble standard deviation) and of $A_T$ (LIAR error propagation). The uncertainty of SST, SSS, and nutrient concentrations are set to the same values as in the previous section (Sect. 3.2). Equilibrium constants' standard errors are default values (see Table 1, Orr et al., 2018). As for FFNN and LIAR,
uncertainty values of each carbonate system variable are computed for each month in 1985-2021 and at each $0.25°$-grid box over the global surface ocean.

## 4 Evaluation metrics

### 4.1 Model best estimate and uncertainty quantification

The 100 FFNN models result in an ensemble of 100 estimates of global monthly, $0.25°$ surface ocean $pCO_2$ fields (Sect. 3.1).
Specify any $t = 1 : 444$ (month), $i = 1 : 180$ (latitude), and $j = 1 : 360$ (longitude), the best estimate ($\mu_{tij}$) and uncertainty ($\sigma_{tij}$) at time $t$ and grid cell $ij$ are deduced from 100 FFNN $pCO_2$ estimates $(X(t,i,j,m))_{m=1}^{m=100}$ as follows.

$$\mu_{tij} = \frac{\sum_{m=1}^{m=100} X(t,i,j,m)}{100}, \tag{4a}$$

$$\sigma_{tij} = \sqrt{\frac{\sum_{m=1}^{m=100} \left[X(t,i,j,m) - \mu_{tij}\right]^2}{100}}. \tag{4b}$$

For $pH$, $A_T$, DIC, $\Omega_{ar}$, and $\Omega_{ca}$, the best estimates and associated uncertainties ($\mu_{tij}$ and $\sigma_{tij}$) are obtained directly from the
290 LIAR and CO2SYS speciation tools and their error propagation (Sects. 3.2 and 3.3).

To assign representatives of $\mu$ and $\sigma$ estimates for carbonate system variables at a specific space-time window, we define statistics with respect to each of the three following cases:





i) a representative over a period of time ($T$ months)

$$\mu_{ij} = \frac{\sum_t \mu_{tij}}{T}, \tag{5a}$$

$$\sigma_{ij} = \sqrt{\frac{\sum_t \sigma_{tij}^2}{T}}. \tag{5b}$$

ii) a representative over a region (e.g., ocean basins and sub-basins, the global ocean)

$$\mu_t = \frac{\sum_{ij} \mu_{tij} \times A_{ij}}{\sum_{ij} A_{ij}}, \tag{6a}$$

$$\sigma_t = \sqrt{\frac{\sum_{ij} \sigma_{tij}^2 \times A_{ij}}{\sum_{ij} A_{ij}}}. \tag{6b}$$

iii) a representative over a period of time and a region

$$\mu = \frac{\sum_{t,ij} \mu_{tij} \times A_{ij}}{T \times \sum_{ij} A_{ij}}, \tag{7a}$$

$$\sigma = \sqrt{\frac{\sum_{t,ij} \sigma_{tij}^2 \times A_{ij}}{T \times \sum_{ij} A_{ij}}}. \tag{7b}$$

where $t$ is in a time period with length $T$ and $A_{ij}$ is the area of each grid cell in a desired region. It is noteworthy that the statistics in Eqs. (5b)-(7b) are not the standard deviation associated to the mean quantities in Eqs. (5a)-(7a), but they stand for the best representative of uncertainty estimates over an ocean basin and/or time period. These statistics also support for the comparison with model-observation deviation (e.g., Eq. 10) which is typically used in the calculation of standard uncertainty proposed in the previous studies (Jiang et al., 2019; Iida et al., 2021; Gregor and Gruber, 2021). Subscripts in the notations of the best model estimates ($\mu$) and model uncertainties ($\sigma$) in Eqs. (4)-(7) are dropped out for general situations.

The best model estimate ($\mu$) is assessed against model uncertainty ($\sigma$) through $\sigma$-to-$\mu$ ratio (%)

$$R(\mu, \sigma) = 100\% \, \frac{\sigma}{|\mu|}. \tag{8}$$

The $\sigma$-to-$\mu$ ratio allows evaluating the significance of the model estimate. Model estimates of carbonate variables are reliable with $R(\sigma, \mu)$ values less than $20\%$ (Rose, 2013).

## 4.2 Model performance in comparison with evaluation data

Assume that $\mu_{tij}$ and $O_{tij}$ are the best model estimate and an observation (or its gridded data) available at time $t$ and grid cell $ij$, and $\mu$ and $O$ are respectively their means over the total number of evaluation data ($N$). Model skills are assessed against observation data (Table 3) with the following metrics:

- mean model-observation differences (Bias)

$$\text{Bias} = \frac{\sum_{t,ij} (\mu_{tij} - O_{tij})}{N}, \tag{9}$$

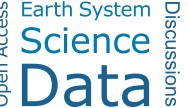

- root-of-mean-square-deviation (RMSD)

$$\text{RMSD} = \sqrt{\frac{\sum_{t,ij}\left(\mu_{tij} - O_{tij}\right)^2}{N}}, \tag{10}$$

• coefficient of determination $(r^2)$

$$r^2 = \frac{\left[\sum_{t,ij}\left(\mu_{tij} - \mu\right) \times \left(O_{tij} - O\right)\right]^2}{\sum_{t,ij}\left(\mu_{tij} - \mu\right)^2 \times \sum_{t,ij}\left(O_{tij} - O\right)^2}. \tag{11}$$

## 5   Results

### 5.1   Surface ocean $p\mathrm{CO_2}$

This section presents the reconstruction of surface ocean $p\mathrm{CO_2}$ at monthly and $0.25°$ resolutions. The reconstruction skill is
evaluated against SOCATv2022 data from global to ocean basin scale and at the level of grid cells (Sect. 2.2). We compare the
novel reconstruction at a higher spatial resolution to the one at a coarser spatial resolution (Chau et al., 2022b). Emphasis is put
on the skill to reproduce spatial and temporal variations of $p\mathrm{CO_2}$ across a variety of coastal regions and time series stations.

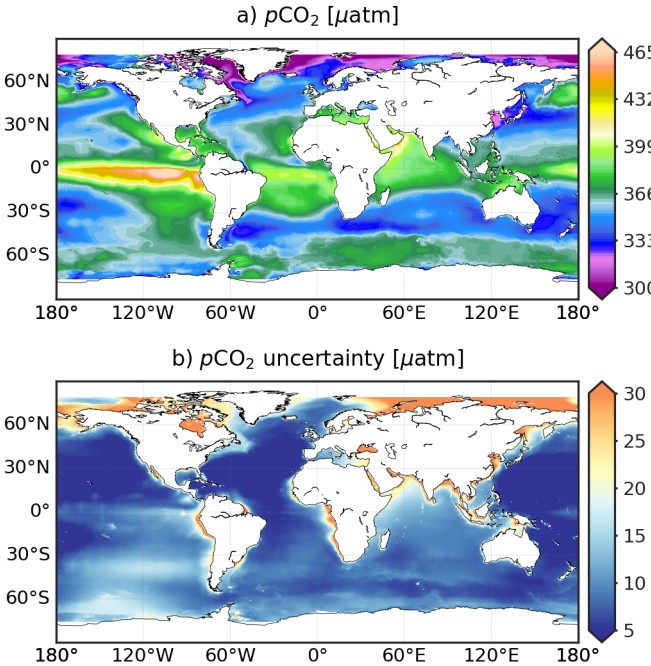

**Figure 1.** CMEMS-LSCE-FFNN $p\mathrm{CO_2}$ over the global ocean at a spatial resolution of $0.25°$. Temporal means of the model best estimate
and $1\sigma$-uncertainty per grid cell over 1985-2021 are calculated by using Eq. (5).





**Table 4.** Skill scores for monthly CMEMS-LSCE-FFNN reconstructions of $pCO_2$ at $1°$ (r100) and $0.25°$ (r025) spatial resolutions computed over the period 1985-2021. r100 $\rightarrow$ 025 and r025 $\rightarrow$ 100 are referred to the versions upscaled or downscaled from the original CMEMS-LSCE-FFNN $pCO_2$ at $1°$ and $0.25°$ resolutions. SOCATv2022 gridded data independent from CMEMS-LSCE-FFNN training are used as benchmarks for model evaluation (see text for details). Statistics including total numbers of data, RMSD (Eq. 10), and $r^2$ (Eq. 11) are reported for both the open ocean (O) and coastal region (C). $*$ marks results with respect to the primary product proposed in this study.

| | Basins | | Number of data | | RMSD [µatm] | | | | $r^2$ | | | |
|---|---|---|---|---|---|---|---|---|---|---|---|---|
| | | | r100 | r025* | r100 | r025 $\rightarrow$ 100 | r025* | r100 $\rightarrow$ 025 | r100 | r025 $\rightarrow$ 100 | r025* | r100 $\rightarrow$ 025 |
| 0. | Globe | (O) | 207174 | 3317273 | 14.32 | 14.08 | 14.29 | 14.38 | 0.83 | 0.83 | 0.83 | 0.83 |
| | | (C) | 101007 | 431758 | 26.61 | 26.48 | 27.55 | 28.50 | 0.72 | 0.72 | 0.74 | 0.72 |
| 1. | Arctic | (O) | 537 | 8589 | 27.93 | 27.43 | 28.04 | 28.06 | 0.69 | 0.69 | 0.67 | 0.67 |
| | | (C) | 5897 | 25844 | 38.74 | 38.56 | 41.46 | 43.17 | 0.55 | 0.56 | 0.55 | 0.52 |
| 2. | Atlantic | (O) | 54797 | 876116 | 13.76 | 13.57 | 13.69 | 13.78 | 0.81 | 0.81 | 0.81 | 0.81 |
| | | (C) | 49770 | 227665 | 24.99 | 24.78 | 25.17 | 26.05 | 0.76 | 0.76 | 0.77 | 0.77 |
| 3. | Pacific | (O) | 120604 | 1932981 | 14.59 | 14.30 | 14.54 | 14.67 | 0.85 | 0.85 | 0.85 | 0.85 |
| | | (C) | 26847 | 104269 | 26.79 | 26.90 | 28.46 | 28.95 | 0.71 | 0.71 | 0.69 | 0.67 |
| 4. | Indian Ocean | (O) | 4485 | 71719 | 10.34 | 10.17 | 10.26 | 10.34 | 0.88 | 0.88 | 0.88 | 0.88 |
| | | (C) | 1522 | 6187 | 23.50 | 22.82 | 25.40 | 26.51 | 0.69 | 0.71 | 0.69 | 0.69 |
| 5. | Southern Ocean | (O) | 26751 | 427868 | 14.42 | 14.29 | 14.52 | 14.43 | 0.69 | 0.69 | 0.69 | 0.69 |
| | | (C) | 16971 | 67793 | 26.01 | 25.80 | 27.35 | 28.80 | 0.61 | 0.61 | 0.64 | 0.59 |

Figure 1 presents global maps at $0.25°$-resolution of long-term averages of $pCO_2$ and corresponding uncertainty estimates. Reconstructed $pCO_2$ distributions reveal well documented large scale structures. Values are high over upwelling regions (e.g.,
Equatorial Pacific, California Boundary Current, Western Arabian Sea). Low $pCO_2$ is associated with increased $CO_2$ solubility in cold high latitudes seawater (e.g., Arctic), strong biological production (e.g., China Sea), or the combination of both (e.g., subpolar Northern Atlantic, Southern Ocean between $35-50°$S). Spatial structures appear coherent from small to large spatial scales, both along the coast and moving towards the open ocean (see also in Figs. 2-4). The combination of a down-scaled version of open-ocean and higher-resolution coastal SOCATv2022 data (Sect. 2.1) yields $pCO_2$ distributions without disconti-
nuities. The uncertainty map (Fig 1b) represents the confidence level in surface ocean $pCO_2$ estimates (Fig 1a). Predominantly low uncertainty estimates ($\sigma < 5$ µatm) indicate the global stability of the ensemble reconstruction. Exceptions are found in many coastal regions, open-ocean areas with sparse data coverage (e.g., Southern Pacific, Indian Ocean), or regions with substantially high or low surface ocean $pCO_2$ (e.g., Arctic, eastern equatorial Pacific). However, $pCO_2$ is reconstructed with a high degree of confidence over most of the global ocean with a $\sigma$-to-$\mu$ ratio (Eq. 8) below $10\%$ (Fig. A9a).
Skill scores of the monthly, $0.25°$-resolution reconstruction are presented in Table 4 (columns marked by an asterisk). The global RMSD (Eq. 10) between the best reconstruction and SOCATv2022r025 $pCO_2$ over the entire period is $14.29$ µatm for the open ocean and $27.55$ µatm for the coastal ocean. These two model errors are lower than $4\%$ and $8\%$ of the global mean $pCO_2$ (Table 7). Moreover, variability present in observation-based data is reproduced by the CMEMS-LSCE-FFNN with high values of $r^2$ (open ocean: 0.83, coast: 0.74). The reconstruction quality is similar among major ocean basins. Spatial
distributions of SOCATv2022 data, bias, and RMSD are shown in Figs. A2-bd and A3-bdfh. Estimation skills are low in the





ocean basins with sparse data coverage and significant space-time variability of $pCO_2$ (e.g., Arctic, eastern Equatorial Pacific, land-ocean continuum).

Table 4 also presents statistics for the monthly FFNN products of surface ocean $pCO_2$ at spatial resolutions of $0.25°$ (r025) and $1°$ (r100) together with their variants (r100 → 025 and r025 → 100). The latter are respectively extrapolation and interpo-

lation versions of the original r100 and r025 datasets, i.e., FFNN model outputs regridded to a finer or coarser spatial resolution. For compatibility, we compare statistics between:

i) FFNN(r025) and FFNN(r100 → 025) by using SOCATv2022r025 as evaluation data,

ii) FFNN(r025 → 100) and FFNN(r100) by using SOCATv2022r100 as evaluation data.

The FFNN(r025) central to this study yields a lower RMSD and a higher correlation to the SOCAT data than the FFNN(r100 →

025). As expected, the improvement in reconstruction skill with higher model resolution is larger over coastal regions than in the open ocean. The FFNN(r025) product after interpolating to a coarser resolution, i.e., FFNN(r025 → 100), agrees with the original $1°$-resolution data product over all the ocean.

The motivation to increase the spatial resolution of the reconstruction is to improve the representation of horizontal gradients of $pCO_2$ at fine scales. Figures 2-4 exemplify spatial distributions for the two reconstructions (r025 and r100) over the coastal-

open-ocean continuum. Ten distinct oceanic regions are considered (see Fig. A1a and Table A1 for the ten locations), which can be classified into three groups:

- permanent Eastern Boundary current upwelling systems with relatively high $pCO_2$ (California Current System - CCS, Humboldt Current System - HCS, Canary Current System - CnCS, and Benguela Current System - BCS),

- regions characterized by low $pCO_2$ values driven by cold water temperatures and strong biological production (Labrador

Sea, Western South Atlantic, Northern Europe, and Sea of Japan),

- other regions either under the influence of strong river runoff (Amazon mount) or monsoon-driven upwelling (Western Arabian Sea).

The legend of Figs 2-4 includes regional RMSD and $r^2$ computed between the best estimates of two models and coastal-ocean SOCATv2022r025 data. The coarser spatial resolution product is co-located at the same $0.25°$-grid cells for this analysis. These

figures illustrate important discrepancies in $pCO_2$ data density between coastal regions with poorly monitored regions (e.g., HCS, BCS, Amazon mount) contrasting with areas with higher data coverage (e.g., Northern Europe, Sea of Japan).

Over 7 out of the 10 analysed regions the reconstruction at monthly, $0.25°$ resolutions yields RMSDs below $10\%$ of the global mean of coastal-ocean $pCO_2$ estimates (Table 7) and $r^2$ values higher than 0.3; e.g., Northern Europe (RMSD = 33.90 µatm, $r^2 = 0.80$), Sea of Japan (RMSD = 20.84 µatm, $r^2 = 0.70$), and CnCS (RMSD = 30.36 µatm, $r^2 = 0.35$). The CMEMS-

LSCE-FFNN model projections of $pCO_2$ lack skill over the HCS (RMSD = 54.54 µatm, $r^2 = 0.29$), the region under influence of the Amazon river (RMSD = 45.93 µatm, $r^2 = 0.37$), and the Western Arabian Sea (RMSD = 45.31 µatm, $r^2 = 0.47$). In nearshore sectors of these coastal areas, $pCO_2$ estimates are also subject to a substantial amount of uncertainty ($\sigma > 20$ µatm).


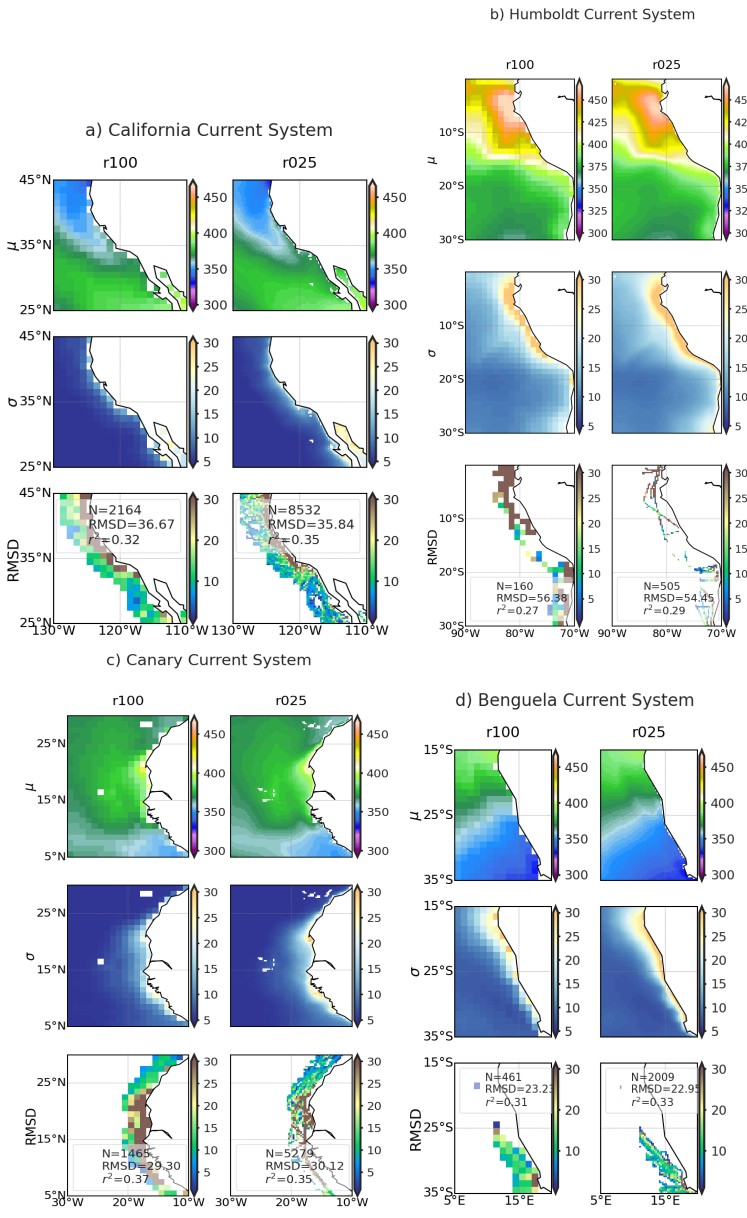

**Figure 2.** Comparison of CMEMS-LSCE-FFNN mapping $pCO_2$ at $1°$ (r100) and $0.25°$ (r025) resolutions over 4 permanent upwelling regions associated with the Eastern Boundary Currents (California, Peru, Canary, and Benguela; see Figure A1-ABGH for geographical locations). For each region, spatial distributions of $pCO_2$ ($\mu$) and uncertainty ($\sigma$) estimates, and coastal-ocean RMSD of $pCO_2$ averaged over 1985-2021 (Eqs. 5 and 10) are shown. Metrics presented in the legend for each of the 3rd row include the number of coastal-ocean SOCATv2022 data ($N$), regional RMSD (Eq. 10) and $r^2$ (Eq. 11).



**Figure 3.** Comparison of CMEMS-LSCE-FFNN mapping $pCO_2$ at $1°$ (r100) and $0.25°$ (r025) resolutions over 4 regions characterized with low $pCO_2$ values (Labrador, South America, Northern Europe, and Japan; see Figure A1-CEFJ for geographical locations). For each region, spatial distributions of $pCO_2$ ($\mu$) and uncertainty ($\sigma$) estimates, and coastal-ocean RMSD of $pCO_2$ averaged over 1985-2021 (Eqs. 5 and 10) are showed. Metrics present in the legend for each of the 3rd row include the number of coastal-ocean SOCATv2022 data ($N$), regional RMSD (Eq. 10) and $r^2$ (Eq. 11).

The lack of model skill reflects the combination of low data density and strong $pCO_2$ gradients driven by multiple underlying physical and biogeochemical processes. The HCS, for instance, is characterized by the highest $pCO_2$ levels (Fig. 2) among the


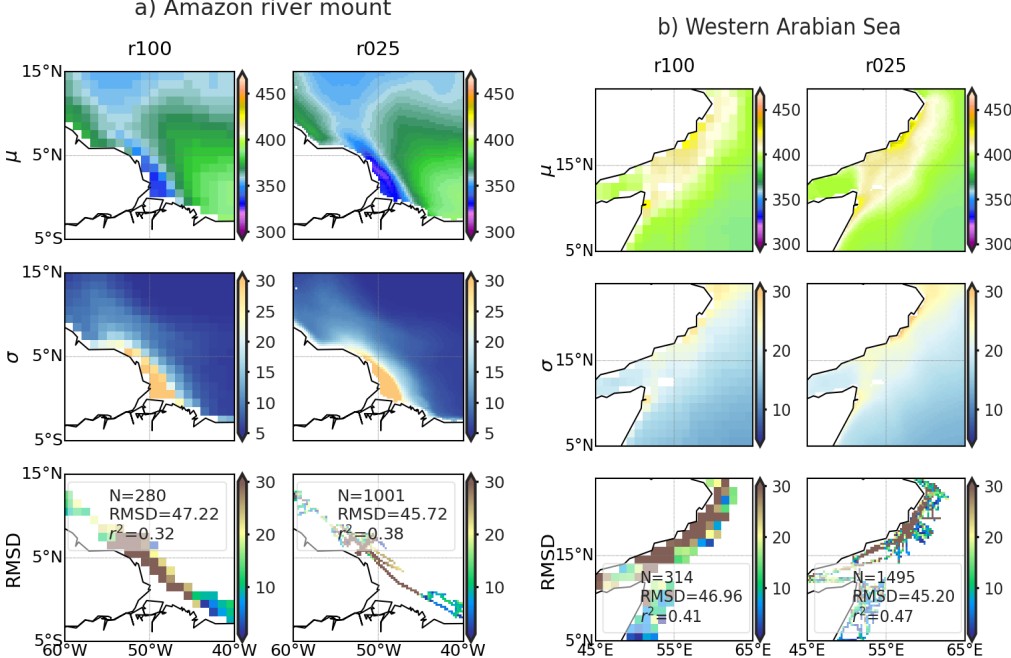

**Figure 4.** Comparison of CMEMS-LSCE-FFNN mapping $p\mathrm{CO}_2$ at $1°$ (r100) and $0.25°$ (r025) resolutions over the mouth of the river Amazon and the Western Arabian Sea (see Fig. A1-DI for geographical locations). For each region, spatial distributions of $p\mathrm{CO}_2$ ($\mu$) and uncertainty ($\sigma$) estimates, and coastal-ocean RMSD of $p\mathrm{CO}_2$ averaged over 1985-2021 (Eqs. 5 and 10) are showed. Metrics present in the legend for each of the 3rd row include the number of coastal-ocean SOCATv2022 data ($N$), regional RMSD (Eq. 10) and $r^2$ (Eq. 11).

four Eastern Boundary Current Systems, with interannual variability amplified with the El Niño–Southern Oscillation (ENSO) events (Feely et al., 1999; Landschützer et al., 2016). Similarly, high $p\mathrm{CO}_2$ levels with substantial seasonal variability are observed over the Western Arabian Sea (Fig. 4b), the key driver being monsoonal upwelling (Sabine et al., 2002; Sarma et al., 2013). In contrast to the two aforementioned coastal regions, high $\mathrm{CO}_2$ undersaturation as well as strong $p\mathrm{CO}_2$ gradients (Fig. 4a) are found in the area under the influence of Amazon river discharge (Olivier et al., 2022). Extreme values and large
variability of $p\mathrm{CO}_2$ challenge any approach to estimate $p\mathrm{CO}_2$ data over these regions (Ibánhez et al., 2015; Bakker et al., 2016; Landschützer et al., 2020).

   The two FFNN reconstructions (r025 and r100) share similarities in overall structures of $p\mathrm{CO}_2$ over the coastal-open-ocean continuum (Figs. 2-4). However, the higher spatial resolution outperforms its lower resolution counterpart is reproducing fine-scale features of $p\mathrm{CO}_2$ in the transition from nearshore regions to the adjacent open ocean. The increase in model spatial
resolution translates into a greater spatial coverage of the continental shelves such as Labrador Sea, Northern Europe, and Sea of Japan (Fig. 3), and thus an increase in the number of data over the coastal domain. The increase in spatial resolution allows a gain in prediction probability of $p\mathrm{CO}_2$ variations on the order of roughly $2\%$ over the Eastern Boundary Currents to $8\%$ over the Western South Atlantic (Figs. 2-3b).

**Figure 5.** Time series of surface ocean $p\text{CO}_2$ at coastal observing stations (Table A2 and Fig. A1b): model best estimate (curve), $1\sigma$-uncertainty (envelope), and monthly average of in situ observations (point). The reconstructed data at $1°$ (r100) and $0.25°$ (r025) resolutions are co-located to in situ observations provided by Sutton et al. (2019). Means of the best estimate and $1\sigma$-uncertainty ($\mu \pm \sigma$) calculated over the observing time span are shown in brackets. Statistics include number of months with observations ($N$), Bias, RMSD, and $r^2$ computed for the two reconstructions. $\sigma^t_{p\text{CO}_2}$ stands for temporal standard deviation from monthly averages of $p\text{CO}_2$ observations.

Reconstruction skill of seasonal to inter-annual variability of $p\text{CO}_2$ is further assessed at eight coastal monitoring sites (Sut-
ton et al., 2019) and illustrated in Fig. 5 (see Sect. 2.2 and Table A2 for data description and Fig. A1b for station locations).
The temporal variability of $p\text{CO}_2$ reported for these time series sites reflects a combination of processes (Sutton et al., 2019),
e.g., California Current System (CAPEARAGO and CCE2), western coastal upwelling (CAPEELIZABETH), eutrophica-



tion enhancing respiration of $CO_2$ (FIRSTLANDING), and multiple stressors on coral reef environments (CHEECAROCKS, GREYREEF). Results from reconstructions at two spatial resolutions are compared: $1°$ (100, black curve) and $0.25°$ (r025,

color curve). As shown in Fig. 5 (scattered points for observations) time series of coastal $pCO_2$ are still short. The longest time series covers 127 months of $pCO_2$ monitoring since 2010 (CCE2) while the shortest one contributes 17 months with observations (FIRSTLANDING).

Analyzing the eight station time series, we have found that data have been sampled within a few days with an average offset of about a week from the month center. At these coastal sites, the temporal standard deviation from monthly averages of $pCO_2$

($\sigma^t_{pCO_2}$) exceeds measurement errors (2 µatm, Sutton et al., 2019). $\sigma^t_{pCO_2}$ ranges from 20.12 µatm at GREYREFF to values as large as 65.6 µatm at CAPEARAGO or 69.98 µatm at FIRSTLANDING. The monthly average of $pCO_2$ might not be adequately represented by discreet samples at sites with a large temporal standard deviation of $pCO_2$. The misfit between the monthly reconstruction and discreet observations is exacerbated in dynamical coastal environments and might explain in part the large RMSD of reconstructions of monthly coastal $pCO_2$ (e.g., GREYREEF: 38.34 µatm, CAPEARAGO: 79.86 µatm,

FIRSTLANDING: 77.32 µatm) for the r025 reconstruction. The RMSD is mostly lower for the FFNN reconstruction at $0.25°$ resolution compared to the FFNN at $1°$ resolution by 2.11 µatm (CCE2) to 23.32 µatm (COASTALMS). Similarly, $r^2$ increases between 7%-23% at higher resolution. Overall, seasonal to interannual variations of coastal-ocean $pCO_2$ are better reproduced in the reconstruction at $0.25°$ resolution (Fig. 5).

### 5.2 Total alkalinity and dissolved inorganic carbon

This section presents and analyzes global ocean surface reconstructions of total alkalinity ($A_T$) and dissolved inorganic carbon (DIC) at monthly, $0.25°$ resolutions over 1985-2021. GLODAPv2.2022 bottle data (Sect. 2.2) serve as reference data for model evaluation. Model reconstruction skill is further assessed at the four Eulerian time series sites: BATS, DYFAMED, ESTOC, and HOTS (Table 3).

Figure 6 shows spatial distributions of the climatological mean and uncertainty (Eq. 5) for $A_T$ and DIC. Despite being in part

influenced by common biological and physical processes, both properties have contrasting distributions due to the strong correlation between surface ocean $A_T$ and salinity (Lee et al., 2006; Broullón et al., 2019), as well as the contribution of air-sea gas exchange and biological productivity on surface ocean DIC levels (Feely et al., 2001; Takahashi et al., 2014). Over subtropical Atlantic gyres and the Mediterranean Sea, oceanic areas with net evaporation, $A_T$ exceeds 2400 µmol kg$^{-1}$. Total alkalinity falls below 2150 µmol kg$^{-1}$ in regions where precipitation, river freshwater runoff, or seasonal sea-ice melting dilute surface

water salinity (e.g., subpolar North Pacific, Arctic, and equatorial river outflows). The distribution of DIC is relatively uniform between the Atlantic, Pacific, and Indian Ocean basins, but shows pronounced latitudinal gradients. High concentrations of DIC are found throughout the Southern Ocean (DIC > 2100 µmol kg$^{-1}$) where strong upwelling brings up subsurface water enriched in $CO_2$ and nutrients. The inefficient utilization of nutrients in this high nutrient low chlorophyll region limits the biological drawdown of DIC allowing the massive DIC input to be spread horizontally by westerlies (Key et al., 2004; Men-

viel et al., 2018). Levels of DIC below 1900 µmol kg$^{-1}$ are reconstructed over the Equatorial Pacific, the Equatorial Eastern Atlantic, the Eastern Indian Ocean, and coastal areas on the Arctic Ocean. While low DIC levels associated with Equatorial up-

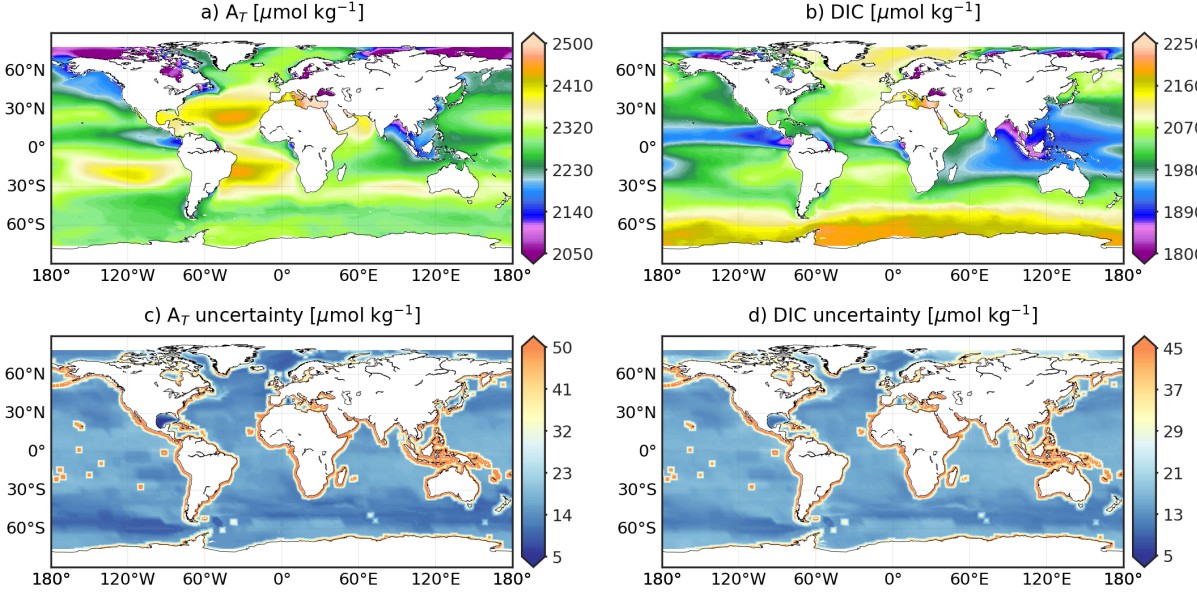

**Figure 6.** CMEMS-LSCE $A_T$ and DIC over the global ocean at a spatial resolution of $0.25°$. Temporal means of the model best estimate and $1\sigma$-uncertainty per grid cell over 1985-2021 are calculated by using Eq. (5).

welling reflect gas exchanges across the air-sea interface and enhanced biological production, the interaction between physical and biogeochemical processes at work in the Indian Ocean are less well understood (Takahashi et al., 2014). Low DIC levels found close to river mouths reflect outgassing of $CO_2$ across the salinity gradient, as well as enhanced biological uptake fueled by river nutrient inputs. Representation uncertainty (Fig. 6-cd) associated with monthly alkalinity and DIC reconstructions is lower than $20\,\mu\mathrm{mol\,kg^{-1}}$ throughout the open ocean. The open-ocean $\sigma$-to-$\mu$ ratio (Eq. 8) ranges between $0.5 - 1.5\%$ which is relatively small (Fig. A9-cd). The largest model uncertainty ($\sigma > 30\,\mu\mathrm{mol\,kg^{-1}}$) is computed nearshore and surrounding oceanic islands, a feature inherited from input uncertainty associated with the CMEMS salinity product (Fig. A8a).

We qualify monthly, $0.25°$ reconstructions of $A_T$ and DIC with measurements from GLODAPv2.2022 (Lauvset et al., 2022) for the 37-year period (Table 5 and Fig. 8). The global open-ocean reconstruction scores a RMSD of $22.09\,\mu\mathrm{mol\,kg^{-1}}$ and a $r^2$ of 0.9 in $A_T$. Similar numbers are found for DIC (RMSD$= 22.67\,\mu\mathrm{mol\,kg^{-1}}$ and $r^2 = 0.9$). The model scores the good fit in the open Indian Ocean with RMSD smaller than $15.5\,\mu\mathrm{mol\,kg^{-1}}$ and $r^2$ above $0.92$ for both variables. The reconstruction deviates from GLODAP data in the western North Atlantic, subpolar North Pacific, tropics, and nearby major rivers (Fig. 8-abcd).

$A_T$ and DIC are underestimated in the continental shelves of north Alaska and the northeastern Atlantic, the Mediterranean Sea, South China Sea, and nearby river plumes (Fig. 8-ac). The Arctic yields the poorest estimations among all the ocean basins with a global RMSD over $100\,\mu\mathrm{mol\,kg^{-1}}$ (Table 5). The prediction probability of variability in $A_T$ [DIC] is relatively large for the open ocean $79\%$ [$71\%$], but rather unsatisfying over the coastal ocean ($46\%$ [$40\%$]). Extrapolating these carbonate variables towards the shore remains challenging with much higher errors and uncertainty estimates obtained over the continental shelf

Earth System
Science
Data

Open Access / Discussions



**Figure 7.** Monthly time series of $A_T$ and DIC at BATS, DYFAMED, ESTOC and HOT stations (Table 3 and Fig. A1b): model best estimate (curve), $1\sigma$-uncertainty (envelope), and monthly average of surface (0-10 m) observations (point). Means of the best estimate and $1\sigma$-uncertainty ($\mu \pm \sigma$) calculated over the observing time span are shown in brackets if accessible. Statistics include number of months with observations ($N$), Bias, RMSD, and $r^2$. $\sigma_{A_T}^t$ [$\sigma_{DIC}^t$] stands for temporal standard deviation from monthly averages of $A_T$ [DIC] observations.

compared to the open-ocean reconstruction (Table 5, Figs. 6-cd and . 8-abcd). The coastal-ocean errors are on the order of $10\%$
of the global mean values of $A_T$ and DIC (Table 7).

The reconstruction of $A_T$ distributions relies on LIAR coefficients fit with GLODAPv2 data (Olsen et al., 2016) covering the years before 2015. These data are also part of the latest version GLODAPv2.2022 (Lauvset et al., 2022). They do therefore not correspond to an independent dataset for the evaluation data of the CMEMS-LSCE reconstruction. To overcome this limitation, reconstructions of $A_T$ and DIC are compared to observations for Eulerian time series stations: BATS, DYFAMED, ESTOC, and
HOT (see Table 3 and Fig. A1b for data sources and station locations). Figure 7 illustrates the comparison between monthly time series of $A_T$ and DIC extracted from the CMEMS-LSCE datasets and measurements at these long-term monitoring sites. The four stations stand out as sustained long-term observation time series for carbonate system variables. More than 270 [80]

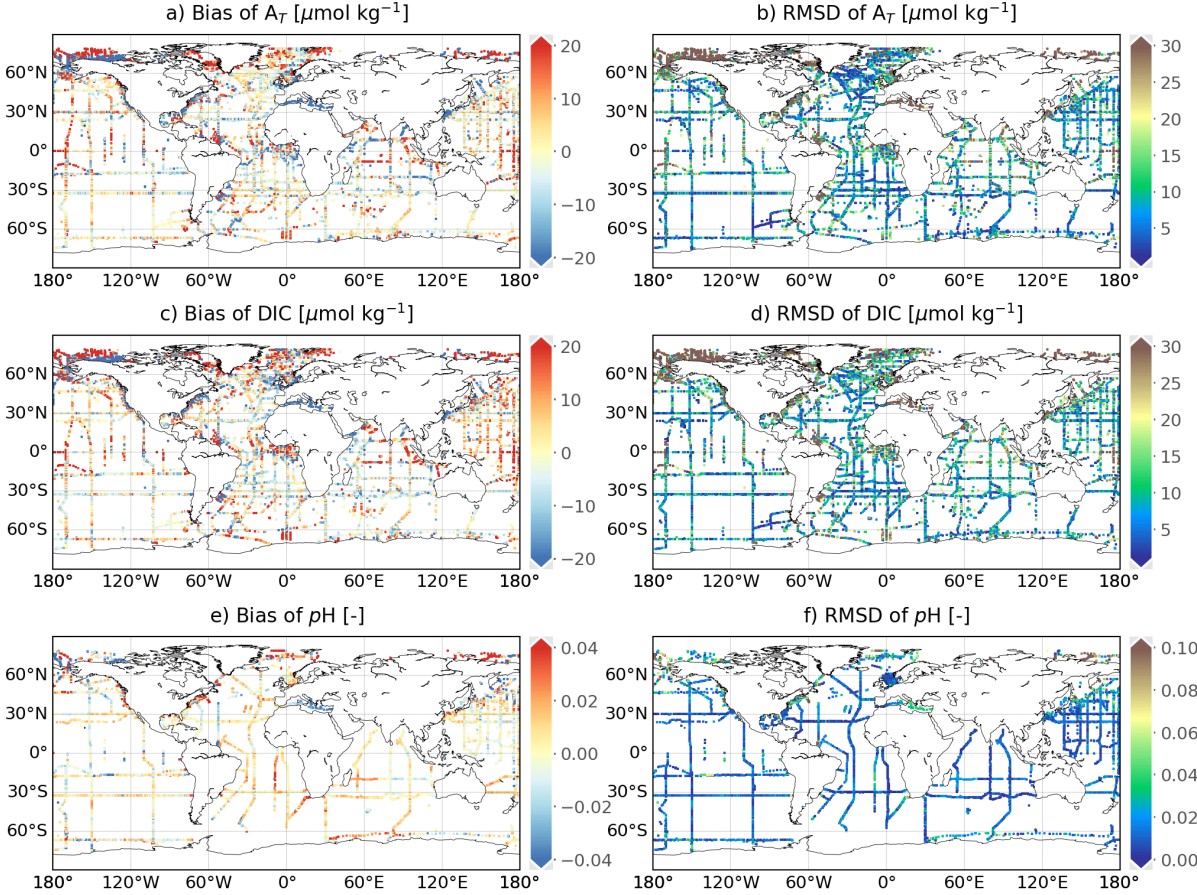

**Figure 8.** Spatial distribution of reconstruction skills for $A_T$, DIC, and $pH$ over 1985-2021. Mean model-data difference (Bias) and root-of-mean square-deviation (RMSD) between the reconstruction and GLODAPv2.2022 surface data (0-10 m) at a spatial resolution of $0.25°$. The size of grid cells is scaled upon a better visualization.

months in the years 1988-2021 [1995-2009 and 1998-2017] include measurements of $A_T$ and DIC at BATS and HOT [ESTOC
and DYFAMED]. The reconstructed time series fit monthly averages of in situ measurements well. Mean estimates of $A_T$ [DIC]

over the observing period are about 2305 [1983] $\mu mol\,kg^{-1}$ at HOT to 2420 [2129] $\mu mol\,kg^{-1}$ at DYFAMED. At all the
stations (DYFAMED excepted) and for the two variables, model-observation misfit is small (Bias $< 10\,\mu mol\,kg^{-1}$, RMSD $<$
13 $\mu mol\,kg^{-1}$) relative to the aforementioned mean estimates. The highest offset between the CMEMS-LSCE estimation and
observations for all the stations is found at DYFAMED ($A_T$: $-145.1\,\mu mol\,kg^{-1}$, DIC: $-124.69\,\mu mol\,kg^{-1}$). DYFAMED
provides long-term time series of $A_T$ and DIC measurements in the Northwestern Mediterranean Sea (Fig. A1b). Salinity and

alkalinity have substantial values due to the net evaporation (Coppola et al., 2020). The average of $A_T$ in the Mediterranean Sea
exceeds that for the global ocean by $10\%$ (Palmiéri et al., 2015). These characteristics set the Mediterranean Sea aside from
the ocean basins. LIAR (Carter et al., 2018) was trained on GLODAPv2 (Olsen et al., 2016) including a few observations in



**Table 5.** Skill scores computed between CMEMS-LSCE and GLODAPv2.v2022 in $A_T$, DIC, and $pH$ over the period 1985-2021. Total numbers of data, RMSD (Eq. 10), and $r^2$ (Eq. 11) are reported for both the open ocean (O) and coastal region (C).

| | Basins | | Number of data | | $A_T$ $[\mu mol\,kg^{-1}]$ | | DIC $[\mu mol\,kg^{-1}]$ | | pH [-] | |
|---|---|---|---|---|---|---|---|---|---|---|
| | | | $A_T$-DIC | $(pH)$ | RMSD | $r^2$ | RMSD | $r^2$ | RMSD | $r^2$ |
| 0. | Globe | (O) | 10269 | (5411) | 22.09 | 0.90 | 22.67 | 0.90 | 0.022 | 0.70 |
| | | (C) | 6309 | (2080) | 82.01 | 0.72 | 72.39 | 0.62 | 0.060 | 0.45 |
| 1. | Arctic | (O) | 103 | (26) | 107.09 | 0.79 | 113.28 | 0.71 | 0.106 | 0.32 |
| | | (C) | 1635 | (300) | 148.71 | 0.46 | 126.77 | 0.4 | 0.107 | 0.48 |
| 2. | Atlantic | (O) | 2785 | (932) | 30.10 | 0.74 | 28.66 | 0.72 | 0.028 | 0.58 |
| | | (C) | 2422 | (941) | 44.50 | 0.71 | 39.09 | 0.69 | 0.046 | 0.45 |
| 3. | Pacific | (O) | 4539 | (3222) | 13.61 | 0.92 | 15.95 | 0.92 | 0.019 | 0.74 |
| | | (C) | 1380 | (639) | 28.43 | 0.76 | 44.36 | 0.45 | 0.057 | 0.34 |
| 4. | Indian Ocean | (O) | 1177 | (551) | 15.05 | 0.92 | 13.79 | 0.96 | 0.012 | 0.90 |
| | | (C) | 328 | (62) | 16.56 | 0.92 | 21.97 | 0.90 | 0.013 | 0.82 |
| 5. | Southern Ocean | (O) | 1665 | (680) | 10.96 | 0.64 | 13.21 | 0.92 | 0.019 | 0.68 |
| | | (C) | 544 | (138) | 22.53 | 0.50 | 24.48 | 0.77 | 0.023 | 0.65 |

this area. The distinct relationship between alkalinity and salinity prevailing in the Mediterranean Sea is likely not reproduced by LIAR leading to an underestimation of $A_T$ and a systematic bias to DIC at DYFAMED (Fig. 7). ESTOC is located close

to the North Atlantic east coast and under the influence of the Canary Current System (CCS, Fig. A1). Spatial gradients and temporal variability are higher in the CCS (Fig. 2c) compared to BATS and HOT which are both located in the center of subtropical gyres. The lowest prediction skill of temporal variability is obtained for ESTOC. Particularly, seasonality to multi-year variations in DIC are predicted at $r^2 = 0.47$ for ESTOC compared to $r^2 > 0.7$ for BATS and HOT. Over all the stations, the model underestimates temporal changes of $A_T$ (Fig. 7a; BATS: $r^2 = 0.33$, DYFAMED: $r^2 = 0.12$, ESTOC: $r^2 = 0.03$,

HOT: $r^2 = 0.32$) which can be attributed to the large discrepancy in variability between in situ measurements and the CMEMS time series of salinity (Fig. A10a; BATS: $r^2 = 0.33$, DYFAMED: $r^2 = 0.19$, ESTOC: $r^2 = 0.03$, HOT: $r^2 = 0.35$). Model uncertainty ($1\sigma$-envelop) of monthly $A_T$ and DIC estimates (Fig. 7a) is also inflated somewhat proportional to the CMEMS salinity product uncertainty (Fig. A10a).

### 5.3 Surface ocean $p$H and saturation state with respect to carbonate minerals

Surface ocean $p$H and saturation states with respect to aragonite ($\Omega_{ar}$) and calcite ($\Omega_{ca}$) are critical indicators used to measure ocean acidification. This section first presents an overall evaluation of these variables. We then introduce estimates involved in the monitoring of ocean acidification in 1985-2021 as an essential application of the CMEMS-LSCE surface ocean carbon product.


### 5.3.1 General analysis and evaluation

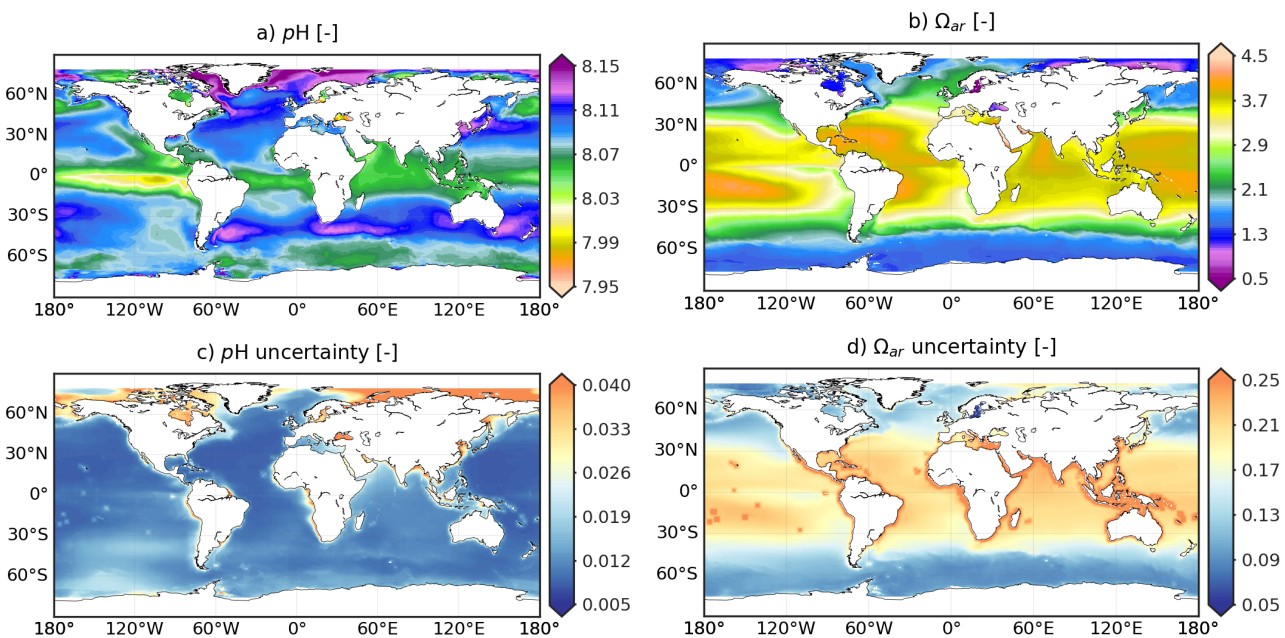

**Figure 9.** CMEMS-LSCE $pH$ and $\Omega_{ar}$ over the global ocean at a spatial resolution of $0.25°$. Temporal means of the model best estimate and $1\sigma$-uncertainty per grid cell over 1985-2021 are calculated by using Eq. (5).

The spatial distribution of surface ocean $pH$ reported on total hydrogen ion ($H^+$) scale is shown in Fig. 9 (the corresponding figure for $H^+$, Fig. A4, is included in the supplementary). Both temporal means of the best model estimate and $1\sigma$-uncertainty of $pH$ share spatial patterns with $pCO_2$ (Fig. 1). Variables $pH$ and $pCO_2$ correlate closely through equilibrium relationships of dissolved $CO_2$ in seawater: an increase in $pCO_2$ generally corresponds to a decrease in $pH$. The distribution of the climatological mean of $pH$ displays a gradient with latitude between $8.03$ and $8.11$ $pH$ units across most of the basins (Fig. 9a). Values

of pH below 8 are associated with the upwelling of $CO_2$-rich waters (e.g., Eastern Equatorial Pacific, Western Arabian Sea). $pH$ exceeds $8.15$ in sub- and polar cold surface water and in the regions with high biological productivity (e.g. Labrador Sea, Nordic Seas, Southern Ocean between $35°S$-$50°S$).

The saturation state of surface ocean waters with respect to calcium carbonate minerals aragonite and calcite is defined as the ratio of the product of the concentrations of calcium ions ($Ca^{2+}$) and carbonate ions ($CO_3^{2-}$) to the solubility of the respective

calcium carbonate mineral ($CaCO_3$) in surface seawater (Eq. A5). Aragonite being the more soluble polymorph, its degree of saturation ($\Omega_{ar}$) is smaller than that of calcite ($\Omega_{ca}$) (Mucci, 1983). With the exception of this offset, the spatial distributions of their climatological means share common spatial patterns over the global ocean (Figs. 9b and A5a). Surface seawater is generally supersaturated, i.e., $\Omega_{ar}$ and $\Omega_{ca}$ greater than 1. The magnitude of surface ocean calcium carbonate saturation state varies with latitude. Values as large as $3.7$-$4.5$ [$5$-$7$] for aragonite [calcite] are reconstructed in subtropical and tropical regions.





$\Omega_{ar}$ and $\Omega_{ca}$ decrease toward the poles. In the Southern Ocean, surface seawater enriched in $CO_2$ from vertical mixing has
$\Omega_{ar}$ [$\Omega_{ca}$] values in the range of 1.5-2.1 [2-3.4]. Low saturation states are also computed in the Arctic and for waters of
upwelling regimes (Fig. 9b). Locally $\Omega_{ar}$ drops below 1.3, and even fall under the $CaCO_3$ dissolution threshold of 1 (Gattuso
and Hansson, 2011) in the Arctic water runoff and Baltic sea.

The uncertainty ($1\sigma$) of $pH$, $\Omega_{ar}$, and $\Omega_{ca}$ propagated the speciation of the $CO_2$ system takes into account the ensemble
spread of $pCO_2$ estimates and analysis errors of other variables (Sect. 3.3). Monthly $pH$ uncertainty estimates fall in the 95%
confidence interval of $[0.008, 0.036]$ with a global mean value of 0.011. These estimates are in close agreement with the global
uncertainty between 0.01-0.022 $pH$ units calculated by Jiang et al. (2019), Iida et al. (2021), and Gregor and Gruber (2021). $pH$
uncertainty is typically larger than 0.03 in the Arctic and in coastal regions (Figs. 9c). In contrast, the reconstructions of $\Omega_{ar}$
and $\Omega_{ca}$ are subject to high uncertainty ($\sigma > 0.175$) between 30°S-30°N (Fig. 9d and A5b). Regarding the $\sigma$-to-$\mu$ ratio, mean
uncertainty estimates per cell for the saturation states in the (sub-) tropical band are relatively small compared to the mean
of the best monthly estimates (Figs. A9-ef). The Arctic and the coastal oceans remain the regions with largest reconstruction
uncertainties for $\Omega_{ar}$ and $\Omega_{ca}$, as well as for $pCO_2$ and $pH$ (Figs. A9-ab). Excluding these regions, $R(\sigma, \mu)$ (Eq. 8) is less than
0.3% for $pH$ and 8% for $\Omega_{ar}$ and $\Omega_{ca}$.

The monthly CMEMS-LSCE reconstruction at $0.25°$ resolution is assessed against $pH$ measurements from GLODAPv2.2022
bottle data (Table 3). For the period 1985-2021, the global RMSD amounts to 0.022 [0.060] $pH$ units and $r^2$ scores at 0.70
[0.45] over the open [coastal] ocean (Table 5). Model bias lies within $[-0.01, 0.01]$ $pH$ units and RMSD is below 0.02 $pH$ units
over the open ocean, except for high latitudes over $60°$ (Figs. 8-ef). At local scale, the eight coastal time series from Sutton
et al. (2019) are used for further evaluation. There exists much less evaluation data for $pH$ than for $pCO_2$, e.g., only 2 months
of monitoring $pH$ at COASTALLA and FIRSTLANDING (Table A3). Monthly time series of CMEMS-LSCE $pH$ are coherent
with these $pH$ measurements (Tables A2 and A3). Measurement uncertainty of $pH$ at these coastal sites is reported to be around
0.02 $pH$ units. RMSD can be as small as 0.035 and 0.04 $pH$ units at CCE2 and GRAYSREEF while it is over 0.05 $pH$ units
at the other stations (e.g., COASTALLA: 0.068, CAPEARAGO: 0.069). Similar to the $pCO_2$ time series (Fig. 5, Sect. 5.1),
$pH$ has been monitored with low sampling frequency (roughly a few days in the tracking month) and the temporal sampling
deviation of instantaneous observations from monthly averages ($\sigma^t_{pH}$) is significant. This temporal sampling uncertainty of $pH$
contributes to the mismatch between model estimates and observations. For example, $\sigma^t_{pH}$ amounts to 0.048 $pH$ units at CCE2
and 0.020 $pH$ units at GRAYSREEF, and reaches highest values of 0.078 $pH$ units at COASTALLA and 0.086 $pH$ units at CA-
PEARAGO. Although model-observation misfit and model uncertainty remain high over the coastal sector (see also Figs. 8-ef
and 9c), their estimates do not surpass 1% of the global mean $pH$ (8.082). The reconstructed $pH$ time series reproduce mea-
surement variability with relatively high correlation, $r^2$ in $[0.21, 0.69]$, that reinforces the reliability of CMEMS-LSCE $pH$
data.

### 5.3.2 Ocean acidification: key features from global to local scales

The monthly, $0.25°$ CMEMS-LSCE datasets of $pH$, $\Omega_{ar}$, and $\Omega_{ca}$ are at the basis of of two CMEMS ocean indicators moni-
toring surface ocean acidification from 1985 to 2021: (1) annual global means and (2) global trend maps.





In Fig. 10, we present annual global means of surface ocean $pH$ and saturation states for aragonite ($\Omega_{ar}$). An illustration of

calcite ($\Omega_{ca}$) is provided in the Appendix (Fig. A13a). For each variable, the calculation of annual global area-weighted means of best estimates (line) and $1\sigma$-uncertainties (envelope) follows Eq. (7). The trends reported in the legend result from linear least-squares regression on annual global means of 100-ensembles of the carbonate system variables. These ensembles are generated with Gaussian distribution having the mean and variance as best model estimate $\mu$ and squared uncertainty ($\sigma^2$) at monthly time steps and $0.25°$-grid cells, respectively. $pH$ decreases from $8.110 \pm 0.017$ in 1985 to $8.049 \pm 0.014$ in 2021 with

a descend rate of $-0.017 \pm 0.004$ decade$^{-1}$. Similar trends are found for the surface ocean saturation states with respect to calcium carbonate minerals. The global mean estimates of $\Omega_{ar}$ [$\Omega_{ca}$] amount to $3.141 \pm 0.198$ [$4.807 \pm 0.302$] and $2.862 \pm 0.174$ [$4.372 \pm 0.266$] for the open and coastal oceans. The saturation state declines at a rate of $-0.080 \pm 0.029$ decade$^{-1}$ with respect to aragonite while the reduction is steeper for calcite ($-0.114 \pm 0.045$ decade$^{-1}$).

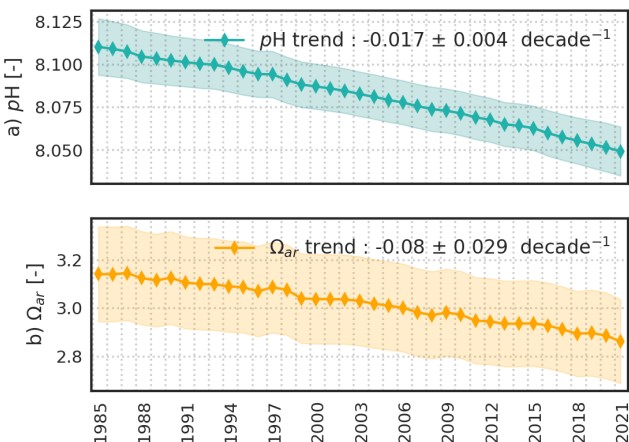

**Figure 10.** Yearly global area-weighted mean of surface seawater $pH$ reported on total scale (a) and surface ocean saturation states with respect to aragonite (b). Global means of the best estimate ($\mu$, plain line) and of uncertainty ($\sigma$, envelop) are computed with Eq. (7a). Trend and uncertainty in the legend are computed with linear regressions on the 100-member ensemble of yearly global means for each variable.

Global trend maps of surface ocean $pH$, $\Omega_{ar}$, and $\Omega_{ca}$ over the entire period are illustrated in Figs. 11 and A13b. Linear least-

squares regression is used to estimate secular trends at every $0.25°$-grid cell. The linear fits of each variable against time rely on the 100-member ensemble generated with the best estimates and propagated uncertainties of $pH$, $\Omega_{ar}$, and $\Omega_{ca}$ (see Figs. A14 for examples). Regression slope and residual standard deviation estimates are defined as linear trend and uncertainty of $pH$, $\Omega_{ar}$, and $\Omega_{ca}$. Hatched area represents $pH$ [$\Omega_{ar}$ and $\Omega_{ca}$] trend estimates ($\mu$) with highest uncertainties ($\sigma$), i.e., $\sigma$-to-$\mu$ ratio (Eq. 8) above $10\%$ [$20\%$]. These regions include a portion of the Arctic, Antarctic, equatorial Pacific, and coastal ocean (Figs. 11,

A11, and A12). $95\%$ of $pH$ trend estimates over the global ocean is in the range of $[-0.022, -0.012]$ decade$^{-1}$ (Fig. 11a). In the broad open ocean of the tropics and subtropics, $pH$ has been declining around $-0.018$ decade$^{-1}$ to $-0.012$ decade$^{-1}$. Faster decrease rates are found in the Indian Ocean and Southern Ocean with values between $-0.022$ and $-0.018$ decade$^{-1}$. Fastest reductions are computed for the eastern equatorial Pacific and the Arctic with rates exceeding $-0.025$ decade$^{-1}$. A



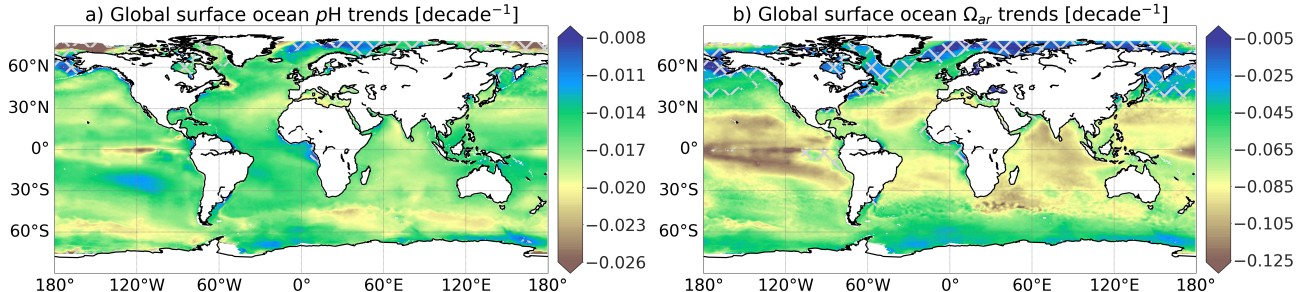

**Figure 11.** Global trend maps of surface seawater $p$H reported on total scale (a) and surface ocean saturation states with respect to aragonite (b). Linear trend of CMEMS-LSCE $p$H and $\Omega_{ar}$ is estimated per $0.25°$-grid cell over 1985-2021. Cross-hatching covers the regions where uncertainty over $10\%$ [$20\%$] of $p$H [$\Omega_{ar}$] trend estimates.

similar magnitude of $p$H trends over these regions is also found in (Lauvset et al., 2015; Leseurre et al., 2022; Ma et al.,

2023). The spatial distribution of saturation states with respect to calcium carbonate minerals generally shows the opposite latitudinal pattern (Figs. 11b and A13b). The magnitude of $\Omega_{ar}$ [$\Omega_{ca}$] trends over the $30°$S-$30°$N band can be as large as $-0.086$ decade$^{-1}$ [$-0.134$ decade$^{-1}$] to the greatest extent of $-0.186$ decade$^{-1}$ [$-0.275$ decade$^{-1}$] (e.g., eastern equatorial Pacific). Trends of $\Omega_{ar}$ and $\Omega_{ca}$ computed in polar and subpolar northern hemisphere regions are not significant.

Trend estimates derived from reconstructions of $p$H and $\Omega_{ar}$ are evaluated at seven time series stations (Bates et al., 2014)

in Table 6. Time series locations are shown in Fig. A1b. With the exception of CARIACO and HOT excepted for which pH measurements are available, long-term trend estimates by Bates et al. (2014) rely on time series of $p$H and $\Omega_{ar}$ calculated via speciation from measurements of $A_T$ and DIC. A 100-member ensemble of monthly time series of $p$H and $\Omega_{ar}$ are extracted from the $0.25°$-grid box nearest to each monitoring station. Linear least-squares regression is then used to infer estimates of their secular trends and associated uncertainties (see Fig. A14 for illustration). Trend estimates derived from CMEMS-LSCE

reconstructions at HOT, BATS, ESTOC, and Munida are in line with previous studies for both $p$H and $\Omega_{ar}$ (Dore et al., 2009; González-Dávila and Santana-Casiano, 2009; Bates et al., 2014). The magnitude of the trend estimate at Irminger Sea for 1985-2012 ($p$H: $-0.014 \pm 0.001$ decade$^{-1}$, $\Omega_{ar}$: $-0.006 \pm 0.011$ decade$^{-1}$) is smaller than that determined by Bates et al. (2014). However, the CMEMS-LSCE $p$H trend is consistent with the estimate by Pérez et al. (2021) ($-0.017 \pm 0.002$ decade$^{-1}$). Moreover, $1\sigma$-uncertainty reported for both $p$H and $\Omega_{ar}$ trend estimates by Bates et al. (2014) is large at this station ($p$H:

$-0.025 \pm 0.006$ decade$^{-1}$, $\Omega_{ar}$: $-0.080 \pm 0.040$ decade$^{-1}$) highlighting the associated uncertainty. Long-term trends of $p$H and $\Omega_{ar}$ are also under-estimated at the Iceland Sea monitoring site, but the bias is not as large as at Irminger Sea (Table 6). Low data sampling frequency at these two stations (Table 1, Bates et al., 2014) could be on account of trend estimate deviation. At CARIACO, the CMEMS-LSCE time series yields a decrease in $\Omega_{ar}$ of $-0.059 \pm 0.053$ decade$^{-1}$, relatively close to Bates et al. (2014) ($-0.066 \pm 0.028$ decade$^{-1}$). The decrease in pH derived from CMEMS-LSCE is, however, larger than in Bates

et al. (2014).



**Table 6.** Secular trend estimates of $p$H and $\Omega_{ar}$ at seven time-series stations (Bates et al., 2014). Trend and uncertainty estimates are reported as $\mu \pm \sigma$. Monthly time series in the CMEMS-LSCE datasets are extracted at the grid box nearest to each station location (Fig. A1b). For the first three stations, this study calculates linear trends starting in the year 1985. Brackets show values computed over the full period 1985-2021.

| Stations | Coordinates | Time span | $p$H trend [decade$^{-1}$] | | $\Omega_{ar}$ trend [decade$^{-1}$] | |
|---|---|---|---|---|---|---|
| | | | Bates et al. (2014) | This sudy | Bates et al. (2014) | This sudy |
| 1. Iceland Sea | 68.00°N 12.66°W | 1983-2012 | $-0.014 \pm 0.005$ | $-0.010 \pm 0.001$ ($-0.014 \pm 0.001$) | $-0.018 \pm 0.027$ | $-0.013 \pm 0.011$ ($-0.025 \pm 0.008$) |
| 2. Irminger Sea | 64.30°N 28.00°W | 1983-2012 | $-0.026 \pm 0.006$ | $-0.014 \pm 0.001$ ($-0.016 \pm 0.001$) | $-0.080 \pm 0.040$ | $-0.006 \pm 0.011$ ($-0.039 \pm 0.009$) |
| 3. BATS | 32.00°N 64.00°W | 1983-2012 | $-0.017 \pm 0.001$ | $-0.014 \pm 0.001$ ($-0.016 \pm 0.001$) | $-0.095 \pm 0.007$ | $-0.079 \pm 0.016$ ($-0.074 \pm 0.010$) |
| 4. ESTOC | 29.04°N 15.50°W | 1995-2012 | $-0.018 \pm 0.002$ | $-0.018 \pm 0.002$ ($-0.019 \pm 0.001$) | $-0.115 \pm 0.023$ | $-0.103 \pm 0.031$ ($-0.089 \pm 0.011$) |
| 5. HOT* | 22.75°N 158.00°W | 1988-2012 | $-0.016 \pm 0.001$ | $-0.016 \pm 0.001$ ($-0.019 \pm 0.001$) | $-0.084 \pm 0.011$ | $-0.100 \pm 0.020$ ($-0.102 \pm 0.011$) |
| 6. CARIACO* | 10.50°N 64.66°W | 1995-2012 | $-0.025 \pm 0.004$ | $-0.017 \pm 0.003$ ($-0.018 \pm 0.001$) | $-0.066 \pm 0.028$ | $-0.059 \pm 0.053$ ($-0.099 \pm 0.018$) |
| 7. Munida | 45.70°S 171.50°E | 1998-2012 | $-0.013 \pm 0.003$ | $-0.017 \pm 0.002$ ($-0.017 \pm 0.001$) | $-0.085 \pm 0.026$ | $-0.088 \pm 0.032$ ($-0.070 \pm 0.009$) |

*Stations with direct observations of $p$H.

## 6 Conclusions and Discussion

This study presents the CMEMS-LSCE product, a dataset of six carbonate system variables (Table 1) covering the global surface ocean at a spatial resolution of $0.25°$ for every month in the period 1985-2021 (444 months). Datasets of individual carbonate system variables are built on the combination of the three methods. First, we adapt an ensemble of 100 feed-forward neural network models (CMEMS-LSCE-FFNN, Chau et al., 2022b) to estimate surface ocean partial pressure of $CO_2$ ($p$CO$_2$) at the pre-defined data resolution. Second, the high-resolution total alkalinity (A$_T$) reconstruction is obtained by using locally interpolated alkalinity regression (LIAR, Carter et al., 2016, 2018). Finally, surface ocean $p$H, total dissolved inorganic carbon (DIC), and saturation states with respect to aragonite ($\Omega_{ar}$) and calcite ($\Omega_{ca}$) are calculated with the carbonate system speciation software (CO2SYS, Lewis and Wallace, 1998; Van Heuven et al., 2011; Orr et al., 2018), given the global monthly reconstructions of $p$CO$_2$ and A$_T$ and other environmental input data (Sect. 3). Results are 2D-fields of the best estimate and associated uncertainty ($1\sigma$) of carbonate system variables available at each grid box of $1\text{month} \times 0.25° \times 0.25°$. $1\sigma$-uncertainty is referred to as the ensemble standard deviation of 100 FFNN outputs for $p$CO$_2$ while it is propagated through the processing chain of LIAR and CO2SYS taking into account different uncertainty sources of input parameters for other variables.





Multiple observation-based datasets, which are not used for the CMEMS-LSCE reconstructions at monthly and $0.25°$ resolu-
tions, serve as benchmarks in the assessments of product quality from global to local scales (e.g., Tables 4, 5, and A3; Figs. 2-5
and 7-8). A summary of the primary statistics for all the six carbonate variables is presented Table 7. Over the full period
1985-2021, CMEMS-LSCE yields global RMSDs of $14.29\,\mathrm{\mu atm}$ and $27.55\,\mathrm{\mu atm}$ in comparison with SOCATv2022 $p\mathrm{CO}_2$ for
the open and coastal oceans, respectively. Temporal variability of observation-based data is well reproduced with $r^2$ of $0.83$
for the open ocean and $0.74$ for the coastal domain. In comparison to CMEMS-LSCE at monthly and $1°$ resolutions (Chau
et al., 2022b), the reconstructions over coastal areas are improved at higher resolution (Figs. 2-4). Furthermore, the monthly,
$0.25°$ reconstruction outperforms its $1°$ counterpart in reproducing horizontal and temporal gradients of $p\mathrm{CO}_2$ over a variety of
oceanic regions as well as at nearshore time series stations (Figs. 2-5). Evaluations with GLODAPv2022 bottle data and time
series stations results in good reconstruction skills for $\mathrm{A_T}$, DIC, and $pH$ at monthly and $0.25°$ resolutions (Tables 5 and A3,
Figs. 7 and 8). At the global scale, the open-ocean reconstruction scores a RMSD smaller than $23\,\mathrm{\mu mol\,kg^{-1}}$ and a $r^2$ of $0.9$ in
$\mathrm{A_T}$ and DIC. The model-observation deviation is higher in the coastal zone. However, it does not exceed $5\%$ of the global mean
values and $r^2$ is above $0.6$ for both coastal $\mathrm{A_T}$ and DIC. Regarding $pH$, the CMEMS-LSCE reconstruction provides estimates
with RMSD$= 0.022$ $[0.060]$ and $r^2 = 0.7$ $[0.45]$ over the open [coastal] ocean. From the statistics in Tables 4 and 5, the Indian
Ocean and the Southern Ocean have poor data density (Fig. A2) but generally show the best global reconstruction among the
ocean basins. Thus, model evaluation with different numbers of observation data might not reflect a fair comparison of skill
scores (e.g., RMSD and $r^2$) between regions. Data density is much higher in the Arctic, Atlantic, and Pacific than in the Indian
and Southern Oceans. The increased data density reveals stronger spatio-temporal variability, for instance, related to coastal
dynamics or upwelling than resolved in the two latter basins. RMSD and $r^2$ computed on the lower data variability result in
better model scores.

The spatial distribution of long-term mean $1\sigma$-uncertainty estimates (Figs. 1b, 6cd, and 9cd) indicates higher confidence
levels for open-ocean estimates than over the coastal sector. The evaluation of temporal mean $1\sigma$-uncertainty estimates relative
to climatological mean values $\mu$ (Figs. 1a, 6ab, and 9ab) results in $\sigma$-to-$\mu$ ratio (Eq. 8) below $5\%$ and $8\%$ for $p\mathrm{CO}_2$ and $\Omega_{ar}$,
$2\%$ for $\mathrm{A_T}$ and DIC, and $0.4\%$ for $pH$ over the open ocean (Fig. A9). The $\sigma$-to-$\mu$ ratio reaches values as high as $10\%$ to $20\%$
for $p\mathrm{CO}_2$ and $\Omega_{ar}$ in the coastal domain. The global mean of open-ocean $1\sigma$-uncertainty estimates (Eq. 7a) for CMEMS-LSCE
$p\mathrm{CO}_2$ $(8.48\,\mathrm{\mu atm})$, $\mathrm{A_T}$ $(16.66\,\mathrm{\mu mol\,kg^{-1}})$, DIC $(15.75\,\mathrm{\mu mol\,kg^{-1}})$, $pH$ $(0.011)$, and $\Omega_{ar}$ $(0.180)$ are in line with those reported
by previous studies despite being derived from different statistics. For instance, Iida et al. (2021) calculated $1\sigma$-uncertainty
based on the median absolute deviation of regression model fits from open-ocean observations. Their approach yielded global
$\sigma$-averages of $17.8\,\mathrm{\mu atm}$, $11.5\,\mathrm{\mu mol\,kg^{-1}}$, $0.018$, and $0.110$ for $p\mathrm{CO}_2$, normalized DIC, $pH$, and $\Omega_{ar}$, respectively. In Gregor
and Gruber (2021), the authors propagated the sum squared errors (global RMSD and measurement uncertainties) of $p\mathrm{CO}_2$
$(15\,\mathrm{\mu atm})$ and $\mathrm{A_T}$ $(22\,\mathrm{\mu mol\,kg^{-1}})$ obtaining global uncertainty estimates of $19\,\mathrm{\mu mol\,kg^{-1}}$ in DIC and $0.022$ in $pH$. Mean
uncertainty estimates over the coastal region are on the order of twofold that computed for the open ocean for these four
variables (Table 7), corroborating results by Gregor and Gruber (2021) (Fig. 7).

Our high-resolution carbon data product opens the door to various analyses of the marine carbonate system from global to lo-
cal scale. This study exemplifies an application of the data for monitoring ocean acidification over recent years. The monitoring



**Table 7.** Summary in global evaluation statistics for CMEMS-LSCE surface ocean carbonate system datasets at monthly, $0.25^\circ$ resolutions over the period 1985-2021. $\mu$ and $\sigma$ stand for the global area-weighted means of monthly best estimates and $1\sigma$-uncertainties for each variable (Eq. 7). RMSD (Eq. 10) and $r^2$ (Eq. 11) are computed with SOCATv2022 for $pCO_2$ and GLODAPv2.2022 for $pH$, $A_T$, and DIC. The division between the coastal (C) and open (O) oceans is at 400 km on a distance from the shore line (Fig. A1a).

| Variables | Units | Sector | $\mu$ | $\sigma$ | RMSD | $r^2$ |
|---|---|---|---|---|---|---|
| 1. $pCO_2$ | $\mu$atm | (O) | 364.48 | 8.48 | 14.29 | 0.83 |
| | | (C) | 359.35 | 17.10 | 27.55 | 0.74 |
| 2. $A_T$ | $\mu$mol kg$^{-1}$ | (O) | 2305.78 | 16.66 | 22.09 | 0.90 |
| | | (C) | 2263.02 | 38.36 | 82.01 | 0.72 |
| 3. DIC | $\mu$mol kg$^{-1}$ | (O) | 2031.12 | 15.75 | 22.67 | 0.90 |
| | | (C) | 2008.65 | 33.40 | 72.39 | 0.62 |
| 4. $pH$ | - | (O) | 8.082 | 0.011 | 0.022 | 0.70 |
| | | (C) | 8.082 | 0.021 | 0.060 | 0.45 |
| 5. $\Omega_{ar}$ | - | (O) | 3.059 | 0.180 | - | - |
| | | (C) | 2.864 | 0.206 | - | - |
| 6. $\Omega_{ca}$ | - | (O) | 4.674 | 0.275 | - | - |
| | | (C) | 4.384 | 0.314 | - | - |

indicators derived from the monthly, $0.25^\circ$ surface ocean CMEMS-LSCE product consist of (1) yearly global means of surface ocean $pH$ and saturation states with respect aragonite $\Omega_{ar}$ and calcite $\Omega_{ca}$ and (2) global maps of multi-annual trends of surface ocean $pH$, $\Omega_{ar}$, and $\Omega_{ca}$ (Figs. 10, 11, and A13). In 1985, the global mean surface ocean $pH$ was $8.110 \pm 0.017$. It was $8.049 \pm 0.014$ in 2021 (Fig. 10a). Over the same 37-year time period, $\Omega_{ar}$ decreased from $3.141 \pm 0.198$ to $2.862 \pm 0.174$ (Fig. 10b). The rate of decline of surface ocean $pH$ and $\Omega_{ar}$ was respectively $-0.017 \pm 0.004$ decade$^{-1}$ and $-0.080 \pm 0.029$ decade$^{-1}$ since 1985 (see also results for $\Omega_{ca}$ in Sect. 5.3.2). Estimates of $pH$ trend lie between $[-0.022, -0.012]$ decade$^{-1}$ across most of the open ocean (Fig. 11a). In general, surface ocean $pH$ decreased more rapidly in the Indian Ocean and Southern Ocean than the tropics and subtropics. These findings are in close agreement with the suggestions by Lauvset et al. (2015) and Ma et al. (2023) but future studies would need to include analyses of underlying drivers to provide insight into regional differences in pH changes. By contrast, the greatest reduction in surface ocean saturation states (Fig. 11b) was computed for the two latter regions. The global trend maps of $pH$ and $\Omega_{ar}$ highlight the Eastern Equatorial Pacific as one of the vulnerable regions with respect to ocean acidification. In this area, the decline rate of $pH$ exceeds $-0.025$ decade$^{-1}$ and $-0.186$ decade$^{-1}$ for $\Omega_{ar}$. The comparison of multi-annual trends of $pH$ and $\Omega_{ar}$ at time series stations (Table 6 and Fig. A14) highlighted the consistency between CMEMS-LSCE estimates and previous studies (Dore et al., 2009; González-Dávila and Santana-Casiano, 2009; Bates et al., 2014; Pérez et al., 2021). For most of these sites, the trends evaluated for 1985-2021 are greater than those relative to the sub-period before the year 2012. The faster rate of ocean acidification over the full period compared to the pre-2012 probably reflects a steeper acceleration in ocean uptake of anthropogenic $CO_2$ in the last decade.

The production chain of CMEMS-LSCE carbonate system variables will be maintained and further improvements with the aim to reduce model-observation misfit and improve the quantification of model uncertainty are on the way forward. Being at the core of the chain, model upgrades of CMEMS-LSCE-FFNN will be tackled first. At the time, SOCAT does not provide open-ocean data of $CO_2$ fugacity gridded at monthly, $0.25^\circ$ resolutions. Our ensemble-based approach draws thus



on two SOCATv2022 data sources: a "downscaled" version of the $1°$-open-ocean data and the $0.25°$-coastal-ocean data (see Sect. 2.1). Open-ocean SOCAT datasets gridded at finer regular resolutions (if accessible) will be updated to gain more accuracy in our model fitting. Selections of data products for predictors needed for model input are equally important. For instance, the CMEMS SSS product used here results in a globally good reconstruction of total alkalinity (Table 5). However, the temporal variability in CMEMS SSS data does not match that in observations (Fig. A10) and this feature is retained in time series of total

alkalinity (Fig. 7). Despite best efforts in determining overall product uncertainty in estimates of carbonate system variables, part of input uncertainty is still not taken into account or only partially quantified due to lack of time-space varying uncertainty fields associated with predictor variables (e.g. SSH, Chl-a, MLD, nutrient concentrations). Moreover, temporal sampling bias in $pCO_2$ and $pH$ is likely to contribute to deviations between observations and model output (Fig. 5 and Table A3). The total measurement error uncertainty should be considered with great care during reconstruction and model output evaluation.

The CMEMS-LSCE approach leads as the first series of long-term reconstructions of $pCO_2$, $pH$, $A_T$, DIC, $\Omega_{ca}$ and $\Omega_{ar}$ extending seamlessly from the global open ocean to coastal regions at monthly, $0.25°$ resolutions. Future use cases recommended for this high-resolution product include (1) estimation of monthly to interannual variations, long-term trends of carbonate system variables, as well as of air-sea $CO_2$ exchanges at the surface layer from local scale to large ocean basins, (2) analyses in interactions between these variables and effects of other physical and biogeochemical factors on ocean acidification and

changes in the marine carbonate system, (3) assessments of horizontal and temporal gradients of carbonate system variables in the coastal-open ocean continuum, (4) evaluation or combination with other model- or observation-based products (e.g., Biogeochemistry Argo, Southern Ocean Carbon and Climate Observations and Modeling), and (5) improvements in coastal reconstructions based on observation system simulation experiments (e.g., with finer spatio-temporal model resolutions). The CMEMS-FFNN surface ocean carbon product at monthly, $0.25°$ resolutions will be accessible through the CMEMS data portal

(see Sect. 7).

## 7 Data availability

The CMEMS-LSCE datasets of six carbonate system variables have been delivered to the European Copernicus Marine Environment Monitoring Service (CMEMS, Product ID: MULTIOBS_GLO_BIO_CARBON_SURFACE_REP_015_008, DOI: 10.48670/moi-00047). Since November 2022, the product with monthly and $1°$ resolutions is available at the CMEMS portal

(Chau et al., 2022a, b). The CMEMS-LSCE data product at monthly and $0.25°$ resolutions proposed in this study will replace its coarser resolution version in due course. For the time being, the high-resolution data product described in this manuscript can be accessed via repository under data DOI: 10.14768/a2f0891b-763a-49e9-af1b-78ed78b16982 (Chau et al., 2023).



## Appendix A: Definitions of ocean carbonate system variables

Chemical reactions of dissolved $CO_2$ in seawater follow a series of the following equilibria,

$$CO_2(g) \rightleftharpoons CO_2(aq), \tag{A1a}$$

$$CO_2(aq) + H_2O \rightleftharpoons H^+(aq) + HCO_3^-(aq), \tag{A1b}$$

$$HCO_3^-(aq) \rightleftharpoons H^+(aq) + CO_3^{2-}(aq), \tag{A1c}$$

where (g) and (aq) stand for a gas or the species in an aqueous solution. $CO_2(aq)$ refers to the combination of aqueous $CO_2$ and its weak acid $H_2CO_3$. $HCO_3^-$ and $CO_3^{2-}$ are bicarbonate and carbonate ions.

Definitions of the essential variables involved in the carbonate system equilibria (A1) are on the list below (see in Dickson et al., 2007; Dickson, 2010; Gattuso and Hansson, 2011, for further details).

i) Surface ocean $pCO_2$ is partial pressure of $CO_2$ in air which is in equilibrium with that in water sample. It is not the same as surface ocean fugacity of $CO_2$ ($fCO_2$). $pCO_2$ can be converted from $fCO_2$ via

$$pCO_2 = fCO_2 \exp\left(-P\frac{B+2\delta}{RT^*}\right). \tag{A2}$$

where $P$ is total atmospheric pressure at surface water, $T^*$ is absolute temperature, $R$ is the gas constant, and $B$ and $\delta$ are cross-virial coefficients (Körtzinger, 1999).

ii) Seawater $pH$ is a negative logarithmic scale of total concentration of hydrogen ions ($H^+$) in aqueous solution. Total $H^+$ is the sum of concentrations of free $H^+$ and $HSO_4$ ions. The $pH$ scale typically ranges from 0 to 14. $pH = 7$ is the threshold specifying whether a water sample is in acidic (i.e., $pH < 7$) or basic (i.e., $pH > 7$) conditions.

iii) Total alkalinity ($A_T$) measures the capacity of seawater against acidification. By definition, $A_T$ is total concentration of dissolved alkaline substances corresponding to the ability in $H^+$ attracting over $H^+$ releasing. The major contributions to alkalinity includes bicarbonate ($HCO_3^-$), carbonate ($CO_3^{2-}$), and hydroxide ($OH^-$) ions. Total alkalinity can be approximated with

$$A_T = [HCO_3^-] + 2[CO_3^{2-}] + [OH^-] - [H^+]. \tag{A3}$$

iv) Total dissolved inorganic carbon (DIC) is the sum in concentrations of the three primary aqueous species in seawater,

$$DIC = [HCO_3^-] + [CO_3^{2-}] + [CO_2(aq)]. \tag{A4}$$

v) Calcium carbonate saturation state ($\Omega$) is defined as follows,

$$\Omega = \frac{[Ca^{2+}][CO_3^{2-}]}{K_{sp}}, \tag{A5}$$





where $[\text{Ca}^{2+}]$ is the concentration of dissolved calcium ions and $\text{K}_{sp}$ is the solubility of calcium carbonate in seawater. $\text{CaCO}_3$ has two principal minerals: aragonite and calcite. Aragonite, which is more soluble than calcite ($\Omega_{ar} < \Omega_{ca}$), is produced by many marine shells and skeletons including corals, pteropods, clams, and mussels. A $\Omega_{ar}$ value greater than 1, i.e., preferable conditions in shell formation, indicates supersaturated seawater with respect to aragonite, and vice versa.

vi) Revelle factor (RF) measures the buffer capacity for the carbonate system in seawater that decreases as $p\text{H}$ increases. Revelle factor is expressed by the ratio between instantaneous changes of dissolved $\text{CO}_2$ $\left( \frac{[\Delta\text{CO}_2(\text{aq})]}{[\text{CO}_2(\text{aq})]} \right)$ and of DIC $\left( \frac{[\Delta\text{DIC}]}{[\text{DIC}]} \right)$ in seawater,

$$\text{RF} = \frac{[\Delta\text{CO}_2(\text{aq})]}{[\text{CO}_2(\text{aq})]} \left( \frac{[\Delta\text{DIC}]}{[\text{DIC}]} \right)^{-1}. \tag{A6}$$

*Author contributions.* TTTC, FC, and MG developed the CMEMS-LSCE-FFNN model at a quarter-degree resolution. TTTC has prepared script codes and executed the experiments with support from FC in setting and running the model at the HPC resources of TGCC. TTTC, MG, and FC shaped the first manuscript version. All the authors contribute to revise the manuscript based on NM's suggestions.

*Competing interests.* The author and co-authors have declared that they have no competing interests.

*Acknowledgements.* This research has been supported by the MOB TAC project of the European Copernicus Marine Environment Monitoring Service (CMEMS) (https://marine.copernicus.eu/about/producers/mob-tac, last access: 14 March 2023). It was granted access to the HPC resources of TGCC under the allocation A0110102201 made by GENCI. The Surface Ocean $\text{CO}_2$ Atlas (SOCAT, www.socat.info, last access: 20 March 2023) is an international effort, endorsed by the International Ocean Carbon Coordination Project (IOCCP), the Surface Ocean Lower Atmosphere Study (SOLAS) and the Integrated Marine Biogeochemistry and Ecosystem Research program (IMBER), to deliver a uniformly quality-controlled surface ocean $\text{CO}_2$ database. We thank Anna Conchon for her help in testing and wrapping LIAR and CO2SYS Matlab toolboxes.





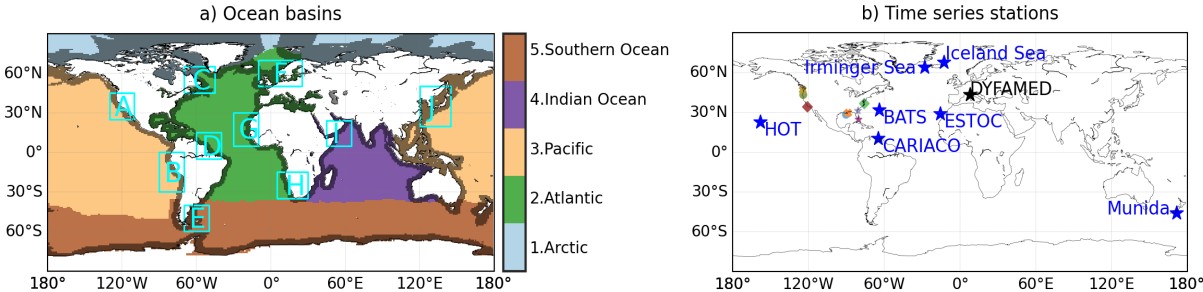

**Figure A1.** a) Ocean basins (https://github.com/RECCAP2-ocean/RECCAP2-shared-resources/tree/master/data/regions, last access: 11/7/2022): coastal mask (grey, approximately 400 km from the shore line), feature regions analyzed in this study (cyan box, Table A1); b) Location of time series stations recording in situ observations used in data evaluation (Table 3): blue stars (Bates et al., 2014), black star (Coppola et al., 2021), and other coloured scattered objects (Sutton et al., 2019).

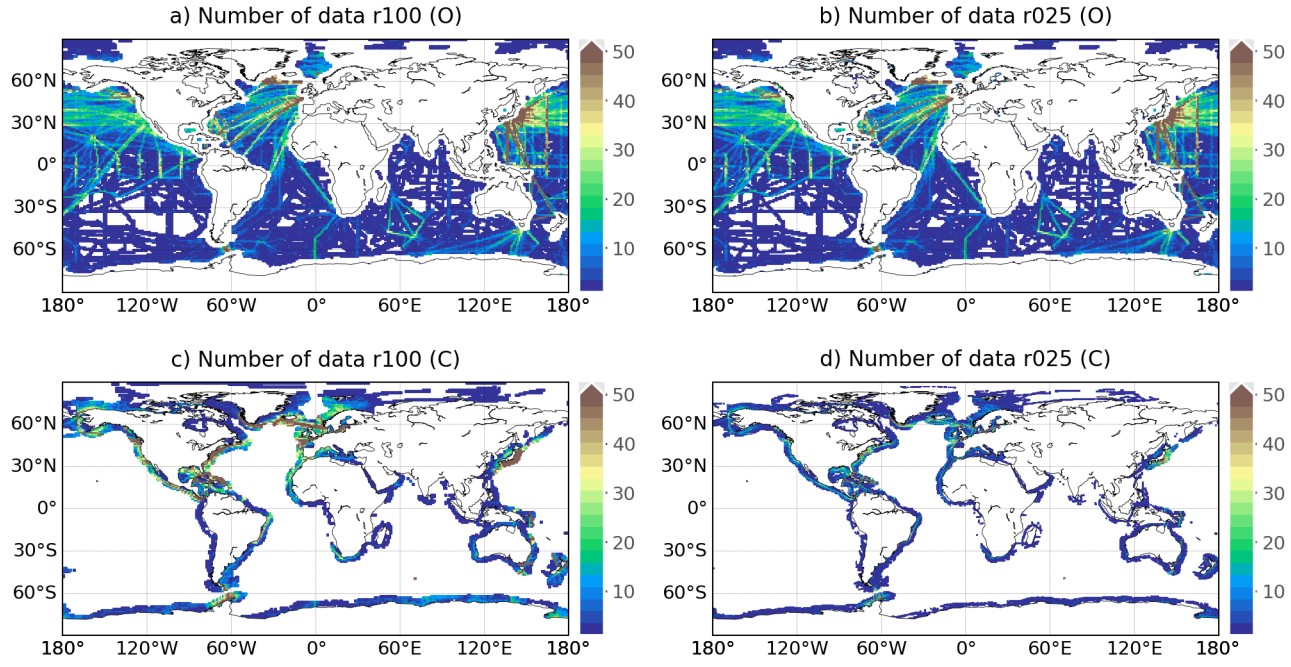

**Figure A2.** Spatial distribution of total months in 1985-2021 containing SOCATv2022 gridded data. Left: $1°$-data product (r100), right: $0.25°$-data product (r025). Open-ocean data (O) in each $0.25°$-grid box is created by setting conservatively the open-ocean SOCATv2022 data at the $1°$-grid box containing it. The coastal-ocean SOCATv2022 data (C) are assigned within 400 km from the shoreline (Fig. A1a).



**Table A1.** Information of feature regions analyzed in this study (Fig. A1a - cyan boxes).

| Notations | Regions | Coordinates | |
|---|---|---|---|
| | | Latitude | Longitude |
| A | California Current System | 25°N-45°N | 130°W-110°W |
| B | Humboldt Current System | 30°S-0° | 90°W-70°W |
| C | Labrador Sea | 45°N-65°N | 70°W-45°W |
| D | Amazon river mount | 5°S-15°N | 60°W-40°W |
| E | Western South Atlantic | 60°S-40°S | 70°W-50°W |
| F | Northern Europe | 50°N-70°N | 10°W-25°E |
| G | Canary Current System | 5°N-30°N | 30°W-10°W |
| H | Benguela Current System | 35°S-15°S | 5°E-20°E |
| I | Western Arabian Sea | 5°N-24°N | 45°E-65°E |
| J | Sea of Japan | 30°N-50°N | 120°E-150°E |

**Table A2.** Information of moored time series of coastal-surface-ocean $p\text{CO}_2$ and $p$H observations (Sutton et al., 2019).

| Stations | Abbreviations | Coordinates | Date range |
|---|---|---|---|
| 1. Cape Arago | CAPEARAGO | 43.3°N, 124.5°W | 06/2017-12/2020 |
| 2. Cape Elizabeth | CAPEELIZABETH | 47.4°N, 124.7°W | 06/2006-05/2020 |
| 3. California Current Ecosystem 2 | CCE2 | 34.3°N, 120.8°W | 01/2010-06/2021 |
| 4. Cheeca Rocks Ocean Acidification Mooring in Florida Keys National Marine Sanctuary | CHEECAROCKS | 24.9°N, 80.6°W | 12/2011-12/2021 |
| 5. Coastal Louisiana buoy | COASTALLA | 28.5°N, 90.3°W | 07/2017-08/2020 |
| 6. Central Gulf of Mexico Ocean Observing System Station 01 | COASTALMS | 30.0°N, 88.6°W | 05/2009-05/2017 |
| 7. Chesapeake Bay Interpretive Buoy System Ocean Acidification Buoy at First Landing | FIRSTLANDING | 37.0°N, 76.1°W | 04/2018-09/2020 |
| 8. NDBC Buoy 41008 in Gray's Reef National Marine Sanctuary | GRAYSREEF | 31.4°N, 80.9°W | 07/2006-08/2018 |

**Table A3.** Statistics computed between CMEMS-LSCE datasets ($0.25°$) and time series of $p\text{CO}_2$ and $p$H measurements (Sutton et al., 2019): total numbers of monthly mean observations (N), temporal standard deviation of observations from their monthly averages ($\sigma^t$), RMSD (Eq. 10), and $r^2$ (Eq. 11). See Table A2 and Fig. A1b for stations' information and locations.

| Stations | $p\text{CO}_2$ [$\mu$atm] | | | | $p$H [-] | | | |
|---|---|---|---|---|---|---|---|---|
| | N | $\sigma^t$ | RMSD | $r^2$ | N | $\sigma^t$ | RMSD | $r^2$ |
| 1. CAPEARAGO | 33 | 65.60 | 79.86 | 0.19 | 31 | 0.086 | 0.069 | 0.22 |
| 2. CAPEELIZABETH | 92 | 42.54 | 41.44 | 0.52 | 11 | 0.061 | 0.057 | 0.69 |
| 3. CCE2 | 127 | 45.31 | 32.44 | 0.16 | 58 | 0.048 | 0.035 | 0.24 |
| 4. CHEECAROCKS | 73 | 44.05 | 62.42 | 0.25 | 40 | 0.038 | 0.066 | 0.21 |
| 5. COASTALLA | 22 | 59.15 | 57.41 | 0.52 | 2 | 0.078 | 0.068 | - |
| 6. COASTALMS | 41 | 43.04 | 42.50 | 0.51 | 15 | 0.062 | 0.065 | 0.25 |
| 7. FIRSTLANDING | 17 | 69.98 | 77.32 | 0.49 | 2 | 0.061 | 0.042 | - |
| 8. GRAYSREEF | 96 | 20.12 | 38.34 | 0.65 | 49 | 0.020 | 0.040 | 0.66 |



**Figure A3.** Global maps of mean model-data difference (Bias, abcd) and root-of-mean-square-deviation (RMSD, efgh) between the reconstruction and SOCATv2022 $p$CO$_2$ [µ atm] over 1985-2021. Left: CMEM-LSCE-FFNN with a resolution of $1°$ (r100), right: CMEM-LSCE-FFNN with a resolution of $0.25°$ (r025). Open-ocean data (O) in each $0.25°$-grid box used for evaluation is created by setting conservatively the open-ocean SOCATv2022 data value at the $1°$-grid box containing it. Coastal-ocean data (C) are extracted from each of the two SOCATv2022 gridded data products.

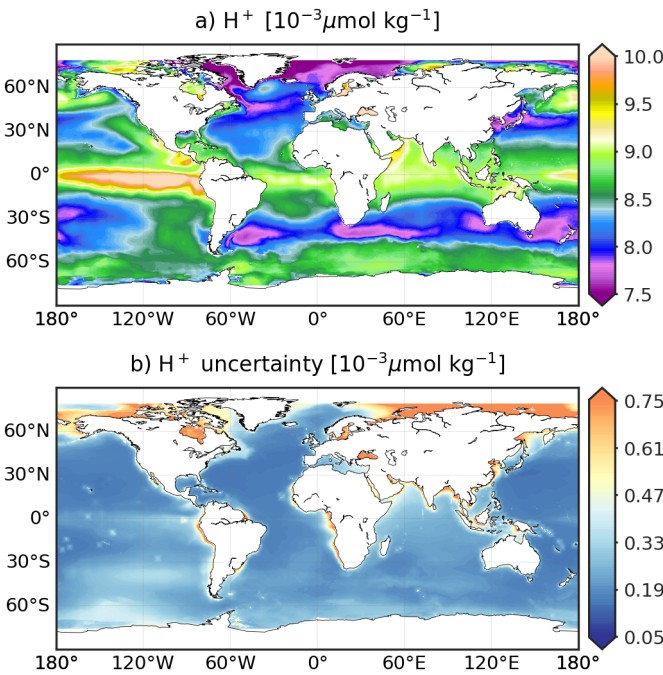

**Figure A4.** CMEMS-LSCE $H^+$ over the global ocean at a spatial resolution of $0.25°$. Temporal means of the model best estimate and $1\sigma$-uncertainty per grid cell over 1985-2021 are calculated by using Eq. (5).

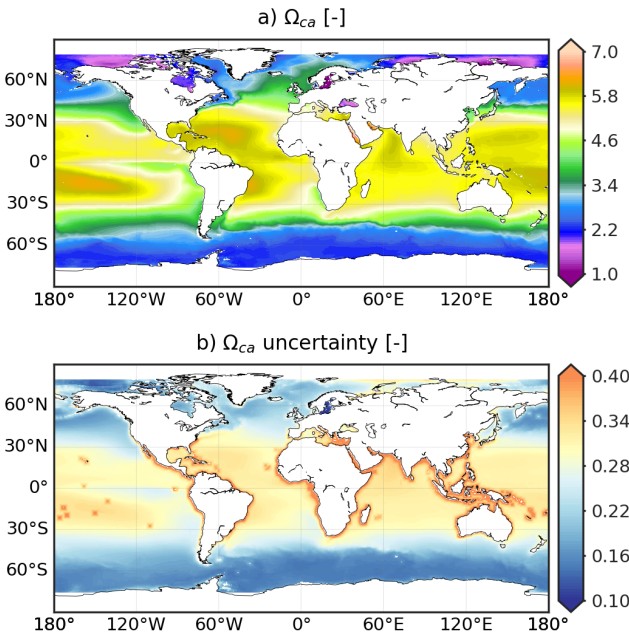

**Figure A5.** CMEMS-LSCE $\Omega_{ca}$ over the global ocean at a spatial resolution of $0.25°$. Temporal means of the model best estimate and $1\sigma$-uncertainty per grid cell over 1985-2021 are calculated by using Eq. (5).

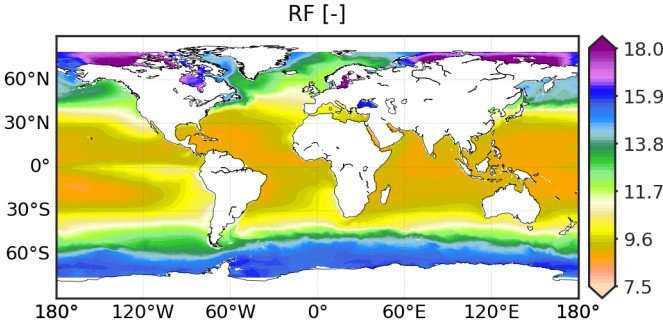

**Figure A6.** CMEMS-LSCE Revelle Factor (RF) over the global ocean at a spatial resolution of $0.25°$. Temporal means of the model best estimate per grid cell over 1985-2021 are calculated by using Eq. (5).



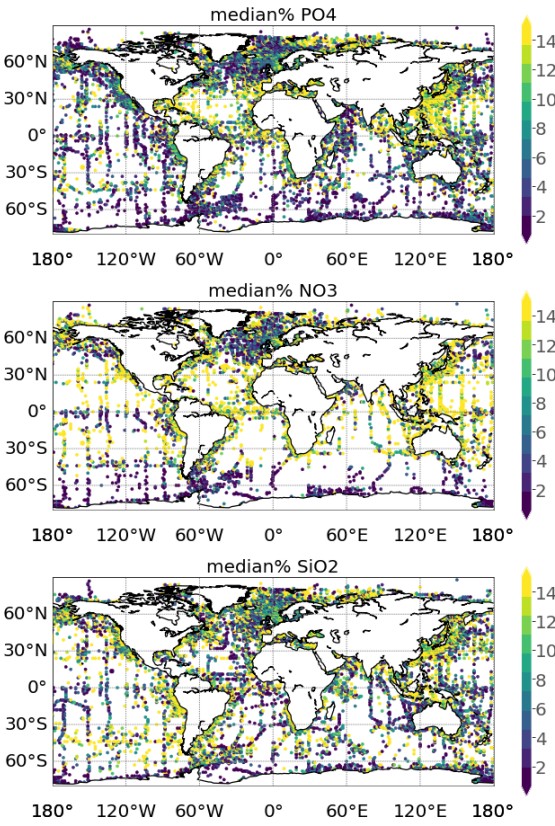

**Figure A7.** Median percentage of analysis error uncertainty against of climatological mean of surface WOA18 nutrient data: phosphate ($PO_4$), nitrate ($NO_3$), and silicate ($SiO_2$).

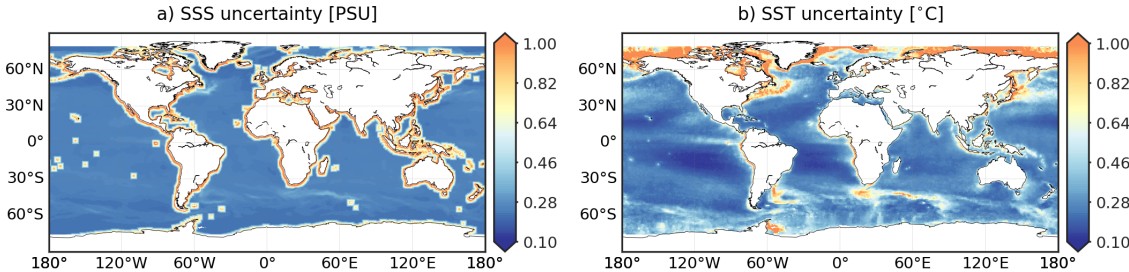

**Figure A8.** Spatial distribution of CMEMS SSS and SST product uncertainty over the global ocean at a spatial resolution of $0.25°$. $1\sigma$-uncertainty is computed per grid cell by using Eq. (5) over 1985-2021.

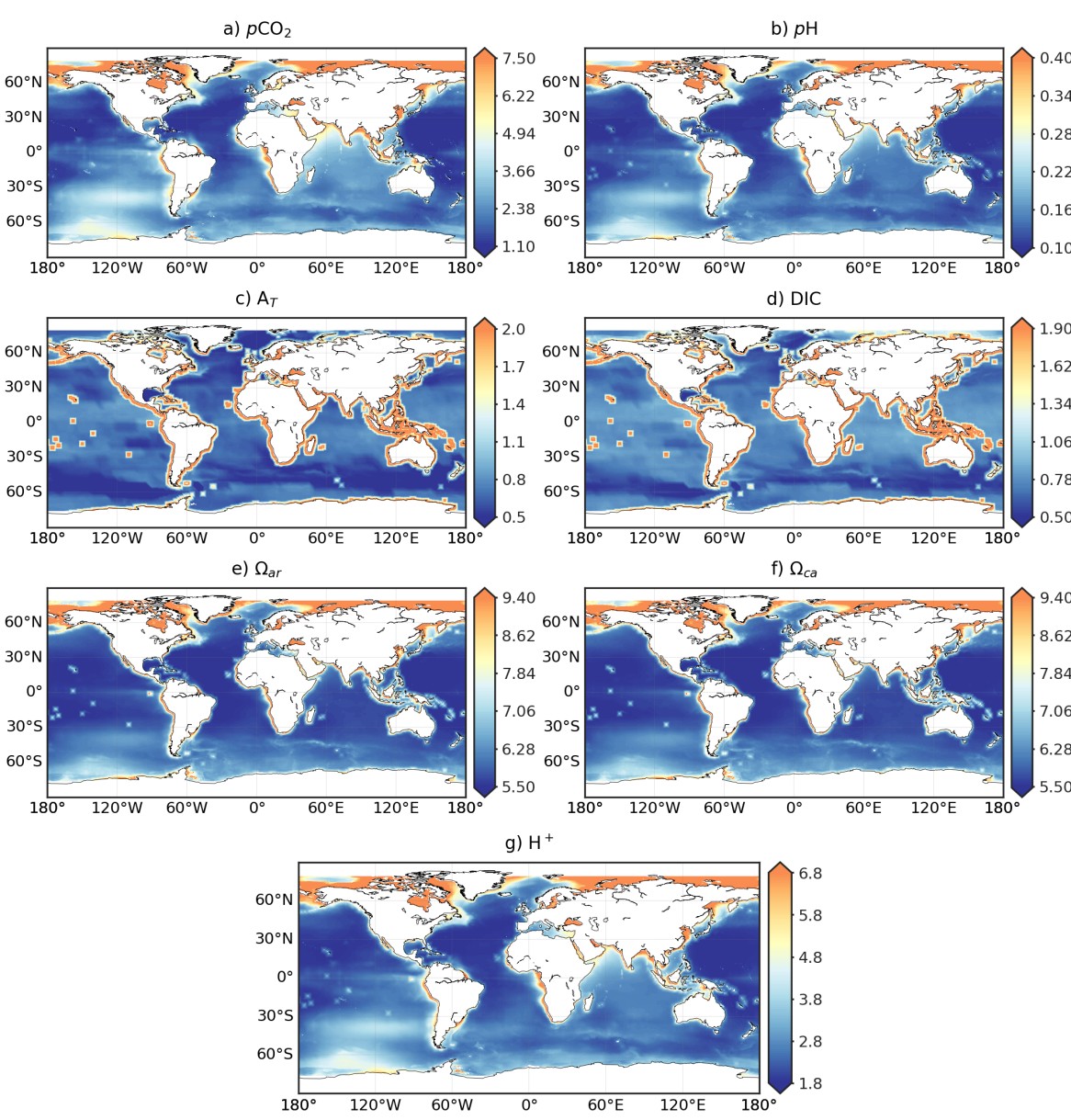

**Figure A9.** Spatial distribution of $R(\sigma, \mu)$ [%] (Eq 8), i.e., the ratio of model uncertainty ($\sigma$) against model best estimate ($\mu$).



**Figure A10.** Monthly time series of SSS and SST at BATS, DYFAMED, ESTOC, and HOT stations (Table 3 and Fig. A1b): model best estimate (curve), $1\sigma$-uncertainty (envelope), and monthly average of observations (point). Means of the best estimate and $1\sigma$-uncertainty ($\mu \pm \sigma$) calculated over the observing time span are shown in brackets if accessible. Statistics include number of months with observations ($N$), Bias, RMSD, and $r^2$. $\sigma^t_{\text{SSS}}$ [$\sigma^t_{\text{SST}}$] stands for temporal standard deviation from monthly averages of SSS and SST observations. Temporal variations in SSS observations are poorly described in the CMEMS SSS time series (Table 2) used in CMEMS-LSCE reconstructions.



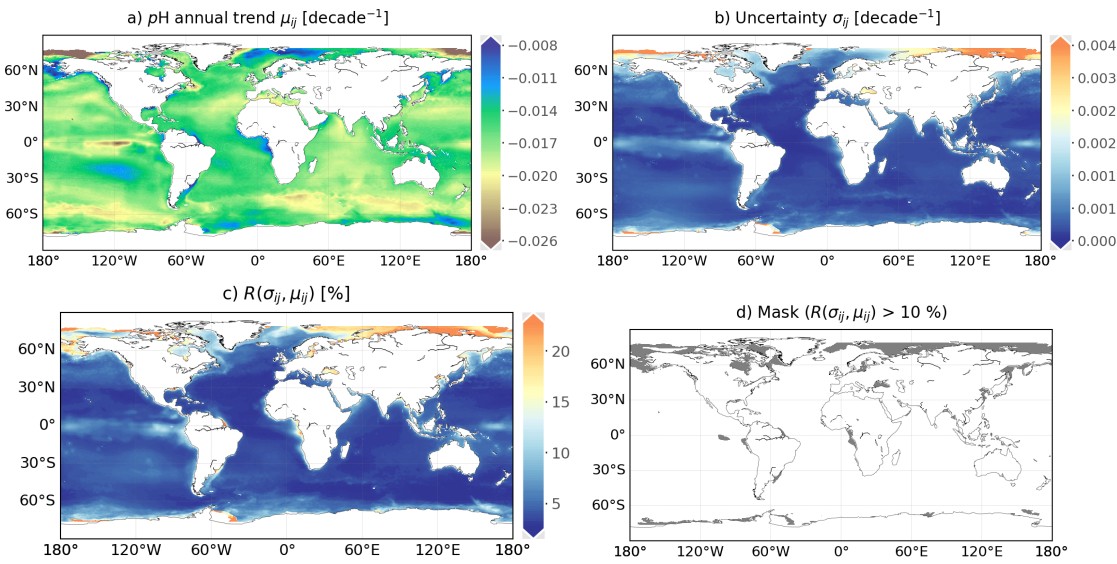

**Figure A11.** a) Global surface seawater $pH$ trend over the period 1985-2021, b) $1\sigma$-uncertainties associated to trend estimates, c) $\sigma$-to-$\mu$ ratio $R(\sigma, \mu)[\%]$ (Eq. 8) between uncertainty estimates (b) and the best trend estimates (a), d) mask applied over the regions where $R(\mu, \sigma) > 10\%$.

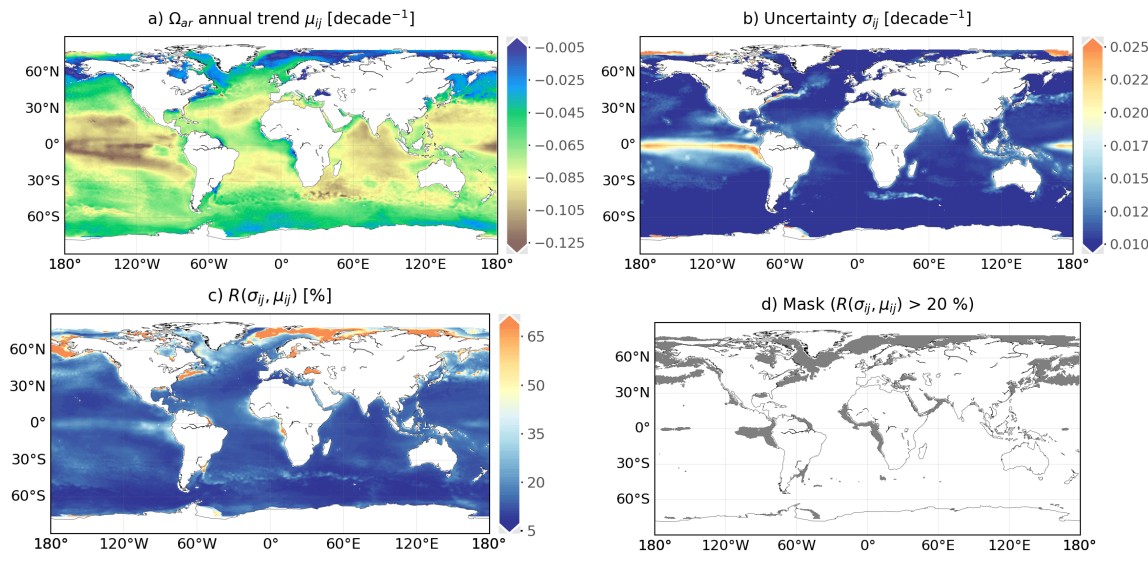

**Figure A12.** a) Global surface seawater $\Omega_{ar}$ trend over the period 1985-2021, b) $1\sigma$-uncertainties associated to trend estimates, c) $\sigma$-to-$\mu$ ratio $R(\sigma, \mu)[\%]$ (Eq. 8) between uncertainty estimates (b) and the best trend estimates (a), d) mask applied over the regions where $R(\mu, \sigma) > 20\%$.

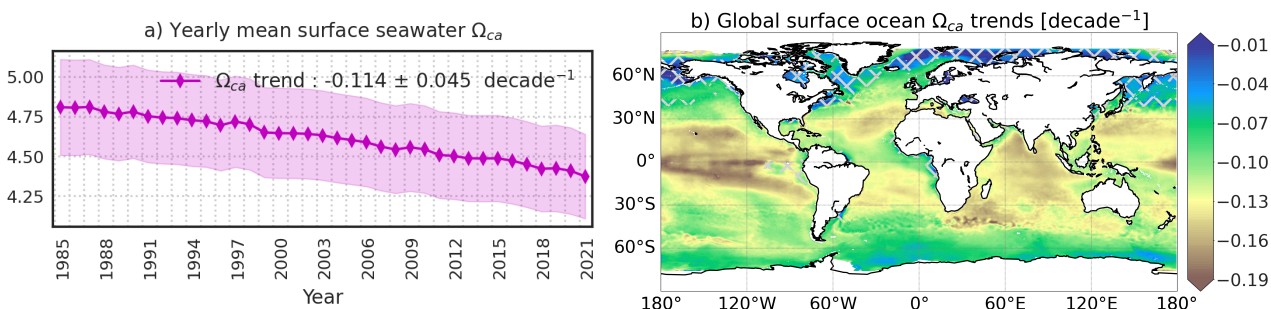

**Figure A13.** a) Yearly global area-weighted mean of surface ocean saturation states with respect to calcite ($\Omega_{ca}$): Global means of the best estimate ($\mu$, plain line) and uncertainty ($\sigma$, envelop) are computed with Eq. (7a). b) Global trend maps of $\Omega_{ca}$ over 1985-2021: Cross-hatching covers the regions where uncertainty of a trend estimate over 20% of the trend value.



**Figure A14.** Linear trend estimates learned on 100-member ensemble (grey points) of yearly mean time series of $pH$ and $\Omega_{ar}$ at different stations (Bates et al., 2014). $\mu \pm \sigma$ present linear slope and residual standard deviation. Black or blue lines stand for linear fits over the full or sub-period in 1985-2021 (see Table 6 for comparison).





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
