# Peer review of "CMEMS-LSCE: A global 0.25-degree, monthly reconstruction of the surface ocean carbonate system"

_Earth System Science Data, 2023_

## Referee Comment (RC1)

**General comments**

Chau et al. present a exclusive approach using discrete ocean surface data of $pCO_2$ and total alkalinity (TA) to obtain a new monthly reconstruction for the period 1985 to 2021 with 0.25° resolution of the marine carbonate system variables. The reconstruction is based on the use of a feed-forward neural network (FFNN) for $pCO_2$. For TA they use locally interpolated alkalinity regression (LIAR). The reconstruction is based on the CMEMS (Copernicus Marine Environment Monitoring Service) product, which provides global reconstructions of sea surface temperature (SST) and surface salinity (SSS) for the same period, including chlorophyll and other physical variables such as sea surface height. The authors start from a previous work where they published a similar database made with a resolution of 1° where only $pCO_2$ has been reconstructed. Here they expand the resolution by increasing it to 0.25° with the inclusion of TA, and then using the thermodynamic equations of the marine carbonate system they obtain the variables: Dissolved Inorganic Carbon (DIC), pH and degree of saturation of aragonite and calcite. In this way a product is generated that can be used to evaluate the impact of ocean acidification by other users and stake-holders. The quality of the reconstruction is contrasted with values observed at a series of oceanic and other coastal time stations. The authors provide two databases, one with 1°C resolution and the other with 0.25° resolution.

The motivation and idea behind the paper is not original in the sense that this has been done before on a seasonal climate scale, but instead, here, the authors exploit the potential of CMEMS to obtain a reconstruction of all carbonate system variables on a spatial scale that has not been achieved so far and that can certainly be very useful in the evaluation of biogeochemical models and for the study of ocean acidification and in coastal regions of higher variability.

The article is well written and provides detailed information both in the formalization of the equations and in the graphical information that is extended in the figures and equations of the appendix. However, it does not develop a specific discussion section of this new database or a comparison with other climatologies of $pCO_2$, DIC, AT and pH that would allow us to see the benefits, improvements and qualities of the new product. The authors, instead, compare in the 'Conclusions and Discussion' section the acidification rates with other observational results of other authors.

The source of information for the $pCO_2$ reconstruction is the Surface Ocean $CO_2$ Atlas version 2022 (SOCATv2022, 1985-2021) observations of CO2 fugacity ($fCO_2$). This database provides not only $fCO_2$ but the data are **REQUIRED** to be accompanied by SST and SSS. The $fCO_2$ data cannot be used independently of the SST and SSS with which it has been reported, since temperature in the observation of $fCO_2$ has a high impact on the $fCO_2$ value itself (a bias of 1°C generates a bias in $pCO_2$ of 4.2%, ~18 µatm). The development of the $pCO_2$ reconstruction expressed in equation (1) does not meet that

requirement. The authors mix the SOCAT observations with the SST and SSS reconstructions of the CMEMS product. This generates important errors as they themselves show in the reconstruction at oceanic (Figure 7) and coastal (Figure 5) fixed stations. Similarly, with TA, the observations used in LIAR also use temperature and salinity in Global Ocean Data Analysis Project bottle data version 2.2022 (GLODAPv2.2022, Lauvset et al., 2022). GLODAPv2 does not report TA without temperature and salinity observations so neither should different data sources be mixed when applying the LIAR methodology as is done in equation 2. Therefore, methodologically, **the manuscript is seriously flawed in its numerical approach**. The process should be done in two stages, first obtaining a set of FFNNs trained with $fCO_2$, SST and SSS with the SOCAT data (and additionally the variables already included in equation 1), and then projecting that FFNN onto CMENS' own reconstructions of SST and SSS. The same is true for TA and the use of LIAR. At least the SSS used in equation 2 should include the GLODLAP SSS and not the CMEMS SSS. Better is to include the GLODAP SST, also. Then the coefficients developed with LIAR are used on the CMENS reconstruction. This would greatly improve the reliability of the algorithms by better reproducing both the oceanic and coastal time series, not to mention that the GLODAP reconstructions shown in Figure 8 will do so as well. All this allows us to have a better estimate of the quality of the obtained algorithms since we can apply them to both oceanic and coastal time series with their own predictors and validate these algorithms. As currently performed in the manuscript, this validation is strongly biased because the SST and SSS reconstructions of CMENS on these series clearly disagree when comparing point data with monthly means as indicated in the manuscript itself in Figure A8. In addition, a simple linear regression of TA versus salinity would report a better fit than the LIAR model applied in the manuscript.

As shown above, the monthly reconstruction proposed by the authors would be strongly improved if the two-step process is applied. The current product shown has a very poor quality in terms of validity since its comparison with the fixed time-series station used shows very high RMSD values (Figure 5, Table 7 and A3).

**Minor comments**

Line 18: "*reconstructions with root-of-mean–square–deviation from observations less than 8%, 4%, and 1% relative to the global mean*" The relative percentage of RMSD over the mean is not a good parameter to evaluated the goodness of the results. For example, the accuracy of AT is better than 0.1%, and $pCO_2$ is similar. The percentages of RMSD reported are about two orders of magnitude higher. .

Line 20: "*and 0.4% for pH*" It is a bit odd to report percentages of a logarithmic magnitude such as pH.

Line 92: The associated uncertainty reported in the article (σ) refers only to the uncertainty of the 100 replicate FFNNs, but they do not incorporate the uncertainty that each of the FFNNs has with respect to the SOCAT $pCO_2$ values they are trying to replicate. The paper is only assessing a part of the uncertainty, by the way the smallest part and therefore not evaluating the ability of the FFNN set to reconstruct the input values.

Table 1, Table 2 and also Table 3 should include a value or an estimate of the uncertainty of each of the variables, either in their analytical determination or that which each product or reconstruction generates for each of the variables. This helps the reader to evaluate the quality of the reconstruction as a function of own error in the determination of each of the reconstructed variables.

Line 126 Table 3 is cited before Table 2.

Line 214. It is not sufficiently clear how to proceed with the reconstruction. It talks about excluding data in the month of reconstruction. Therefore, it would appear that for each month 100 FFNN reconstructions are performed. If this is correct, the RMSD for each month should be included in the figure or table of the SOCAT $pCO_2$ reconstruction since that data is not used in the month-specific reconstruction.

Line 249 Fig A7 is not cited in order.

Line 285 and 210: How do you solve the discontinuities of the variable 'longitude' around the prime meridian 0°. This is usually solved using the sine and cosine functions of longitude. Any reason for not doing so? Does this variable really bring any improvement in the FFNN?

Table 4. First of all, it should be pointed out that there is an excess of significant figures, not only in this table but throughout the text. Regarding the $pCO_2$ results, the authors should remove all decimal places since analytically its precision is 2 μatm as described in the article. But more importantly, once the superfluous decimal places have been removed, what is observed is that there is practically no significant improvement between the product 'r025' and 'r100'.

Line 350 How is the regriding process performed? What type of interpolation is performed?

Line 354 *'The FFNN(r025) central to this study yields a lower RMSD and a higher correlation to the SOCAT data than the FFNN(r100→ 025)'*. **Unfortunately, there is no significant difference between the two products. This statement is not correct.**

Line 375. The differences in RMSD between the regridded r100 and r025 products are very small, or even in some as in Canary Current System it is larger (strange?). There is no significant improvement in the coastal regions between the two products.

Line 393. It seems a very marginal the 2% improvement in pCO$_2$ reconstruction capability

Lines 404-423. "*Analyzing the eight station time series, we have found that data have been sampled within a few days with an average offset of about a week from the month center. At these coastal sites, the temporal standard deviation from monthly averages of pCO$_2$ exceeds measurement errors (2 µatm, Sutton et al., 2019). pCO2 ranges from 20.12 µatm at GREYREFF to values as large as 65.6 µatm at CAPEARAGO or 69.98 µatm at FIRSTLANDING. The monthly average of pCO$_2$ might not be adequately represented by discreet samples at sites with a large temporal standard deviation of pCO2. The misfit between the monthly reconstruction and discreet observations is exacerbated in dynamical coastal environments and might explain in part the large RMSD of reconstructions of monthly coastal pCO$_2$ (e.g., GREYREEF: 38.34 µatm, CAPEARAGO: 79.86 µatm, FIRSTLANDING: 77.32 µatm) for the r025 reconstruction. The RMSD is mostly lower for the FFNN reconstruction at 0.25° resolution compared to the FFNN at 1° resolution by 2.11 µatm (CCE2) to 23.32 µatm (COASTALMS). Similarly, r2 increases between 7%-23% at higher resolution. Overall, seasonal to interannual variations of coastal-ocean pCO$_2$ are better reproduced in the reconstruction at 0.25° resolution (Fig. 5).*" Here, it becomes evident that comparing monthly reconstructions with point values in coastal areas of high variability results in very low predictive ability on the part of the product produced. As indicated in the general comment, this should be evaluated considering the variability of SST and SSS in the study area because in this way the biases that the CMENS product has to reproduce point values from monthly mean values are being transferred to pCO$^2$. The aforementioned increases in r$^2$ are relatively small if we consider the important biases involved, which in some products even increase as the resolution improves, as in FIRSTLANDING or CHEECAROCKS.

Line 437-438 "*The largest model uncertainty (σ > 30 µmol kg$^{-1}$) is computed nearshore and surrounding oceanic islands, a feature inherited from input uncertainty associated with the CMEMS salinity product (Fig. A8a).*" This described here is very relevant. In fact, it would be necessary to show graphically the correlation between the uncertainty in TA and SSS in the CMEMS product in both the coastal and oceanic domains. Possibly it shows a very relevant correlation. A similar should be done with the uncertainties of pCO$_2$ and SST in the CMEMS product.

Line 451-465. "*The reconstruction of AT distributions relies on LIAR coefficients fit with GLODAPv2 data (Olsen et al., 2016) covering the years before 2015. These data are also part of the latest version GLODAPv2.2022 (Lauvset et al., 2022). They do therefore not*

*correspond to an independent dataset for the evaluation data of the CMEMS-LSCE reconstruction. To overcome this limitation, reconstructions of AT and DIC are compared to observations for Eulerian time series stations: BATS, DYFAMED, ESTOC, and HOT (see Table 3 and Fig. A1b for data sources and station locations). Figure 7 illustrates the*

*comparison between monthly time series of AT and DIC extracted from the CMEMS-LSCE datasets and measurements at these long-term monitoring sites".* These lines and Figure 7 show again how a large part of the discrepancies between the TA and DIC reconstruction is due to the discrepancies in SSS and SST of the CMEMS product, indicating that the reconstruction is not well done. In the case of the DYFAMED station it is very noticeable and contrasts that other products such as climatologies like those cited in the article (Lauvset et al. 2016; Broullón et al. 2019) do not show bias as high as the reconstruction performed here.

Line 473 *"The lowest prediction skill of temporal variability is obtained for ESTOC. Particularly, seasonality to multiyear variations in DIC are predicted at $r^2=0.47$ for ESTOC compared to $r^2 > 0.7$ for BATS and HOT."* This is not correct. The regression coefficient is not the only criterion for assessing predictive ability. In this case the variability observed at ESTOC is lower than at BATS and HOT, so a lower $r^2$ does not mean lower skill. In fact, the RMSD at is the lowest of all the stations evaluated in TA. In terms of DIC the three stations show similar RMSD.

Line 478 *"Model uncertainty (1σ-envelop) of monthly AT and DIC estimates (Fig. 7a) is also inflated somewhat proportional to the CMEMS salinity product uncertainty (Fig. A10a)."* Evidently. A figure showing that would be useful. That is why including this product in the LIAR training phase for TA does not help to obtain the best possible reconstruction.

Linea 527 *"The reconstructed pH time series reproduce measurement variability with relatively high correlation, $r^2$ in [0.21,0.69], that reinforces the reliability of CMEMS-LSCE pH"*. It does not seem that the level of correlation obtained with this reconstruction is significant with such low levels of $r^2$. Additionally, the fact that there is no discussion in the article where these levels are compared with other products even if they are only climatic such as those of Takahashi et al. 2014, or others cited in the article for AT and DIC.

Line 576 .- "Conclusions and Discussion" It should be "Discussion and Conclusions" But on the other hand the **discussion is made not in terms of the assessment of the quality of the reconstruction of the product but in terms of the results in terms of ocean acidification**.

Line 594 *"In comparison to CMEMS-LSCE at monthly and 1° resolutions (Chau et al., 2022b), the reconstructions over coastal areas are improved at higher resolution (Figs. 2-4)."* This is not demonstrated in the article. The reduction in RMSD between the two products is very small or marginal.

Line 609 *"The spatial distribution of long-term mean 1σ-uncertainty estimates (Figs. 1b, 6cd, and 9cd) indicates higher confidence levels for open-ocean estimates than over the coastal sector"*. This is very unrepresentative of product quality since it represents the

reproducibility of the 100 FFNN but does not evaluate the RMSD between input and reconstructed data.

Table 7 Both pCO$_2$, AT and DIC quantities should not have decimal places (mean, RMSD).

Line 655 No comparisons with other reconstructions like MODO-DIC of Keppler et al. 2020, or AT from Broullon et al. 2019 or Lee et al. 2006.

References: Keppler, L., Landschützer, P., Gruber, N., Lauvset, S. K., & Stemmler, I. (2020). Seasonal carbon dynamics in the near-global ocean. Global Biogeochemical Cycles, 34, e2020GB006571. https://doi.org/10.1029/2020GB006571

---

## Author Comment (AC1)

**Response to Reviewers' comments on the manuscript**:

**CMEMS-LSCE: A global 0.25-degree, monthly reconstruction of the surface ocean carbonate system**

T. T. T. Chau, M. Gehlen, N. Metzl, F. Chevallier

We would like to thank the two reviewers for reading the manuscript thoroughly and providing constructive feedback. Based on their comments, we have improved the manuscript. Detailed replies are given hereafter. Each section comprises replies for general comments and minor (or specific) comments. Relies to Reviewer 2 start at page 22. The text quoted from the initial manuscript is in blue and revisions are presented in green. The revised manuscript with track changes is attached in this document (after all the replies to the two reviewers).

On behalf of the authors,
Thi-Tuyet-Trang Chau

**Replies to comments by Reviewer 1**

**General comments by Reviewer 1 (GC1)**

**GC1.1.** *Chau et al. present a exclusive approach using discrete ocean surface data of pCO2 and total alkalinity (TA) to obtain a new monthly reconstruction for the period 1985 to 2021 with 0.25° resolution of the marine carbonate system variables. The reconstruction is based on the use of a feed-forward neural network (FFNN) for pCO2. For TA they use locally interpolated alkalinity regression (LIAR). The reconstruction is based on the CMEMS (Copernicus Marine Environment Monitoring Service) product, which provides global reconstructions of sea surface temperature (SST) and surface salinity (SSS) for the same period, including chlorophyll and other physical variables such as sea surface height. The authors start from a previous work where they published a similar database made with a resolution of 1° where only pCO2 has been reconstructed. Here they expand the resolution by increasing it to 0.25° with the inclusion of TA, and then using the thermodynamic equations of the marine carbonate system they obtain the variables: Dissolved Inorganic Carbon (DIC), pH and degree of saturation of aragonite and calcite. In this way a product is generated that can be used to evaluate the impact of ocean acidification by other users and stake-holders. The quality of the reconstruction is contrasted with values observed at a series of oceanic and other coastal time stations. The authors provide two databases, one with 1°C resolution and the other with 0.25° resolution.*

*The motivation and idea behind the paper is not original in the sense that this has been done before on a seasonal climate scale, but instead, here, the authors exploit the potential of CMEMS to obtain a reconstruction of all carbonate system variables on a spatial scale that has not been achieved so far and that can certainly be very useful in the evaluation of biogeochemical models and for the study of ocean acidification and in coastal regions of higher variability.*

**Authors:**

We thank **Reviewer 1** for highlighting potential use cases of our new CMEMS-LSCE datasets of surface ocean carbonate variables at high resolution. The two key points that set this contribution apart from previous studies include (1) a model upgrade for monthly $p\text{CO}_2$ reconstructions in spatial resolution from 1º (Chau et al., 2022) to 0.25º (this study) and (2) an extension to provide high-resolution datasets of $A_T$, DIC, $p$H, and calcium carbonate saturation states covering the 37-year period. We hope the CMEMS-LSCE data product will be explored in further analyses of fine-scale spatiotemporal variations in marine carbonate variables complementary to previous contributions.

%%%%%%%%%%%%%%%%%%%%%%%%%%%%%%%%%%%%%%

**GC1.2.** *The article is well written and provides detailed information both in the formalization of the equations and in the graphical information that is extended in the figures and equations of the appendix. However, it does not develop a specific discussion section of this new database or a comparison with other climatologies of pCO2, DIC, AT and pH that would allow us to see the benefits, improvements and qualities of the new product. The authors, instead, compare in the 'Conclusions and Discussion' section the acidification rates with other observational results of other authors.*

**Authors:**

We thank **Reviewer 1** for appreciating the manuscript presentation. Our evaluation strategy is based on gridded SOCAT, GLODAP, and various time series station data. As suggested by **Reviewer 2** (comment **GC2.2.**) to gain reliability for our data evaluation, in this revision, we have shown new results at additional 38 time series stations with $p$CO$_2$ and $p$H measurements and 4 sites with A$_T$ and DIC measurements. Our product assessment is now stretched across the tropics, the subpolar sector, and the Southern Ocean to complement our previous data evaluation over the subtropical regions (See Figure A1b and statistics added in Tables A3 and A4 in the revised manuscript). An intercomparison with 1°-climatological data reconstructions (e.g., Broullon et al., 2019, 2020; Keppler et al., 2020) seems to us too outside the scope of our study given the importance of underlying methodological choices in such intercomparisons: (1) the discrepancy in mapping methods, input data resource, and the ratio of training and validation datasets used in model fitting, (2) uncertainty from post-processing applied for some products (e.g., filtering, smoothing, calibration), (3) the normalization of different data covering periods, and (4) quality of evaluation (or reference) data; e.g., observation data paucity should be one major concern to evaluate seasonal cycle reconstructions. The effect of such methodological choices calls into question the interpretation of differences between products if the different data providers do not actively contribute with sensitivity studies.

%%%%%%%%%%%%%%%%%%%%%%%%%%%%%%%%%%%%%%

**GC1.3.** *The source of information for the pCO2 reconstruction is the Surface Ocean CO2 Atlas version 2022 (SOCATv2022, 1985-2021) observations of CO2 fugacity (fCO2). This database provides not only fCO2 but the data are REQUIRED to be accompanied by SST and SSS. The fCO2 data cannot be used independently of the SST and SSS with which it has been reported, since the temperature in the observation of fCO2 has a high impact on the fCO2 value itself (a bias of 1°C generates a bias in pCO2 of 4.2%, ~18 μatm). The development of the pCO2 reconstruction expressed in equation (1) does not meet that requirement. The authors mix the SOCAT observations with the SST and SSS reconstructions of the CMEMS product. This generates important errors as they themselves show in the reconstruction at oceanic (Figure 7) and coastal (Figure 5) fixed stations. Similarly, with TA, the observations used in LIAR also use temperature and salinity in Global Ocean Data Analysis Project bottle data version 2.2022 (GLODAPv2.2022, Lauvset et al., 2022). GLODAPv2 does not report TA without temperature and salinity observations so neither should different data sources be mixed when applying the LIAR methodology as is done in equation 2. Therefore, methodologically, the manuscript is seriously flawed in its numerical approach. The process should be done in two stages, first obtaining a set of FFNNs trained with fCO2, SST and SSS with the SOCAT data (and additionally the variables already included in equation 1), and then projecting that FFNN onto CMENS' own reconstructions of SST and SSS. The same is true for TA and the use of LIAR. At least the SSS used in equation 2 should include the GLODLAP SSS and not the CMEMS SSS. Better is to include the GLODAP SST, also. Then the coefficients developed with LIAR are used on the CMENS reconstruction. This would greatly improve the reliability of the algorithms by better reproducing both the oceanic and coastal time series, not to mention that the GLODAP reconstructions shown in Figure 8 will do so as well. All this allows us to have a better estimate of the quality of the obtained algorithms since we can apply them to both oceanic and coastal time series with their own predictors and validate these algorithms. As currently performed in the manuscript, this validation is strongly biased because the SST and SSS reconstructions of CMENS on these series clearly disagree when comparing point data with monthly means as indicated in the manuscript itself in Figure A8. In addition, a simple linear regression of TA versus salinity would report a better fit than the LIAR model applied in the manuscript.*

*As shown above, the monthly reconstruction proposed by the authors would be strongly improved if the two-step process is applied. The current product shown has a very poor quality in terms of validity since its comparison with the fixed time-series station used shows very high RMSD values (Figure 5, Table 7 and A3).*

**Authors:**

- **For $pCO_2$**: Our FFNNs formally link a specific $pCO_2$ estimate (from the gridded SOCAT products) and specific estimates of environmental conditions (from the datasets listed in Table 1[(1)]), as expressed in Equation 1. There is no flaw in this approach that simply exploits the power of FFNNs (which are themselves non-linear regressors). The two-step approach proposed by the
* * *
[1] Table 2 in the initial manuscript (Table 1 has been removed as suggested by **Review 2**, comment **SC2.2**.).

reviewer unnecessarily complicates the reconstruction process. In addition, how the second step ("*projecting that FFNN onto CMENS' own reconstructions of SST and SSS*") can be made is not obvious: it seems to us that it would lose all the benefit gained by the first step.

Note that the bias between CMEMS SST and SOCAT data (or in situ observations) is relatively small (see Figures **GC1.3.** and **MC1.13.** in this document and also Figure A10 in the manuscript). Besides, the quality control has not been done for SOCAT salinity. There are some cruises in SOCAT with no salinity data and they have been replaced by SSS from the World Ocean Atlas (WOA) to recalculate CO2 fugacity (see Pfeil et al, 2013). SOCAT SSS would not be used in data reconstruction unless a critical quality control is performed. We have not changed the method.

[Figure]

*Figure* **GC1.3.** *Scatter plot of SOCAT SST and CMEMS SST gridded data over the global ocean in the period 1985-2021. The two datasets well fit to the bisector (red line) with no systematic bias, a RMSD of 0.17 and a r² of 1.*

- **For A_T**: **Reviewer 1**'s comment (**GC1.3.**) quoted below does not reflect precisely our method application.
  "*Similarly, with TA, the observations used in LIAR also use temperature and salinity in Global Ocean Data Analysis Project bottle data version 2.2022 (GLODAPv2.2022, Lauvset et al., 2022). GLODAPv2 does not report TA without temperature and salinity observations so neither should different data sources be mixed when applying the LIAR methodology as is done in*

*equation 2. Therefore, methodologically, the manuscript is seriously flawed in its numerical approach.*"

LIAR coefficients were estimated with GLODAPv2 data (Olsen el al., 2016) of SSS, SST, $A_T$,... (Carter et al., 2018). In this study, we do not retrain LIAR models but use these existing coefficients to predict $A_T$ with CMEMS SSS and SST (see Lines 257-260[2] in the revision attached at the end of this document):

*"Locally interpolated alkalinity regression (LIAR; Carter et al., 2016, 2018) is an ensemble-based regression method developed for the global reconstruction of total alkalinity (AT ). Regression coefficients were learned on GLODAPv2 data (Olsen et al., 2016) binned within regular windows of 5◦ × 5◦. For prediction, the LIAR software interpolates between the regression coefficients to arbitrary resolutions specified by the users.*"

%%%%%%%%%%%%%%%%%%%%%%%%%%%%%%%%%%%%%%

**Minor comments by Reviewer 1 (MC1)**

| Reviewer's comments | Replies from Authors | Notes |
|---|---|---|
| **MC1.1. Line 18**
"*reconstructions with root-of-mean–square–deviation from observations less than 8%, 4%, and 1% relative to the global mean*" The relative percentage of RMSD over the mean is not a good parameter to evaluated the goodness of the results. For example, the accuracy of AT is better than 0.1%, and pCO2 is similar. The percentages of | Lines 17-19 (abstract): we quote the full sentence below.
"*Product qualification with observation-based data confirms reliable reconstructions with root-of-mean–square–deviation from observations less than 8%, 4%, and 1% relative to the global mean of pCO₂, A_T (DIC), and pH.*"

We are surprised by this comment because normalizing the RMSD, for instance over the mean, may simplify the interpretation of this statistical quantity. In the quoted sentence of the abstract, it allows using the same metric (the relative amplitude of the error) for the four variables. We have not changed the sentence. | |
* * *
[2] Lines 231-234 in the initial manuscript

| | | |
|---|---|---|
| RMSD reported are about two orders of magnitude higher. | **Reviewer 1** mentions that "*the accuracy of $A_T$ is better than 0.1%, and $pCO_2$ is similar*". These values likely correspond to analytical errors based on measurement quality controls at each station/location, e.g., 4 µmol kg−1 for GLODAPv2.2022 $A_T$ (Lauvset et al, 2022) and 2-5 µatm for SOCATv2.2022 $pCO_2$ (Bakker et al, 2022). Unsurprisingly, our reconstruction RMSD is substantially larger than analytical errors, which is only a minor source of uncertainty in the reconstruction process.

Our reconstruction RMSD (e.g. 14.3 µatm, 22.1 µmol kg−1, 22.7 µmol kg−1, 0.022 for $pCO_2$, $A_T$, DIC, and pH) is in line with those reported in the previous studies (see our discussion in Lines 661-667* quoted below):
*"For instance, Iida et al. (2021) calculated 1σ-uncertainty based on the median absolute deviation of regression model fits from open-ocean observations. Their approach yielded global σ-averages of 17.8 µatm, 11.5 µmol kg−1, 0.018, and 0.110 for $pCO_2$, normalized DIC, pH, and $Ω_{ar}$, respectively. In Gregor and Gruber (2021), the authors propagated the sum squared errors (global RMSD and measurement uncertainties) of $pCO_2$ (15 µatm) and $A_T$ (22 µmol kg−1 ) obtaining global uncertainty estimates of 19 µmol kg−1 in DIC and 0.022 in pH."* | *Lines 615-619 in the initial manuscript |
| **MC1.2. Line 20**

*"and 0.4% for pH"* It is a bit odd to report percentages of a logarithmic magnitude such as pH. | In the statistical sense, we consider pH as a variable similar to $pCO_2$ and other carbonate system variables. All statistics are therefore reported with respect to the reconstructed variable. As explained in the previous comment (**MC1.1.**), with the intention of having a concise abstract, we choose to show the percentage of errors against the global mean value of each variable. It is noteworthy that percentages are also used in the scientific report SDG 14.3.1 (Table 1). | |

| **MC1.3. Line 92** | The CMEMS-LSCE-FFNN 100-ensemble approach subsamples the gridded data of $pCO_2$ and predictors to compose different training and test datasets, i.e., 100 training datasets for 100 FFNN models. In practice, it would allow to account for multiple sources of input data uncertainty from measurement errors, data sampling bias, data post-processing, etc, which have been poorly quantified in the input data products so far. In addition, the first layer of FFNNs is also initialized randomly at each of the 100 iterations. Therefore, our ensemble-based uncertainty includes the randomness in both subsampling datasets of $pCO_2$ and predictors and in FFNN initialization. In Chau et al. (2022) (Section Methods), the authors described the ensemble approach comprehensively. This study extends the model by Chau et al. (2022) and thus recaps its principle. | |
| --- | --- | --- |
| The associated uncertainty reported in the article (σ) refers only to the uncertainty of the 100 replicate FFNNs, but they do not incorporate the uncertainty that each of the FFNNs has with respect to the SOCAT pCO2 values they are trying to replicate. The paper is only assessing a part of the uncertainty, by the way the smallest part and therefore not evaluating the ability of the FFNN set to reconstruct the input values. | We modify the text in Lines 239-241* and add another one (in green) as follow for clarification:

"*After excluding the data in the reconstruction month, the data within the 3-month window are randomly separated into FFNN training and validation subsets with a ratio of 2 : 1. The subsampling process is repeated for each 100 FFNN runs that results in 100 different datasets for model fitting.*" | *Lines 214-215 in the initial manuscript |

| | | |
|---|---|---|
| **MC1.4. Table 1, Table 2 and also Table 3** should include a value or an estimate of the uncertainty of each of the variables, either in their analytical determination or that which each product or reconstruction generates for each of the variables. This helps the reader to evaluate the quality of the reconstruction as a function of own error in the determination of each of the reconstructed variables. | Thank you. We have added the measurement errors with respect to each variable in Tables 1 and 2* if they are available from input data resources. | *Tables 2 and 3 in the initial manuscript (Table 1 has been removed as suggested by **Review 2**, comment SC2.2) |
| **MC1.5. Line 126**

 Table 3 is cited before Table 2 | We have revised the manuscript and cited Tables/Figures in order. | |
| **MC1.6. Line 214**

 It is not sufficiently clear how to proceed with the reconstruction. It talks about excluding data in the month of reconstruction. Therefore, it would appear that for each month 100 FFNN reconstructions are performed. If this is correct, the RMSD for each month should be included in the figure or table of the SOCAT pCO2 reconstruction since that data is not used in the month-specific reconstruction. | We quote Lines 238-241* from the revised manuscript for a straightforward response to **Reviewer 1** (modification in green corresponding to our reply to comment **MC.1.3.**):
 "*The datasets of SOCAT fCO$_2$ and predictors are first reprocessed to match model fitting requirements (Sect. 2.1). After excluding the data in the reconstruction month, the data within the 3-month window are **randomly separated into FFNN training and validation subsets with a ratio of 2 : 1. The subsampling process is repeated for each 100 FFNN runs that results in 100 different datasets for model fitting.** The excluded SOCATv2022 datasets are used in model evaluation.*"

     Here we specify the three independent datasets for FFNN training, validation, and evaluation. In the fitting phase of FFNN, we do not use SOCAT fCO$_2$ in the month specified for reconstruction to train and validate | *Lines 213-215 in the initial manuscript |

| | FFNN models. In the reconstruction step, predictors data are available over the global ocean and FFNNs reconstruct $f$CO$_2$ for the target months. This exclusion strategy, called cross-validation, is widely used within machine learning approaches to avoid overfitting. | |
|---|---|---|
| **MC1.7. Line 249**

Fig A7 is not cited in order. | Thank you. We have revised the manuscript and cited Tables/Figures in order. | |
| **MC1.8. Lines 285 and 210**

How do you solve the discontinuities of the variable 'longitude' around the prime meridian 0°. This is usually solved using the sine and cosine functions of longitude. Any reason for not doing so? Does this variable really bring any improvement in the FFNN? | To preserve the continuity of longitude at 0°, we have applied both the sine and cosine functions to that coordinate. Hence, our global maps of carbonate variables (e.g. Figures 1, 6, 9) do not show discontinuity at the prime meridian. The sine is also used to transform latitude. Data transformation of predictor variables is explicitly presented in a sequence of preceding studies for the CMEMS-LSCE-FFNN model development (Denvil-Sommer et al 2021, Chau et al 2022). In the first manuscript version, we avoided repeating part of the data processing and model description from the previous studies. As the readers would concern, we have called back this information in the revision (Lines 162-164):
"***The sine function is applied to convert latitude while both the sine and cosine are used to transform longitude to conserve their periodical behaviors***." | |
| **MC1.9. Table 4**

First of all, it should be pointed out that there is an excess of significant figures, not only in this table but throughout the text. Regarding the pCO2 results, the | The manuscript describes and evaluates long-term datasets of multiple variables. A significant number of figures and tables corresponds to the presentation of many results of these variables.

    $p$CO$_2$ errors (e.g. Bias, RMSD) have been reported with 1-2 decimals in previous studies (Landschuter et al 2020; Denvil et al 2019, Gregor et al. | |

| | | |
|---|---|---|
| authors should remove all decimal places since analytically its precision is 2 µatm as described in the article. But more importantly, once the superfluous decimal places have been removed, what is observed is that there is practically no significant improvement between the product 'r025' and 'r100'. | 2019, 2021). In this revision, we reduce the decimals from 2 to 1 for $pCO_2$, $A_T$, and DIC. The modification has been applied for Tables, Figures, and texts involving these variables. Note that 2-5 µatm reported in the manuscript represents the precision of measurement replications or analytical errors based on measurement quality control at each station/location (Sutton et al., 2019; Bakker et al., 2022).

 Table 3* shows a marginal improvement from r100 to r025 in terms of global evaluation metrics. For the open ocean, we expect to obtain similar skill scores for both FFNN models as the spatial autocorrelation of open-ocean $pCO_2$ is estimated within 400±250 km (Jones et al., 2012) and the SOCAT 1°-open-ocean dataset was used in model fitting. As also noted by Chau et al., (2022), $pCO_2$ over the coastal ocean is characterized by high variability at small scales. For instance, $pCO_2$ levels can vary with a horizontal gradient as large as 470 µatm over a distance of less than 0.5 km (Chavez et al., 2018; Feely et al., 2008). Probably, statistical models would need a spatial resolution much finer than 0.25° (25 km) and a temporal resolution higher than monthly in order to capture such high variability in surface ocean $pCO_2$ present in observations (see also Bakker et al., 2016; Laruelle et al., 2017). In addition, measurement uncertainty of SOCAT gridded data due to undersampling is possibly one of the major sources of the irreducible model-data errors. Please refer to our reply to comment **MC1.11.** for a discussion on the benefits of the higher resolution. | *Tables 4 in the revied manuscript (Table 1 has been removed as suggested by **Review 2**, comment SC2.2) |
| **MC1.10. Line 350**

 How is the regriding process performed? What type of interpolation is performed? | All the 3-dimensional datasets provided in this study have been saved as netCDF numerical files. To regrid these datasets, we use the Climate Data Operators (CDO) remapping operator, namely "remap". CDO remap supports converting netCDF datasets from one horizontal grid to another. This operator | |

| | | |
|---|---|---|
| | has been widely used in standard processing for numerical and statistical model outputs. | |
| | We have revised the last sentence in Lines 374-377* to make it clear to the readers. | *Lines 348-350 in the initial manuscript |
| | "*Table 4** also presents statistics for the monthly FFNN products of surface ocean pCO$_2$ at spatial resolutions of 0.25∘ (r025) and 1∘ (r100) together with their variants (r100 → 025 and r025 → 100). The latter are respectively extrapolation and interpolation versions of the original r100 and r025 datasets. **We used the Climate Data Operators (CDO) remap operator to regrid** FFNN model outputs (r100 and r025) to a finer or coarser spatial resolution.*" | **Table 4 in the initial manuscript (Table 1 has been removed as suggested by Review 2, comment SC2.2.) |
| **MC1.11. Line 354**

"*The FFNN(r025) central to this study yields a lower RMSD and a higher correlation to the SOCAT data than the FFNN(r100→ 025)*". Unfortunately, there is no significant difference between the two products. This statement is not correct. Line 393. It seems a very marginal the 2% improvement in pCO2 reconstruction capability | Our statement is upheld even though the increment in global skill scores relative to a low to high spatial resolution is not large. Here we do not mention getting a significant improvement but still obtained higher scores in RMSD and r$^2$ when increasing the model spatial resolution. Please refer to Table 3* for verifying the statistics with respect to FFNN(r025) and FFNN(r100→r025) and our reply to comment **MC.1.9.** for analysis.

Apart from Table 3*, benefits by increasing model spatial resolution from 1° to 0.25° are also demonstrated in Figures 2-4 with analyses shown in Lines 414-420**:

"*The two FFNN reconstructions (r025 and r100) share similarities in overall structures of pCO$_2$ over the coastal-open-ocean continuum (Figs. 2-4). However, the higher spatial resolution outperforms its lower resolution counterpart in reproducing fine-scale features of pCO$_2$ in the transition from nearshore regions to the adjacent open ocean. The increase in model spatial resolution translates into a greater spatial coverage of the continental shelves such as Labrador Sea, Northern Europe, and Sea of Japan (Fig. 3), and thus* | *Table 4 in the initial manuscript (Table 1 has been removed as suggested by **Review 2**, comment **SC2.2.**)
**Lines 387-394 in the initial manuscript |

| | | |
|---|---|---|
| | *an increase in the number of data over the coastal domain. The increase in spatial resolution allows a gain in prediction probability of pCO$_2$ variations on the order of roughly 2% over the Eastern Boundary Currents to 8% over the Western South Atlantic (Figs. 2-3b).*"

    This study also points out temporal data sampling bias as a source of uncertainty that would highly constrain model reconstruction skills. Based on the assessment at station time series (Figure 5 and Table A3), we found that in situ observations have been sampled with low frequency and the bias of sampling date is about a week from a month center. With the low number of observations and high variability of pCO$_2$ (20.12 to 69.98 µatm) over these stations, it would not be statistically sufficient to refer to their temporal mean as a representative of monthly averages. A large model-data deviation would be retained even if we increase spatial resolution (see text in Lines 434-447*** for further analysis). | ***Lines 403-414 in the initial manuscript |
| **MC1.12. Line 375 and 393**

Line 375. The differences in RMSD between the regridded r100 and r025 products are very small, or even in some as in Canary Current System it is larger (strange?). There is no significant improvement in the coastal regions between the two products. | Thank you for pointing this out. We have revised Figures 2-4. In the previous version, we made a technical error in co-locating the two model outputs to coastal SOCAT grid cells so statistics were not precise enough. The revision slightly modifies RMSD and r$^2$ values over all regions but does change our conclusion. | |
| **MC1.13. Lines 404-423**

"*Analyzing the eight station time series, we have found that data have been sampled within a few days with an average offset of about a week from the month center. At* |     We have demonstrated the better performance of FFNNr025 in terms of intra-seasonal to interannual variability of coastal sites (Sutton et al., 2019). By increasing the model resolution by 16-fold, this study partly resolves the spatial sampling bias from pCO$_2$ observations (lower RMSD and higher r$^2$ for | |

*these coastal sites, the temporal standard deviation from monthly averages of pCO2 ($\sigma_t^{pCO2}$) exceeds measurement errors (2 μatm, Sutton et al., 2019). $\sigma_t^{pCO2}$ ranges from 20.12 μatm at GREYREFF to values as large as 65.6 μatm at CAPEARAGO or 69.98 μatm at FIRSTLANDING. The monthly average of pCO2 might not be adequately represented by discreet samples at sites with a large temporal standard deviation of pCO2. The misfit between the monthly reconstruction and discreet observations is exacerbated in dynamical coastal environments and might explain in part the large RMSD of reconstructions of monthly coastal pCO2 (e.g., GREYREEF: 38.34 μatm, CAPEARAGO: 79.86 μatm, FIRSTLANDING: 77.32 μatm) for the r025 reconstruction. The RMSD is mostly lower for the FFNN reconstruction at 0.25° resolution compared to the FFNN at 1° resolution by 2.11 μatm (CCE2) to 23.32 μatm (COASTALMS). Similarly, r2 increases between 7%-23% at higher resolution. Overall, seasonal to interannual variations of coastal-ocean pCO2 are better reproduced in the*

the higher resolution) although large model-observation mismatches still persist. As replied to comments **MC1.9.** and **MC1.11.**, the sparsity of data samples (biases from observation locations to the grid cell center about 0.34° ± 0.14° as reported in Sabine et al., 2013) and high variability of coastal $pCO_2$ (e.g., 470 μatm in a distance of 0.5km; see in Chavez et al., 2018 and Feely et al., 2008) would draw the conclusion that much higher resolution or extensions of observing system are necessary to fully capture coastal $pCO_2$.

Temporal data sampling bias should be considered as a great source of uncertainty contributing to large model-observation mismatch even though model spatial resolution is getting finer. We illustrate this through Figure 5 with the corresponding analysis being in the paragraph (Lines 437-439*) quoted by **Review 1**.

The key discussion we found is as follows

"*Analyzing the  station time series, we have found that data have been sampled within a few days with an average offset of about a week from the month center. At these coastal sites, the temporal standard deviation from monthly averages of $pCO_2$ ($\sigma_t^{pCO2}$) exceeds **analytical** errors (2 μatm, Sutton et al., 2019)*".

With the low number of observations and high variability of $pCO_2$ over these stations, it would not be statistically sufficient to refer to the temporal mean of instantaneous observations as a representation of monthly averages. We then provide evidence that the large values $\sigma_t^{pCO2}$ at time series stations (e.g., GREYREEF: 20.12 μatm, CAPEARAGO: 65.6 μatm, FIRSTLANDING: 69.98 μatm) correspond to high RMSDs (e.g., GREYREEF: 38.34 μatm, CAPEARAGO: 79.86 μatm, FIRSTLANDING: 77.32 μatm).

About the effect of model-observation bias of SST on the reconstruction skills of $pCO_2$, we refer to our reply to the general comment **GC1.3** above.

*Lines 403-405 in the initial manuscript

*reconstruction at 0.25° resolution (Fig. 5).*"

Here, it becomes evident that comparing monthly reconstructions with point values in coastal areas of high variability results in very low predictive ability on the part of the product produced. As indicated in the general comment, this should be evaluated considering the variability of SST and SSS in the study area because in this way the biases that the CMENS product has to reproduce point values from monthly mean values are being transferred to pCO2. The aforementioned increases in r2 are relatively small if we consider the important biases involved, which in some products even increase as the resolution improves, as in FIRSTLANDING or CHEECAROCKS.

Also illustrated in Figure **MC.1.13.**, the bias between $SST^{CMEMS}$ and in situ SST from Sutton et al's time series is always lower than 0.5°C for many stations. A bias in SST would not be the dominant source of high reconstruction errors at these stations.

[Figure]

***Figure* MC1.13.** *Time series of surface ocean SST (°C) at coastal observing stations\*\*: CMEMS SST estimate (curve), associated 1σ-uncertainty (envelope), and monthly average of in situ observations (point). CMEMS reanalysis data at 0.25◦ (r025) resolutions are co-located to in situ observations provided by Sutton et al. (2019). Statistics include number of months with observations (N), Bias, RMSD, and r². $\sigma_t^{SST}$ stands for temporal standard deviation from monthly averages of SST observations.*

*\*\* See Table A2 and Fig. A1b in the manuscript.*

| | | |
|---|---|---|
| **MC1.14. Lines 437-438**

"*The largest model uncertainty (σ > 30 µmol kg-1) is computed nearshore and surrounding oceanic islands, a feature inherited from input uncertainty associated with the CMEMS salinity product (Fig. A8a).*" This described here is very relevant. In fact, it would be necessary to show graphically the correlation between the uncertainty in TA and SSS in the CMEMS product in both the coastal and oceanic domains. Possibly it shows a very relevant correlation. A similar should be done with the uncertainties of pCO2 and SST in the CMEMS product. | Total alkalinity ($A_T$) is predominantly controlled by the processes that govern sea surface salinity (SSS) (Broecker and Peng, 1982; Millero et al, 1998). The typical relationship between these two variables is linear and can be estimated at a high precision (Lee et al, 2006; Carter et al, 2018; Broullon et al, 2019). From the statistical point of view, the distribution of $A_T$ uncertainty is generally driven by SSS uncertainty: $A_T$ uncertainty increases as SSS uncertainty increases (see Figure **MC1.17.**).

To the contrary, $p$CO$_2$ is characterized by multiple physical, biological, and chemical processes. Uncertainties from many input data products thus contribute to $p$CO$_2$ uncertainty estimates. Drivers of $p$CO$_2$ uncertainty are not analyzed as input uncertainty has not been fully quantified or published so far for many environmental variables. | |
| **MC1.15. Lines 451-465**

"*The reconstruction of AT distributions relies on LIAR coefficients fit with GLODAPv2 data (Olsen et al., 2016) covering the years before 2015. These data are also part of the latest version GLODAPv2.2022 (Lauvset et al., 2022). They do therefore not correspond to an independent dataset for the evaluation data of the CMEMS-LSCE reconstruction. To overcome this limitation, reconstructions of AT and DIC are* | First of all, Lines 488-496* quoted by **Reviewer 1** describes the evaluation of our data product of $A_T$ and DIC and time series of in situ observations. This complements the assessment with GLODAP data. As opposed to the interpretation by **Reviewer 1**, these lines do not contain any analysis about "*how a large part of the discrepancies between the TA and DIC reconstruction is due to the discrepancies in SSS and SST of the CMEMS product and observations*". But we have revised the following sentence to have a better sense (other modifications follow the revisions according comment **GC2.2.**)
"*To  **accomplish a cross-validation**, reconstructions of $A_T$ and DIC are compared to observations for  **eight** time series* | *Lines 451-465 in the initial manuscript |

*compared to observations for Eulerian time series stations: BATS, DYFAMED, ESTOC, and HOT (see Table 3 and Fig. A1b for data sources and station locations). Figure 7 illustrates the comparison between monthly time series of AT and DIC extracted from the CMEMS-LSCE datasets and measurements at these long-term monitoring sites*". These lines and Figure 7 show again how a large part of the discrepancies between the TA and DIC reconstruction is due to the discrepancies in SSS and SST of the CMEMS product, indicating that the reconstruction is not well done. In the case of the DYFAMED station it is very noticeable and contrasts that other products such as climatologies like those cited in the article (Lauvset et al. 2016; Broullón et al. 2019) do not show bias as high as the reconstruction performed here.

stations: **AWIPEV**, *BATS, DYFAMED, ESTOC,*  *HOT,* **ICELAND, IRMINGER, and KERFIX** *(see Table 3 and Fig. A1b for data sources and station locations).*"

In Figure 7 we illustrate both the relatively good and poor reconstructions at long-term time series of observations. Note that, Lauvet et al, (2016) and Broullon et al, (2019) provided climatologies of $A_T$ and DIC and associated mapping errors. As the climatology is smoother than the monthly fields (with intra- to interannual variability) proposed in this study, the errors reported in the previous studies are evidently smaller than those presented here. Their magnitudes are not comparable. In addition, Lauvet et al, (2016) and Broullon et al, (2019) did not evaluate the reconstruction at DYFAMED as mentioned by **Reviewer 1**. Please kindly refer to our reply to **SC2.14.** for a profound analysis of high model-data mismatch of $A_T$ at DYFAMED.
* * *
**MC1.16. Line 473**
"*The lowest prediction skill of temporal variability is obtained for ESTOC. Particularly, seasonality to multiyear variations in DIC are predicted at r2=0.47*

We thank the reviewer for highlighting our good estimates of $A_T$ and DIC at ESTOC in terms of RMSD. We skipped this element in the initial manuscript. We have added it in Lines 515-520*:

*Line 473-476 in the initial manuscript

| | | |
|---|---|---|
| *for ESTOC compared to r2 > 0.7 for BATS and HOT.*" This is not correct. The regression coefficient is not the only criterion for assessing predictive ability. In this case the variability observed at ESTOC is lower than at BATS and HOT, so a lower r2 does not mean lower skill. In fact, the RMSD at is the lowest of all the stations evaluated in TA. In terms of DIC the three stations show similar RMSD. | "***Despite showing good estimates of $A_T$ and DIC in RMSD at ESTOC, temporal variability of observations are reconstructed at the lowest $r^2$.***  *Particularly, seasonality to multi-year variations in DIC are predicted at r2 = 0.47 for ESTOC compared to $r^2$ > 0.7 for* **AWIPEV, ICELAND, IRMINGER, BATS and HOT.** *Over all the stations, the model underestimates temporal changes of $A_T$ (Fig. 7a; BATS: $r^2$ = 0.33, DYFAMED: $r^2$ = 0.12, ESTOC: $r^2$ = 0.03, HOT: $r^2$ = 0.32) which can be attributed to the large discrepancy in variability between in situ measurements and the CMEMS time series of salinity (Fig. A10a; BATS: r2 = 0.33, DYFAMED: $r^2$ = 0.19, ESTOC: $r^2$ = 0.03, HOT: $r^2$ = 0.35).*"

We indeed analyze both RMSD and $r^2$ throughout the manuscript. To be precise, $r^2$ is not '*regression coefficient*' but the determination coefficient. This metric allows evaluating the model predictive ability in temporal variations of the variables of interest. As shown in Eq (11) in the manuscript, $r^2$ is the model-data covariance normalized with the temporal variability of $A_T$ and DIC reproduced by FFNN and observed at each station. Therefore, $r^2$ values at BATS and HOTs are comparable to the one at ESTOC even though the two former stations show higher temporal variability of $A_T$ and DIC. Our analysis in Lines 515-520* holds true. | |
| **MC1.17. Line 478**
"*Model uncertainty (1σ-envelop) of monthly AT and DIC estimates (Fig. 7a) is also inflated somewhat proportional to the CMEMS salinity product uncertainty (Fig. A10a).*" Evidently. A figure showing that would be useful. That is why including this product in the LIAR training phase for | We include here Figure MC2.17 showing the relationship of AT and SSS uncertainty. It is indeed well-known that SSS is the dominant driver of $A_T$ and same for their uncertainty. Please refer to our replies to comments **MC1.14.** and **GC.1.3.** for a further analysis. | |

| | | |
|---|---|---|
| TA does not help to obtain the best possible reconstruction. | **Figure MC2.17.** Scatter plot of monthly uncertainty values of a) CMEMS-LSCE $A_T$ (Figure 7) against CMEMS SSS (Figure A10) at four stations.

[Figure]
 | |
| **MC1.18. Line 527**
"*The reconstructed pH time series reproduce measurement variability with relatively high correlation, r2 in [0.21,0.69], that reinforces the reliability of CMEMS-LSCE pH*".
It does not seem that the level of correlation obtained with this reconstruction is significant with such low levels of r2. Additionally, the fact that there is no discussion in the article where these levels are compared with other products even if they are only climatic such as those of Takahashi et al. 2014, or others cited in the article for AT and DIC. | Similar to $pCO_2$, measurements of pH are subject to undersampling (see also our reply to comment **MC1.11.**). The data density is even lower for pH (see Figures 8 and A3 and Table A3). Remember that the comparison is between monthly mean estimates of pH and the mean of observations which are normally available within some days to a week. Temporal data sampling bias should be considered as a great source of uncertainty attributed to large model-observation mismatch (and thus moderate values of $r^2$) even with a finer model spatial resolution. We show that through Table A3 and an analysis is given in the paragraph (Lines 558-576*).
Please note that the published data cited by Reviewer 1 are climatology. Strictly speaking, statistical evaluations between the climatological product and in situ observation data (sampled with very low frequency) is not robust (see also our reply to the general comment **GC1.2.**) | *Lines 514-530 in the initial manuscript |

| | | |
|---|---|---|
| **MC1.19. Line 576**
"Conclusions and Discussion" It should be "Discussion and Conclusions" But on the other hand the discussion is made not in terms of the assessment of the quality of the reconstruction of the product but in terms of the results in terms of ocean acidification. | Thank you for your suggestion. We have changed Section "Conclusions and Discussion" to "Summary" as proposed by **Reviewer 2** (comment **SC2.16.**). | |
| **MC1.20. Line 594**
"*In comparison to CMEMS-LSCE at monthly and 1° resolutions (Chau et al., 2022b), the reconstructions over coastal areas are improved at higher resolution (Figs. 2-4).*" This is not demonstrated in the article. The reduction in RMSD between the two products is very small or marginal. | We modify the sentences below (Lines 640-643*) to better summarize our results:
"*In comparison to CMEMS-LSCE at monthly and 1° resolutions (Chau et al., 2022b), the reconstructions over coastal areas are improved at higher resolution (Figs. 2-4). Furthermore, tThe monthly, 0.25° reconstruction outperforms its 1° counterpart in reproducing horizontal and temporal gradients of $pCO_2$ over a variety of oceanic regions as well as at nearshore time series stations (Figs. 2-5).*"
The improvement in terms of global metrics is marginal but we still gain advantages when increasing model spatial resolution from 1° to 0.25° (e.g. better capturing horizontal and temporal gradients). In this manuscript, we also try to address the question why increasing the spatial resolution by 16-fold does not impressively reduce model-observation discrepancy. Please kindly refer to our reply to comments **MC1.9.** and **MC1.11.** for a comprehensive explanation. | *Lines 594-597 in the initial manuscript |
| **MC1.21. Line 609**
Line 609 "The spatial distribution of long-term mean 1σ-uncertainty estimates | The reviewer is correct, but, as discussed in Chau et al (2022b), the ensemble spread is a good proxy for the reconstruction uncertainty. | |

| | | |
|---|---|---|
| (Figs. 1b, 6cd, and 9cd) indicates higher confidence levels for open-ocean estimates than over the coastal sector". This is very unrepresentative of product quality since it represents there producibility of the 100 FFNN but does not evaluate the RMSD between input and reconstructed data. | | |
| **MC1.22. Table 7**
Both pCO2, AT and DIC quantities should not have decimal places (mean, RMSD). | $pCO_2$, $A_T$, DIC, and their reconstruction errors/uncertainty have been reported with 1-2 decimals in the previous studies (Landschuter et al 2013, 2020; Denvil et al 2019, Gregor et al. 2019, 2021, Chau et al. 2022b).
In this revision, we reduce the decimals from 2 to 1 for $pCO_2$, $A_T$, and DIC. The modification has been applied for Tables, Figures, and texts involving these variables. | |
| **MC1.23. Line 655**
No comparisons with other reconstructions like MODO-DIC of Keppler et al. 2020, or AT from Broullon et al. 2019 or Lee et al. 2006. | Please see our answer to comments **GC1.2**. and **MC1.15.** | |

**Replies to comments by Reviewer 2**

**General comments by Reviewer 2 (GC2)**

**GC2.1.** *The authors reconstructed 0.25-degree monthly full carbonate system variables during the period 1985-2021 based on surface ocean observation data. Distributions of pCO2 were reconstructed based on the machine learning method established by the authors (Trang-Chau et al. 2022) and those of TA were based on the LIAR method (Carter et al. 2016; 2018). While few reconstructions of full carbonate system variables are available at this moment, a comprehensive understanding of global surface ocean pH distributions is essential for monitoring ocean acidification, which is related to the SDG indicator 14.3.1. This study can enhance researches on the global carbon cycle as well as provide critical information to policymakers and stakeholders. I think this study has sufficient value to be published in this journal, but major concerns listed below should be addressed appropriately. I would like to encourage the authors to improve the study and revise the manuscript for better understanding.*

**Authors:**

We are grateful for **Reviewer 2**'s positive evaluation and constructive comments which help us to improve our manuscript. Please kindly find our replies to address his/her concerns below.

%%%%%%%%%%%%%%%%%%%%%%%%%%%%%%%%%%%%%%%%%

**GC2.2.** *The concept of this study itself is not novel, and the assessment of uncertainty in the reconstructed fields and the validation of the method become important. The authors derived uncertainty distributions in reconstructed parameters from the spread of 100 model ensemble. They also demonstrated the validity of the method by comparing the result of this study with observation data that were not used for learning and those of time-series points. The time-series used in this study are biasedly located in the subtropical region, so comparing their data with the results of this study does not seem a good indicator of uncertainty. For validation of this method, the authors must take a comparison with other reconstruction(s) into account, if needed.*

**Authors:**

Further to the reviewer's suggestion, we have added comparisons to 38 time series stations located outside the subtropics (the tropics, the subpolar sector, and the Southern Ocean). The results appear in Figure A1b, Tables A3 and A4. They confirm the reliability of CMEMS-LSCE datasets (see our analysis in Lines 421-447, 488-522, 558-576 in the revised manuscript attached at the end of this document).

[Figure]

**Figure GC2.2. Revised Figure A1b (right)**: *b) Location of time series stations recording in situ observations used in data evaluation (Table **2**): blue stars **for ocean acidification** (Bates et al., 2014), black stars **for $A_T$ and DIC** (Metzl et Lo Monaco, 1998; **Coppola et al., 2021; Gattuso et al., 2023**), and other coloured scattered objects **for $pCO_2$ and pH** (Sutton et al., 2019). Asterisk (\*) marks the two stations with $A_T$ and DIC observations (Olafsson et al., 2010) available for assessments.*

Suggestions from the two reviewers about an intercomparison with other products are interesting but would bring us well outside the scope of our study if we take them carefully enough. Such intercomparison between data products deserve proper investigations on (1) the discrepancy in mapping methods, input data resource, and the ratio of training and validation datasets used in model fit, (2) uncertainty from post-processing applied for some products (e.g., filtering, smoothing, calibration), (3) the normalization of different data covering periods, and (4) quality of evaluation (or reference) data; e.g., data paucity should be one major concern to evaluate seasonal cycle reconstructions. For $pCO_2$, the evaluation of multiple products including CMEMS-LSCE at 1°, monthly resolutions (Chau et al., 2022) was done in the previous studies (Hauck et al., 2020; Gregor et al., 2021; Friedlingstein et al., 2022), and it is well confirmed that the quality of CMEMS-LSCE is in line with the others. This manuscript investigates an upgrade of multi-year reconstructions of $pCO_2$ and other carbonate system variables by increasing spatial resolution from 1° to 0.25°, that has not been done in the previous studies.

%%%%%%%%%%%%%%%%%%%%%%%%%%%%%%%%%%%%%%%

**GC2.3.** *The authors use external SST and SSS instead of those incorporated in the datasets in the learning process. This seems unusual because the oceanographic condition represented by temperature and salinity considerably affects the ocean biogeochemistry in the observed area. If the authors think the use of external SST/SSS to be essential, they must demonstrate that the impact of differences between external SST/SSS and those in the datasets is negligible.*

**Authors:**
Our reconstructions require gridded SST and SSS datasets without any gaps, which is not available from SOCAT. We therefore use other data sources**.** Based on statistical assessments, the difference between CMEMS SST and SOCAT SST (or in situ observations) is relatively small (see Figures **GC1.3.** and **MC1.13.** in this document and also Figure A10 in the manuscript). Besides, no quality control has been done for SOCAT salinity. There are some cruises in SOCAT with no salinity data and SOCAT has alternatively used SSS from the World Ocean Atlas (WOA) to recalculate $CO_2$ fugacity (see Pfeil et al, 2013). We have not changed the method.

%%%%%%%%%%%%%%%%%%%%%%%%%%%%%%%%%%%%%%%

**GC2.4.** *In addition, the manuscript seems to contain unnecessary sentences and be lengthened. Shortening the manuscript will increase readability.*

**Authors:**
Thank you. We have shortened the manuscript, in particular based on the reviewer's specific comments.

%%%%%%%%%%%%%%%%%%%%%%%%%%%%%%%%%%%%%%%

**Specific comments by Reviewer 2 (SC2)**

| Reviewer's comments | Replies from Authors | Notes |
|---|---|---|
| **SC2.1. Introduction**
This section seems too long and needs to be shortened. | We have revised the manuscript and removed part of the unnecessary sentences. | |
| **SC2.2. L71**
not only "extrapolate" but also "interpolate". | Thank you. "interpolate" was added. | |
| **SC2.3. L91 Table 1 and Appendix A**
Table 1 only shows six carbonate system variables and is not necessary. Reference to them in the text is enough. In the same context, Appendix A is also unnecessary because it only contains general explanations of carbonate system variables as written in, e.g., Dickson et al. 2007. | We removed both Table 1 and Appendix A. | |

| | | |
|---|---|---|
| **SC2.4. L95-131**
All or a part of these explanations had better be transferred to the beginning of the "3 Reconstruction method" section. | These lines were modified accordingly. | |
| **SC2.5. L135**
Which were used, sea surface height anomaly (SLA) or sea surface dynamic height (SLA+MDT)? Please clarify. | As defined by data providers of CMEMS SSH, we revised the text as follows: "*sea surface height **above geoid***". | |
| **SC2.6. L155**
SOCAT's full name was already mentioned in L66. | We replaced the full name with its abbreviation. | |
| **SC2.7. L160-165**
Using global 0.25 deg binned data derived from SOCAT cruise data is a usual way, even though using 1 deg binned data does not significantly affect the result. | Using a global 0.25° dataset gridded from SOCAT underway measurements for FFNN model training is indeed our ultimate goal. We have contacted SOCAT experts to investigate further how to grid measurements into 0.25° open-ocean datasets knowing that quality control of measurements is critical. | |
| **SC2.8. L213**
It should be clarified how you dealt with longitude and latitude parameters. | To preserve the continuity of longitude at 0°, we have applied the sine and cosine functions to that coordinate. Hence, our global maps of carbonate variables (e.g. Figures 1, 6, 9) do not show discontinuity at the prime meridian. The sine is also used to transform latitude. Data transformation of predictor variables is explicitly presented in a sequence of preceding studies | |

| | for the CMEMS-LSCE-FFNN model development (Denvil-Sommer et al,. 2021, Chau et al,. 2022). In the first manuscript version, we avoided repeating part of the data processing and model description from the previous studies. As the readers would concern, we have called back this information in the revision (Lines 162-164):
 "*The sine function is applied to convert latitude while both the sine and cosine are used to transform longitude to conserve their periodical behaviors*." | |
|---|---|---|
| **SC2.9. Table 4**
 This table contains RMSDs and coefficients of determination, and the name "skill score" is not appropriate. RMSDs of r025 are not significantly different from those of r100 according to Table 4, and therefore the authors should not emphasize an improvement of the prediction skill. The results only show that a fine-scale reconstruction was achieved with no adverse effect. | We have changed "Skill scores" to "**Evaluation statistics**" in the Table caption.
 Table 3* shows a marginal improvement from r100 to r025 in terms of global evaluation metrics. For the open ocean, we expect to obtain similar skill scores for both FFNN models as the spatial autocorrelation of open-ocean $pCO_2$ is estimated within 400±250 km (Jones et al., 2012) and the same SOCAT 1°-open-ocean dataset was used in model fitting. As also reviewed in Chau et al., (2022), $pCO_2$ over the coastal ocean is characterized with high variability at small scales. For instance, $pCO_2$ levels can vary with a horizontal gradient as large as 470 µatm over a distance of less than 0.5 km (Chavez et al., 2018; Feely et al., 2008). Statistical models would need a spatial resolution much finer than 0.25° (25 km) and a temporal resolution higher than monthly in order to capture such high variability in surface ocean $pCO_2$ present in observations (see also Bakker et al., 2016; Laruelle et al., 2017) (see our discussion in Section Summary**).
 Apart from Table 3*, benefits of increasing the model spatial resolution from 1° to 0.25° are also demonstrated in Figures 2-4 with analyses shown in Lines 414-420*** | *Table 4 in the initial manuscript (Table 1 has been removed as suggested by **Review 2**, comment **SC2.2.**)

 ** Section Conclusion and Discussions in the initial manuscript |

| | | |
|---|---|---|
| | "*The two FFNN reconstructions (r025 and r100) share similarities in overall structures of pCO₂ over the coastal-open-ocean continuum (Figs. 2-4). However, the higher spatial resolution outperforms its lower resolution counterpart in reproducing fine-scale features of pCO₂ in the transition from nearshore regions to the adjacent open ocean. The increase in model spatial resolution translates into a greater spatial coverage of the continental shelves such as Labrador Sea, Northern Europe, and Sea of Japan (Fig. 3), and thus an increase in the number of data over the coastal domain. The increase in spatial resolution allows a gain in prediction probability of pCO₂ variations on the order of roughly 2% over the Eastern Boundary Currents to 8% over the Western South Atlantic (Figs. 2-3b).*"

  This study also points out temporal data sampling bias as a source of uncertainty that would highly constrain model reconstruction skills. Based on the assessment at station time series (Figure 5 and Table A3), we found that in situ observations have been sampled with low frequency and the bias of sampling date is about a week from a month center. With the low number of observations and high variability of $pCO_2$ (20.12 to 69.98 μatm) over these stations, it would not be statistically sufficient to refer to their temporal mean as a representative of monthly averages. A large model-data deviation would be retained even if we increase spatial resolution (see text in Lines 421-447 for further analysis). | ***Lines 387-394 in the initial manuscript |
| **SC2.10. Fig 2-4**
The results from the two methods, r100 and r025, have almost the same structure. Please explain the reason why the authors focused on the comparison of them. | As expressed in Line 384-393*, the motivation for a comparison between r100 and r025 in Figures 2-4 to show improvements of horizontal gradients in the higher resolution over different oceanic conditions. The three figures respectively present results in: | *Line 358-367 in the previous version |

| | | |
|---|---|---|
| | ☐ permanent Eastern Boundary current upwelling systems with relatively high $pCO_2$,
☐ regions characterized by low $pCO_2$ values driven by cold water temperatures and strong biological production,
☐ other regions either under the influence of strong river runoff or monsoon-driven upwelling.
It is expected that the two models with different resolutions share the same large-scale structure described above. In Line 413-420** (below), we further analyse the benefits obtained with the higher resolution.
"*The two FFNN reconstructions (r025 and r100) share similarities in overall structures of $pCO_2$ over the coastal-open-ocean continuum (Figs. 2-4). However, the higher spatial resolution outperforms its lower resolution counterpart is reproducing fine-scale features of $pCO_2$ in the transition from nearshore regions to the adjacent open ocean. The increase in model spatial resolution translates into a greater spatial coverage of the continental shelves such as Labrador Sea, Northern Europe, and Sea of Japan (Fig. 3), and thus an increase in the number of data over the coastal domain.*" | **Line 387-391 in the previous version |
| **SC2.11. L374 Fig. 3**
RMSD for the Sea of Japan is suppressed by using data in the subtropical regions (Tsushima warm current area and Kuroshio area) which generally can be estimated more easily. The RMSD must be calculated from data restricted north of the subtropical front. | Thank you for the comment. However, we should use all the available SOCAT data over the Sea of Japan to evaluate RMSD setting the assessment consistent with the other coastal regions. The evaluation over specific sub-basins can be considered in a regional study of $pCO_2$ variability. | |

| | | |
|---|---|---|
| **SC2.12. L403-414**
The discrepancy between the estimated and observed pCO2 not only originated from the timescale but also from the method itself. The method cannot express short-term phenomena inherently because it used external SST and SSS instead of those incorporated in the datasets in the learning process. | We agree with the reviewer. As explained above (replies to comments **GC1.3.** and **GC2.3.**), the use of gridded SST and SSS is a necessity for the reconstruction and there is no significant descrepancy between SOCAT (or in situ) data and CMEMS gridded data. | |
| **SC2.13. 5.2 Total alkalinity and dissolved inorganic carbon**
In the method of this study, discrepancies in estimated and observed DIC were initially derived from pCO2 and TA estimation and propagated via carbonate system calculations. The discussion on uncertainty should be written along with such a concept. | Thank you for this point. We have added the discussion in the revised manuscript (see Lines 471-474*):
"* DIC uncertainty is computed through CO2SYS error propagation with reconstruction uncertainties of $pCO_2$ and $A_T$ set as inputs. The largest values  ($\sigma > 30\ \mu mol\ kg^{-1}$) appear  nearshore and surrounding oceanic islands (Fig. 6d). A similar feature is found on the field of $A_T$ (Fig. 6c),  inherited from input uncertainty associated with the CMEMS salinity product (Fig. A8a).*" | *Lines 437-438 in the initial manuscript |
| **SC2.14. L467-469**
The authors attributed a large σ of DYFAMED estimates to a limited number of observations in the Mediterranean, but GLODAPv2 includes alkalinity measurement data in the Mediterranean. Schneider et al. 2007 successfully derived | There exists indeed a few observations over the surface Mediterranean Sea in GLODAPv2.2022 (Lauvset et al., 2022) used for model evaluation (see f.i. Figure 8 in the manuscript), and even lower data density in GLODAPv2 (Olsen et al., 2016) used for LIAR coefficients fits (Carter et al. 2018). As illustrated in Figure **SC2.14.**, the bias between CMEMS SSS and observations is tiny compared to that of LIAR $A_T$ and observations. The paucity of | |

| | | |
|---|---|---|
| the salinity-alkalinity relationship. The discrepancy in DYFAMED seems to be attributable to salinity discrepancy only. | GLODAPv2 data over the Mediterranean Sea and the distinction in AT-SSS relationship over this region from other basins would lead to a biased LIAR estimate of $A_T$ (-145.1 µmol kg⁻¹). Schneider et al., (2007) derived the estimates of $A_T$-SSS relationship by using local observations, but the estimated $A_T$ is also subject to a large error range (±114.94 µmol kg⁻¹), see Eq 1 quoted below:

$$A_T = 73.7(\pm 3.0) \cdot S - 285.7(\pm 114.94)\,\mu mol\ kg^{-1}$$

**Figure SC2.14**. Monthly time series of $A_T$ (Figure 7) and SSS (Figure A10) at DYFAMED.

[Figure]

We have added one sentence in Lines 508-510* for clarification. "***Although the bias between reanalysed SSS and observations (Fig. A10) is relatively small (-0.15 µmol kg⁻¹), LIAR (Carter et al., 2018) was trained on GLODAPv2 (Olsen et al., 2016) including only few observations in this area. The distinct relationship between alkalinity and salinity prevailing in*** | *Lines 467-469 in the initial manuscript |

| | |
|---|---|
| | *the Mediterranean Sea is likely not reproduced by LIAR leading to an underestimation of $A_T$ and a systematic bias to DIC at DYFAMED (Fig. 7).*" | |

| | | |
|---|---|---|
| **SC2.15. L539-540**
I think that SDG indicator 14.3.1, "Average marine acidity (pH) measured at agreed suite of representative sampling stations", is worth mentioning here. Global mean pH based on observation can be a proxy for the indicator. In addition, it is also valuable information that the global mean pH becomes 8.0 with one decimal place, not 8.1 often said. | Thank you. In lines 686-689, we have mentioned the SDG 14.3.1 indicator for ocean acidification.
*"The global maps of CMEMS-LSCE pH, Ω, and their trend estimates would be potential indicators for ocean acidification along with the SDG 14.3.1 - "Average marine acidity (pH) measured at agreed suite of representative sampling stations" (https://sdgs.un.org/goals/goal14: last access 31/07/2023)."*
However, the global mean CMEMS pH over 1985-2022 is about 8.082 (Table 6*). With one decimal, it becomes 8.1, the same value as reported previously. | *Table 7 in the initial manuscript (Table 1 has been removed as suggested by **Review 2**, comment **SC2.2.**) |
| **SC2.16. 6. Conclusion and discussion**
This section had better be titled "Summary". It does not seem to include discussion. | Thank you. We have modified the section title as suggested. | |

**Other changes**:

- Data repository: the following sentence is added at the end of the Introduction (Lines 136-138) to make data repository visible to the users (details of data access can be found in Section Data availability):

  "*The high-resolution data product described in this manuscript (netCDF format) can be accessed via repository under data DOI: 10.14768/a2f0891b-763a-49e9-af1b-78ed78b16982.*"

- Contribution of GLODAP for this study: we add one sentence in Acknowledgement.

  "*The Global Ocean Data Analysis Project (GLODAP, www.glodap.info, last access: 21 August 2023) provides access to ocean surface-to-bottom quality controlled data of carbonate system variables collected through international cruises.*"

- Typo errors / references:  they are corrected / updated in the revised manuscript.

[revised manuscript text omitted]

**Appendix A:**

725

$$\underline{\text{CO}_2(\text{g})} \quad\quad \rightleftharpoons \text{CO}_2(\text{aq}),$$
$$\underline{\text{CO}_2(\text{aq}) + \text{H}_2\text{O}} \quad \rightleftharpoons \text{H}^+(\text{aq}) + \text{HCO}_3^-(\text{aq}),$$
$$\underline{\text{HCO}_3^-(\text{aq})} \quad \rightleftharpoons \text{H}^+(\text{aq}) + \text{CO}_3^{2-}(\text{aq}),$$

730

i) Surface ocean $p\text{CO}_2$ is partial pressure of $\text{CO}_2$ in air which is in equilibrium with that in water sample. It is not the same as surface ocean fugacity of $\text{CO}_2$ ($f\text{CO}_2$). $p\text{CO}_2$ can be converted from $f\text{CO}_2$ via

$$p\text{CO}_2 = f\text{CO}_2 \, \exp\left(-P\frac{B+2\delta}{RT^*}\right).$$

where $P$ is total atmospheric pressure at surface water, $T^*$ is absolute temperature, $R$ is the gas constant, and $B$ and $\delta$ are cross-virial coefficients (Körtzinger, 1999).

ii) Seawater $p\text{H}$ is a negative logarithmic scale of total concentration of hydrogen ions ($\text{H}^+$) in aqueous solution. Total $\text{H}^+$ is the sum of concentrations of free $\text{H}^+$ and $\text{HSO}_4$ ions. The $p\text{H}$ scale typically ranges from 0 to 14. $p\text{H} = 7$ is the threshold specifying whether a water sample is in acidic (i.e., $p\text{H} < 7$) or basic (i.e., $p\text{H} > 7$) conditions.

[revised manuscript text omitted]

---

## Editor Decision (ED1)

I fully agree with both reviewers on the relevance of the generated product and the quality of the work and the paper itself,

However, I also agree with reviewers comments on the needs of keeping $f$CO2 together with its measurement temperatures. Figure GC1.3 shows a perfect unbiased fit between CMEMS and SOCAT SST but significant scatter is also present, with many values outside the +-1ºC range, so significative in terms pf pCO2 that recommends at least 0.02ºC precision on Temperature (Dickson 2007). A plot of diffs can show this better than the property vs property. However, I can also agree with authors that an extra NN step to fit this would add complexity and probably noise. In my opinion, the manuscript accuracy would be greatly improved by just re-scaling the fCO2 inputs at SOCAT temperatures to CMEMS temperatures according Wanninkhof et al., (2022) or using the MCS equations.

A basic conversion recipe (matlab):

```
co2=CO2SYSv3(400,2300,5,1,35,20,21,0,0,10,1,0,0,1,10,1,2,2); fCO2_out=co2(:,23)
```

or (python):

```
fCO2_out = pyco2.sys(par1=400, par1_type=5, temperature=20, temperature_out=21)['fCO2_out']
```

for example, for converting from 20ºC to 21ºC would be preferable to no conversion at all.

Another detail that I missed on the manuscript is the proper statement on the sets of constants used in the MCS, as well as the pH Scale and conditions. Including those details is a key point for solving the MCS thermodynamics, and so, key for future usage and proper understanding of the dataset. This is particularly needed when computing one parameter from a pair, as you do with pH from fCO2, AT pairs. Apart from the Lewis and so cites that you use, it's convenient to cite the actual toolbox and exact version you are using, be CO2SYS (ven Heuven), v2 (Orr) or v3 (Sharp), or python (Humphreys), as implementations vary. The matlab example in the paragraph above uses CO2SYS v3.2.1, K1&K2 from Lueker, KSO4 from Dickson, KF of Perez and Fraga and TB from Lee, which can be considered the prefered default set to many authors for common oceanic waters right now (but differs from the default set in python version)

And just as final remark, I ask you to review the notation on symbols. Whether the IUPAC would encourage to use pH, $p$CO$_2$, $f$CO$_2$, $A_T$, $C_T$, $SSS$,$T$,.. i.e. the first in italics and the later straight except for common abbreviations, it's also common the usage of $A_T$, $C_T$, pCO2,... but is quite unusual to read $p$H or $A_T$ as I've seen in the manuscript. It's also preferable to use just only acronym for a specific parameter, say $C_T$ or DIC, but not both in same manuscript.

---

## Author Response (AR2)

**Response to the Editor's comments on the manuscript:**

**CMEMS-LSCE: A global 0.25-degree, monthly reconstruction of the surface ocean carbonate system**

**T. T. T. Chau, M. Gehlen, N. Metzl, F. Chevallier**

The authors thank the editor and the two reviewers for their contributions to enhance this study. Further, we appreciate your positive evaluation and that our manuscript has been accepted for publication in Earth Science System Data subject to technical corrections. Please kindly find our replies to the editor's comments (EC) below.

**EC1.** *I fully agree with both reviewers on the relevance of the generated product and the quality of the work and the paper itself.*

*However, I also agree with reviewers comments on the needs of keeping fCO2 together with its measurement temperatures. Figure GC1.3 shows a perfect unbiased fit between CMEMS and SOCAT SST but significant scatter is also present, with many values outside the +-1ºC range, so significance in terms pf pCO2 that recommends at least 0.02ºC precision on Temperature (Dickson 2007). A plot of diffs can show this better than the property vs property. However, I can also agree with authors that an extra NN step to fit this would add complexity and probably noise.*

*In my opinion, the manuscript accuracy would be greatly improved by just re-scaling the fCO2 inputs at SOCAT temperatures to CMEMS temperatures according Wanninkhof et al., (2022) or using the MCS equations.*

*A basic conversion recipe (matlab): co2=CO2SYSv3(400,2300,5,1,35,20,21,0,0,10,1,0,0,1,10,1,2,2); fCO2_out=co2(:,23)*

*or (python): fCO2_out = pyco2.sys(par1=400, par1_type=5, temperature=20, temperature_out=21)['fCO2_out'] for example, for converting from 20°C to 21°C would be preferable to no conversion at all.*

**Authors:**

We thank the Editor for his suggestions to « re-scale the $fCO_2$ inputs at SOCAT temperatures to CMEMS temperatures ». We would like to take advantage of this reply to reemphasize the importance of homogeneity in predictor data during training, validation, and reconstruction. Please note that we use the gridded version of the SOCAT $fCO_2$ throughout the process.

SOCAT accepts SST data with a precision of 0.05°C for flags A and B and 0.2°C for flags C and D (Lauvset et al., 2019). These values are substantially higher than the precision of 0.02°C recommended by Dickson et al., (2007). Furthermore, SST data are not associated with SOCAT quality control (QC) flags for $fCO_2$ (WOCE). SST-based adjustments will add large uncertainty to $fCO_2$ reconstructions if its input data are not fully qualified or have high sampling uncertainty. These data are next interpolated on the 1°x1° grid along with corresponding $fCO_2$ data and yield the gridded version of SOCAT. This is a major source of uncertainty.

We don't intend to downplay the problem which is well recognized by the community and has been addressed in recent publications (e.g. Woolf et al., 2019; Watson et al., 2020). A simple rescaling of SOCAT temperatures to CMEMS temperatures would, however, not solve it. In Wanninkhof et al., (2022), the authors mention in Section Introduction (4$^{th}$-paragraph) that adjustments with differences in SST falling out of ±1°C are proposed with the available $A_T$ and $C_T$ data (the same requirement holds for $fCO_2$ rescaling by using the MCS equations). However, SOCAT does not provide data for these two variables. One would have to rely on GLODAP subsurface measurements, co-locating them on the SOCAT 1°×1° and correcting them for surface temperatures and salinity. Resulting values of $A_T$ and $C_T$ would carry substantial uncertainty making them of only limited use to constrain $fCO_2$.

We hope that the preceding convinces the editor that there is no straightforward (simple) solution to the problem. It is our opinion that the topic should be addressed by the SOCAT and SOCCOM communities in order to propose an approach common to all mapping methods.

**EC2.** *Another detail that I missed on the manuscript is the proper statement on the sets of constants used in the MCS, as well as the pH Scale and conditions. Including those details is a key point for solving the MCS thermodynamics, and so, key for future usage and proper understanding of the dataset. This is particularly needed when computing one parameter from a pair, as you do with pH from fCO2, AT pairs. Apart from the Lewis and so cites that you use, it's convenient to cite the actual toolbox and exact version you are using, be CO2SYS (ven Heuven), v2 (Orr) or v3 (Sharp), or python (Humphreys), as implementations vary. The matlab example in the paragraph above uses CO2SYS v3.2.1, K1&K2 from Lueker, KSO4 from Dickson, KF of Perez and Fraga and TB from Lee, which can be considered the prefered default set to many authors for common oceanic waters right now (but differs from the default set in python version)*

**Authors:**
Knowing the necessity of providing the information on constants and conditions used in the CO2SYS speciation for this study, we have thoroughly presented these elements in "**Section 3.3 Carbonate system speciation**" - see **Lines 262-273** in the previous revision (quoted in blue below). For pH, data have been reported on *total scale* (see for instance Section Results). We however admit that this information should be stated starting from Section Introduction (please find our correction in the new revised manuscript). In addition to this revision, we cite correctly the Matlab version of CO2SYS used in this study: **CO2SYS.v2** (Orr et al., 2018).

 **Lines 262-273** in the previous revision:
"The FFNN best estimate (ensemble mean) of $pCO_2$ reconstructions (Sect. 3.1) and the LIAR outputs of $A_T$ (Sect. 3.2) are used as the prior inputs of the CO2SYS at each grid cell for every month in the period 1985-2021. We take the same data products of SST, SSS, and nutrient concentrations as for the previous reconstructions (Table 1). Pressure (P) is assumed to be 0

dbar at the ocean surface. For equilibrium constants, we choose the best empirical values recommended by Dickson et al. (2007) and Dickson (2010). These settings include (1) the dissociation constants $K_1$ and $K_2$ from Lueker et al. (2000) and $K_{HSO4}$ from Dickson (1990) in combination with the total boron-ratio-salinity formulation by Uppstrom (1974).

The uncertainty of the CO2SYS variables is estimated by error propagation (Orr et al., 2018). Inputs for the CO2SYS error propagation include the reconstruction uncertainty of $p$CO$_2$ (FFNN ensemble standard deviation) and of $A_T$ (LIAR error propagation). The uncertainty of SST, SSS, and nutrient concentrations are set to the same values as in the previous section (Sect. 3.2). Equilibrium constants' standard errors are default values (see Table 1, Orr et al., 2018). As for FFNN and LIAR, uncertainty values of each carbonate system variable are computed for each month in 1985-2021 and at each 0.25° -grid box over the global surface ocean."

**EC3.** *And just as final remark, I ask you to review the notation on symbols. Whether the IUPAC would encourage to use of* pH*,* $p$CO$_2$*,* $f$CO$_2$*,* $A_T$*,* $C_T$*, SSS, T,.. i.e. the first in italics and the later straight except for common abbreviations, it's also common the usage of* $A_T$*,* $C_T$*,* pCO2*,... but is quite unusual to read* $p$H *or* $A_T$ *as I've seen in the manuscript. It's also preferable to use just only acronym for a specific parameter, say* $C_T$ *or* DIC*, but not both in same manuscript.*

**Authors:**
We have corrected the acronyms of pH and $A_T$ following the Editor's comments. For dissolved inorganic carbon, we now denote this variable as $C_T$ consistently throughout the text.

**References**

Dickson, A. G.: Standard potential of the reaction -AgCl(s)+1/2H-2(g)=Ag(s)+HCl(aq) and the standard acidity constant of the ion HSO4 in synthetic sea-water from 273.15-K to 318.15-K, Journal of Chemical Thermodynamics, 22, 113–127, https://doi.org/doi:10.1016/0021-9614(90)90074-z, 1990.

Dickson, A. G., Sabine, C. L., and Christian, J. R.: Guide to best practices for ocean CO2 measurements., North Pacific Marine Science Organization, 2007.

Dickson, A. G.: The carbon dioxide system in seawater: equilibrium chemistry and measurements in Guide to best practices for ocean acidification research and data reporting, https://www.pmel.noaa.gov/co2/files/dickson_thecarbondioxidesysteminseawater_equilibrium chemistryandmeasurementspp17-40.pdf, 2010.

Lauvset S., K. Currie, N. Metzl, S. Nakaoka, D. Bakker, K. Sullivan, A. Sutton, K. O'Brien and A. Olsen. SOCAT Quality Control Cookbook for SOCAT version 7. Int. Report. www.socat.info, 2019.

Lueker, T. J., Dickson, A. G., and Keeling, C. D.: Ocean pCO2 calculated from dissolved inorganic carbon, alkalinity, and equations for K1 and K2: validation based on laboratory measurements of CO2 in gas and seawater at equilibrium, Marine chemistry, 70, 105–119, 2000.

Orr, J. C., Epitalon, J.-M., Dickson, A. G., and Gattuso, J.-P.: Routine uncertainty propagation for the marine carbon dioxide system, Marine Chemistry, 207, 84–107, https://doi.org/https://doi.org/10.1016/j.marchem.2018.10.006, 2018.

Uppstrom, L.: The boron/chlorinity ratio of deep-sea water from the Pacific Ocean, Deep Sea Res., 21, 161–162, 1974.

Wanninkhof, R., Pierrot, D., Sullivan, K., Mears, P. and Barbero, L.. Comparison of discrete and underway CO2 measurements: Inferences on the temperature dependence of the fugacity of CO2 in seawater. *Marine chemistry*, *247*, p.104178, 2022.

Watson, A.J., Schuster, U., Shutler, J.D. *et al.* Revised estimates of ocean-atmosphere CO2 flux are consistent with ocean carbon inventory. *Nat Commun* 11, 4422 (2020). https://doi.org/10.1038/s41467-020-18203-3

Woolf, D. K., Shutler, J. D., Goddijn-Murphy, L., Watson, A. J., Chapron, B., Nightingale, P. D., et al. Key uncertainties in the recent air-sea flux of CO2. *Global Biogeochemical Cycles*, 33, 1548–1563. https://doi.org/10.1029/2018GB006041, 2019.